# Balance is Essence: Accelerating Sparse Training via Adaptive Gradient Correction

## Abstract

Despite impressive performance on a wide variety of tasks, deep neural networks require significant memory and computation costs, prohibiting their application in resource-constrained scenarios. Sparse training is one of the most common techniques to reduce these costs, however, the sparsity constraints add difficulty to the optimization, resulting in an increase in training time and instability. In this work, we aim to overcome this problem and achieve space-time co-efficiency. To accelerate and stabilize the convergence of sparse training, we analyze the gradient changes and develop an adaptive gradient correction method. Specifically, we approximate the correlation between the current and previous gradients, which is used to balance the two gradients to obtain a corrected gradient. Our method can be used with most popular sparse training pipelines under both standard and adversarial setups. Theoretically, we prove that our method can accelerate the convergence rate of sparse training. Extensive experiments on multiple datasets, model architectures, and sparsities demonstrate that our method outperforms leading sparse training methods by up to **5.0%** in accuracy given the same number of training epochs, and reduces the number of training epochs by up to **52.1%** to achieve the same accuracy.

## 1 Introduction

With the development of deep neural networks (DNNs), there is a trend towards larger and more intensive computational models to enhance task performance. Despite of the good performance, such large models are not applicable when memory or computational resources are limited (Bellec et al., 2018; Evci et al., 2020; Liu et al., 2022). In addition, these large models consume a considerable amount of energy and produce a large amount of carbon footprint (Thompson et al., 2021; Patterson et al., 2021; Matus & Veale, 2022). As a result, it attracts more efforts in research to find resource-efficient ways (e.g., less memory & less compute) to train DNNs while maintaining results comparable to the state of the art (Yu & Li, 2021; Rock et al., 2021; Leite & Xiao, 2021).

Sparse training (Mocanu et al., 2018; Evci et al., 2020; Liu et al., 2022) is one of the most popular classes of methods to improve efficiency in terms of space (e.g. memory storage) and is receiving increasing attention. During sparse training, a certain percentage of connections are removed to save memory (Bellec et al., 2018; Evci et al., 2020). Sparse patterns, which describe where connections are retained or removed, are iteratively updated with various criteria (Dettmers & Zettlemoyer, 2019; Evci et al., 2020; Liu et al., 2021; Özdenizci & Legenstein, 2021). The goal is to find a resource-efficient sparse neural network (i.e., removing some connections) with comparable or even higher performance compared to the original dense model (i.e., keeping all connections).

However, sparse training can bring some side effects to the training process, especially in the case of high sparsity (e.g., 99% weights are zero). First, sparsity can increase the variance of stochastic gradients, leading the model to move in a sub-optimal direction and hence slow convergence (Hoefler et al., 2021; Graesser et al., 2022). As shown in Figure 1 (a), we empirically see that the gradient variance grows with increasing sparsity (more details in Section C.1). Second, it can result in training instability (i.e., a noisy trajectory of test accuracy w.r.t. iterations) (Sehwag et al., 2020; Bartoldson et al., 2020), which requires additional time

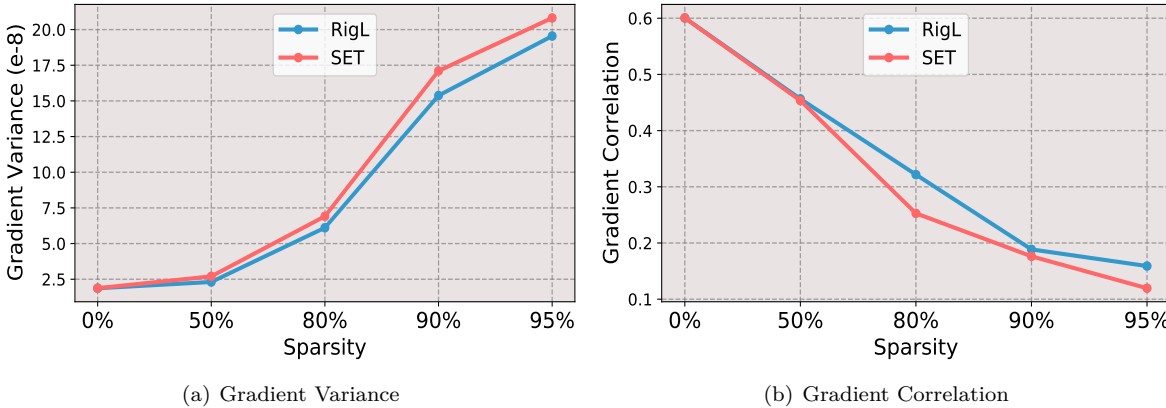

(a) Gradient Variance

(b) Gradient Correlation

Figure 1: Gradient variance (a) and gradient correlation (b) of models obtained by RigL and SET at different sparsities including 0% (dense), 50%, 80%, 90%, 95%. Gradient variance grows with increasing sparsity. Gradient correlation drops with increasing sparsity. The sparse models have larger gradient variance and smaller gradient correlation compared to dense models.

to compensate for the accuracy drop, resulting in slow convergence (Xiao et al., 2019). Additionally, the need to consider the robustness of the model during sparse training is highlighted in order to apply sparse training to a wide range of real-world scenarios where there are often challenges with dataset shifts (Ye et al., 2019; Hoefler et al., 2021; Kundu et al., 2021; Özdenizci & Legenstein, 2021). To address these issues, we raise the following questions:

**Question 1.** *How to simultaneously improve convergence speed and training stability of sparse training?*

Prior gradient correction methods, such as variance reduction (Zou et al., 2018; Chen et al., 2019; Gorbunov et al., 2020), are used to accelerate and stabilize dense training, while we find that it fails in sparse training. They usually assume that current and previous gradients are highly correlated, and therefore they add a large constant amount of previous gradients to correct the gradient (Dubey et al., 2016; Chatterji et al., 2018; Chen et al., 2019). However, this assumption does not hold in sparse training. Figure 1 (b) shows the gradient correlation at different sparsities, implying that the gradient correlation decreases with increasing sparsity (more details in Section C.1), which breaks the balance between current and previous gradients. Therefore, we propose to adaptively change the weights of previous and current gradients based on their correlation to add an appropriate amount of previous gradients.

**Question 2.** *How to design an accelerated and stabilized sparse training method that is effective in real-world scenarios with dataset shifts?*

Moreover, real-world applications are under-studied in sparse training. Prior methods use adversarial training to improve model robustness and address the challenge of data shifts, which usually introduces additional bias beyond the variance in the gradient estimation (Li et al., 2020), increasing the difficulty of gradient correction (more details in Section 4.2). Thus, to more accurately approximate the full gradient, especially during the adversarial setup, we design a scaling strategy to control the weights of the two gradients, determining the amount of previous gradient information to be added to the current gradient, which helps the balance and further accelerates the convergence.

In this work, we propose an **a**daptive **g**radi**en**t correc**t**ion (AGENT) method to accelerate and stabilize sparse training for both standard and adversarial setups. Theoretically, we prove that our method can accelerate the convergence rate of sparse training. Empirically, we perform extensive experiments on multiple benchmark datasets, model architectures, and sparsities. In both standard and adversarial setups, our method improves the accuracy by up to **5.0%** given the same number of epochs and reduces the number of epochs up to **52.1%** to achieve the same performance compared to the leading sparse training methods. In contrast to previous efforts of sparse training acceleration which mainly focus on structured sparse patterns, our method

is compatible with both unstructured ans structured sparse training pipelines (Hubara et al., 2021; Chen et al., 2021).

## 2 Related Work

### 2.1 Sparse Training

Interest in sparse DNNs has been on the rise recently, especially when dealing with resource constraints. The goal is to achieve comparable performance with sparse weights to satisfy the constraints. Different sparse training methods have emerged, where sparse weights are maintained in the training process. Various pruning and growth criteria are proposed, such as weight/gradient magnitude, random selection, and weight sign (Mocanu et al., 2018; Bellec et al., 2018; Frankle & Carbin, 2019; Mostafa & Wang, 2019; Dettmers & Zettlemoyer, 2019; Evci et al., 2020; Jayakumar et al., 2020; Liu et al., 2021; Özdenizci & Legenstein, 2021; Zhou et al., 2021b; Schwarz et al., 2021; Huang et al., 2022; Liu et al., 2022).

However, the aforementioned studies focus on improving the performance of sparse training, while neglecting the side effect of sparse training. Sparsity not only increases gradient variance, thus delaying convergence (Hoefler et al., 2021; Graesser et al., 2022), but also leads to training instability (Bartoldson et al., 2020). It is a challenge to achieve both space and time efficiency. Additionally, sparse training can also exacerbate models' vulnerability to adversarial samples, which is one of the weaknesses of DNNs (Özdenizci & Legenstein, 2021). When the model encounters intentionally manipulated data, its performances may deteriorate rapidly, leading to increasing security concerns Rakin et al. (2019); Akhtar & Mian (2018). In this paper, we focus on sparse training. In general, our method can be applied to any SGD-based sparse training pipelines.

### 2.2 Accelerating Training

Studies have been conducted in recent years on how to achieve time efficiency in DNNs, and one popular direction is to obtain a more accurate gradient estimate to update the model (Gorbunov et al., 2020), such as variance reduction. SGD is the most common training method, where one uses small batches of data to approach the full gradient. In standard training, the batch estimator is unbiased, but can have a large variance and misguide the model, leading to studies on variance reduction (Roux et al., 2012; Johnson & Zhang, 2013; Defazio et al., 2014; Xiao & Zhang, 2014; Shang et al., 2018; Zou et al., 2018; Chen et al., 2019; Gorbunov et al., 2020; Gower et al., 2020). While adversarial training brings bias in the gradient estimation (Li et al., 2020), and we need to face the bias-variance tradeoff when doing gradient correction. A shared idea is to balance the gradient noise with a less-noisy old gradient (Nguyen et al., 2017; Fang et al., 2018; Chen et al., 2019). Some other momentum-based methods have a similar strategy of using old information (Cutkosky & Orabona, 2019; Chayti & Karimireddy, 2022) However, all the above work considers only the acceleration in non-sparse case.

Acceleration is more challenging in sparse training, and previous research on it has focused on structured sparse training (Hubara et al., 2021; Chen et al., 2021; Zhou et al., 2021a). First, sparse training will induce larger variance (Hoefler et al., 2021). In addition, some key assumptions associated with gradient correction methods do not hold under sparsity constraint. In the non-sparse case, the old and new gradients are assumed to be highly correlated, so we can collect a large amount of knowledge from the old gradients (Chen et al., 2019; Chatterji et al., 2018; Dubey et al., 2016). However, sparsity tends to lead to lower correlations, and this irrelevant information can be harmful, making previous methods no longer applicable to sparse training and requiring a finer balance between new and old gradients. Furthermore, the structured sparsity pattern is not flexible enough, which can lead to lower model accuracy. In contrast, our method accelerates sparse training from an optimization perspective and is compatible with both unstructured and structured sparse training pipelines.

# 3 Preliminaries: Stochastic Variance Reduced Gradient

Stochastic variance reduced gradient (SVRG) (Johnson & Zhang, 2013; Allen-Zhu & Hazan, 2016; Dubey et al., 2016) is a widely-used gradient correction method designed to obtain more accurate gradient estimates, which has been followed by many studies (Zou et al., 2018; Baker et al., 2019; Chen et al., 2019). Specifically, after each epoch of training, we evaluate the full gradients $\widetilde{g}$ based on $\widetilde{\theta}$ at that time and store them for later use. In the next epoch, the batch gradient estimate on $\mathbf{B}_t$ is updated using the stored old gradients via Eq. (1).

$$\hat{g}(\theta_t) = \frac{1}{n} \sum_{i \in \mathbf{B}_t} \left( g_i(\theta_t) - g_i(\widetilde{\theta}) \right) + \widetilde{g} \tag{1}$$

where $g_i(\theta_t) = \nabla l(\mathbf{x}_i|\theta_t)$, $l(\theta_t) = (\sum_{i=1}^{N} l(\mathbf{x}_i|\theta_t))/N$ is the loss function, $\widetilde{g} = \nabla l(\widetilde{\theta})$, $\theta_t$ is the current parameters, $n$ is the number of samples in each mini-batch data, and $N$ is the total number of samples. SVRG successfully accelerates many training tasks in the non-sparse case, but does not work well in sparse training, which is similar to many other gradient correction methods.

# 4 Method

We propose an **a**daptive **g**radi**en**t correc**t**ion (AGENT) method and integrate it with recent sparse training pipelines to achieve accelerations and improve training stability. Specifically, to accomplish the goal, our AGENT filters out less relevant information and obtains a well-controlled and time-varying amount of knowledge from the old gradients. Our method overcomes the limitations of previous acceleration methods such as SVRG (Allen-Zhu & Hazan, 2016; Dubey et al., 2016; Elibol et al., 2020), and successfully accelerates and stabilizes sparse training. We will illustrate each part of our method in the following sections. Our AGENT method is outlined in Algorithm 1.

## 4.1 Adaptive Control over Old Gradients

In AGENT, we designed an adaptive addition of old gradients to new gradients to filter less relevant information and achieve a balance between new and old gradients. Specifically, we add an adaptive weight $c_t \in [0, 1]$ to the old gradient as shown in Eq. (2), where we use $g_{\mathrm{new}} = \frac{1}{n} \sum_{i \in \mathbf{B}_t} g_i(\theta_t)$ and $g_{\mathrm{old}} = \frac{1}{n} \sum_{i \in \mathbf{B}_t} g_i(\widetilde{\theta})$ to denote the gradient on current parameters $\theta_t$ and previous parameters $\widetilde{\theta}$ for a random subset $\mathbf{B}_t$, respectively. When the old and new gradients are highly correlated, we need a large $c$ to get more useful information from the old gradient. Conversely, when the relevance is low, we need a smaller $c$ so that we do not let irrelevant information corrupt the new gradient.

$$\hat{g}(\theta_t) = \frac{1}{n} \sum_{i \in \mathbf{B}_t} \left( g_i(\theta_t) - c_t \cdot g_i(\widetilde{\theta}) \right) + c_t \cdot \widetilde{g} = g_{\mathrm{new}} - c_t \cdot g_{\mathrm{old}} + c_t \cdot \widetilde{g}. \tag{2}$$

A suitable $c_t$ should effectively reduce the variance of $\hat{g}(\theta_t)$. To understand how $c_t$ influence the variance of the updated gradient, we decompose the variance of $\hat{g}(\theta_t)$ in Eq. (3) with some abuse of notation, where the variance of the updated gradient is a quadratic function of $c_t$. For simplicity, considering the case where $\hat{g}(\theta_t)$ is a scalar, the optimal $c_t^*$ will be in the form of Eq. (3). As we can see, $c_t^*$ is not close to 1 when the new gradient is not highly correlated with the old gradient. Since low correlation between $g_{\mathrm{new}}$ and $g_{\mathrm{old}}$ is more common in sparse training, directly setting $c_t = 1$ in previous methods is not appropriate and we need to estimate adaptive weights $c_t^*$. In support of this claim, we include a discussion and empirical analysis in the Appendix B.6 to demonstrate that as sparsity increases, the gradient changes faster, leading to lower correlations between $g_{\mathrm{new}}$ and $g_{\mathrm{old}}$.

$$\mathrm{Var}(\hat{g}(\theta_t)) = \mathrm{Var}(g_{\mathrm{new}}) + c_t^2 \cdot \mathrm{Var}(g_{\mathrm{old}}) - 2c_t \cdot \mathrm{Cov}(g_{\mathrm{new}}, g_{\mathrm{old}}), \quad c_t^* = \frac{\mathrm{Cov}(g_{\mathrm{new}}, g_{\mathrm{old}})}{\mathrm{Var}(g_{\mathrm{old}})}. \tag{3}$$

We find it impractical to compute the exact $c_t^*$ and thus propose an approximation algorithm for it to obtain a balance between the new and old gradient. There are two challenges to calculate the exact $c_t^*$. On the

one hand, to approach the exact value, we need to calculate the gradients on every batch data, which is too expensive to do it in each iteration. On the other hand, the gradients are often high-dimensional and the exact optimal $c_t^*$ will be different for different gradients. Thus, inspired by Deng et al. (2020), we design an approximation algorithm that makes good use of the loss information and leads to only a small increase in computational effort. More specifically, we estimate $c_t^*$ according to the changes of loss as shown in Eq. (4) and update $\widehat{c}_t^*$ adaptively before each epoch using momentum. Loss is a scalar, which makes it possible to estimate the shared correlation for all current and previous gradients. In addition, the loss is intuitively related to gradients and the correlation between losses can give us some insights into that of the gradients (some empirical analyses are included in the Appendix B.7).

$$\widehat{c}_t^* = \frac{\text{Cov}(l(\mathbf{B}|\boldsymbol{\theta}_t), l(\mathbf{B}|\widetilde{\boldsymbol{\theta}}))}{\text{Var}(l(\mathbf{B}|\widetilde{\boldsymbol{\theta}}))}, \tag{4}$$

where $\mathbf{B}$ denotes a subset of samples used to estimate the gradients.

We empirically justify the loss-based approximation in Eq. (4). Experimental details are included in Section B.7). We compare the approximation $\widehat{c}_t^*$ and the correlation between the gradient of current weights and the gradient of previous epoch weights. We find that $\widehat{c}_t^*$ and the correlation have similar up-and-down patterns, indicating that our approximation captures the dynamic patterns of the correlation. For differences in magnitude, they can be matched by the scaling strategy we will describe in the next Section 4.2.

### 4.2 Additional Scaling Parameter is Important

To guarantee successful acceleration in sparse and adversarial training, we further propose a scaling strategy that multiplies the estimated $c_t^*$ by a small scaling parameter $\gamma$. There are two main benefits of using a scaling parameter. First, the scaling parameter $\gamma$ can reduce the bias of the gradient estimates

---

**Algorithm 1** Adaptive Gradient Correction

**Input:** $\widetilde{\boldsymbol{\theta}} = \boldsymbol{\theta}_0$, epoch length $m$, step size $\eta_t$, $c_0 = 0$, scaling parameter $\gamma$, smoothing factor $\alpha$

**for** $t = 0$ **to** $T - 1$ **do**
  **if** $t \bmod m = 0$ **then**
    $\widetilde{\boldsymbol{\theta}} = \boldsymbol{\theta}_t$
    $\widetilde{\boldsymbol{g}} = (\sum_{i=1}^{N} \nabla l(\mathbf{x}_i|\widetilde{\boldsymbol{\theta}}))/N$
    **if** $t > 0$ **then**
      Calculate $\hat{c}_t^*$ via Eq. (4)
      $c_t = (1 - \alpha)c_{t-1} + \alpha\hat{c}_t^*$
    **end if**
  **else**
    $c_t = c_{t-1}$
  **end if**
  Sample a mini-batch data $\mathbf{B}_t$ with size $n$
  $\boldsymbol{\theta}_{t+1} = \boldsymbol{\theta}_t - \eta_t \cdot \left( \frac{1}{n} \sum_{i \in \mathbf{B}_t} \left( \boldsymbol{g}_i(\boldsymbol{\theta}_t) - \gamma c_t \cdot \boldsymbol{g}_i(\widetilde{\boldsymbol{\theta}}) \right) + \gamma c_t \cdot \widetilde{\boldsymbol{g}} \right)$
**end for**

---

in adversarial training (Li et al., 2020). In standard training, the batch gradient estimator is an unbiased estimator of the full gradient. However, in adversarial training, we perturb the mini-batch of samples $\mathbf{B}_t$ into $\bar{\mathbf{B}}_t$. The old gradients $\boldsymbol{g}_{\text{old}}$ are calculated on batch data $\bar{\mathbf{B}}_t$, but the stored old gradients $\widetilde{\boldsymbol{g}}$ are obtained from the original data including $\mathbf{B}_t$, which makes $\mathbb{E}[\boldsymbol{g}_{\text{old}} - \widetilde{\boldsymbol{g}}]$ unequal to zero. Consequently, as shown in Eq. (5), the corrected estimator for full gradients will no longer be unbiased. It may have a small variance but a large bias, resulting in poor performance. Therefore, we propose a scaling parameter $\gamma$ between 0 and 1 to reduce the bias from $c_t(\boldsymbol{g}_{\text{old}} - \widetilde{\boldsymbol{g}})$ to $\gamma c_t(\boldsymbol{g}_{\text{old}} - \widetilde{\boldsymbol{g}})$.

$$\mathbb{E}[\hat{\boldsymbol{g}}(\boldsymbol{\theta}_t)] = \mathbb{E}[\boldsymbol{g}_{\text{new}} - c_t(\boldsymbol{g}_{\text{old}} - \widetilde{\boldsymbol{g}})] \neq \mathbb{E}[\boldsymbol{g}_{\text{old}}] = \frac{1}{N} \sum_{i=1}^{N} \boldsymbol{g}_i(\boldsymbol{\theta}_t). \tag{5}$$

Second, the scaling parameter $\gamma$ guarantees that the variance can still be reduced in the face of worst-case estimates of $c_t^*$ to accelerate the training. The key idea is illustrated in Figure 2, where x and y axis correspond to the weight $c_t$ and the gradient variance, respectively. The blue curve is a quadratic function that represents the relationship between $c_t$ and the variance. Suppose the true optimal is $c^*$, and we make an approximation to it. In the worst case, this approximation may be as bad as $\hat{c}_1$, making the variance even larger than $a_3$ (variance in SGD) and slowing down the training. Then, if we replace $\hat{c}_1$ with $\gamma\hat{c}_1$, we can reduce the variance and accelerate the training.

### 4.3 Connection to Momentum-based Method

To some extent, our AGENT is designed with a similar idea to the momentum-based method (Qian, 1999; Ruder, 2016), where old gradients are used to improve the current batch gradient. However, the momentum-based method still suffers from optimization difficulties due to sparsity constraints. The reason is that it does not take into account sparse and adversarial training characteristics such as the reduced correlation between current and previous gradients and potential bias of gradient estimator, and fails to provide an adaptive balance between old and new information. When the correlation is low, the momentum-based method can still incorporate too much of the old information and increase the gradient variance or bias. In contrast, our AGENT is designed for sparse and adversarial training and can establish finer adaptive control over how much information we should take from the old to help the new.

### 4.4 Connection to Adaptive Gradient Method

Our AGENT can be viewed as a new type of adaptive gradient method that adaptively adjusts the amount of gradient information used to update parameters, such as Adam (Kingma & Ba, 2014). However, previous adaptive gradient methods are not designed for sparse training. Although they also provide adaptive gradients, their adaptivity is different and does not take the reduced correlation into account. On the contrary, our AGENT is designed for sparse training and is tailored to the characteristics of sparse training. When old information is used to correct the gradients, the main problem is the reduced correlation between the old and new gradients. Therefore, our AGENT approximates the correlation and adds an adaptive weight to the old gradient to establish a balance between the old and new gradients.

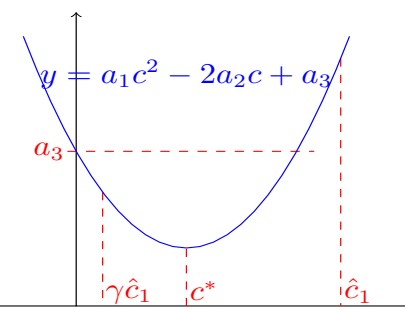

Figure 2: Illustration of how the scaling parameter $\gamma = 0.1$ ensures the acceleration in the face of worst-case estimate of $c_t^*$. The blue curve is a quadratic function, representing the relationship between $c_t$ and the variance. $c^*$ is the optimal value. $\hat{c}_1$ is a poor estimate making the variance larger than $a_3$ (variance in SGD). $\gamma \hat{c}_1$ can reduce the variance.

## 5 Theoretical Justification

Theoretically, we provide a convergence analysis for our AGENT and compare it to SVRG (Reddi et al., 2016). We use $l(.)$ to denote the loss function and $\boldsymbol{g}$ to denote the gradient. Our proof is based on Assumptions 1-2, and detailed derivation is included in Appendix A.

**Assumption 1.** *(L-smooth): The differentiable loss function $l : \mathbb{R}^n \to \mathbb{R}$ is L-smooth, i.e., for all $\boldsymbol{x}, \boldsymbol{y} \in \mathbb{R}^n$, the loss $l$ satisfies $||\nabla l(\boldsymbol{x}) - \nabla l(\boldsymbol{y})|| \leq L||\boldsymbol{x} - \boldsymbol{y}||$. An equivalent definition is that, for all $\boldsymbol{x}, \boldsymbol{y} \in \mathbb{R}^n$, we have:*

$$-\frac{L}{2}||\boldsymbol{x} - \boldsymbol{y}||^2 \leq l(\boldsymbol{x}) - l(\boldsymbol{y}) - \langle \nabla l(\boldsymbol{x}), \boldsymbol{x} - \boldsymbol{y} \rangle \leq \frac{L}{2}||\boldsymbol{x} - \boldsymbol{y}||^2$$

**Assumption 2.** *($\sigma$-bounded): The loss function $l$ has a $\sigma$-bounded gradient, i.e., $||\nabla l_i(\boldsymbol{x})|| \leq \sigma$ for all $i \in [N]$ and $\boldsymbol{x} \in \mathbb{R}^n$.*

**Our convergence analysis framework** outlines four steps:

- We first show that an appropriate choice of $c_t$ will result in smaller variance in our gradient estimates compared to SVRG.

- Next, we show the convergence rate of one arbitrary training epoch.

- We then extend the one-epoch results and analyze the convergence rate for the whole epoch.

- After obtaining the convergence rate, we bring it to the real case of sparse learning and find that our method indeed yields a tighter bound.

Given Assumptions 1-2, we follow the analysis framework above and establish Theorem 1 to show the convergence rate of our AGENT:

**Theorem 1.** *Under Assumptions 1-2, with proper choice of step size $\eta_t$ and $c_t$, the gradient $\mathbb{E}[||g(\boldsymbol{\theta_\pi})||^2]$ using AGENT after $T$ training epochs can be bounded by:*

$$\mathbb{E}[||g(\boldsymbol{\theta_\pi})||^2] \leq \frac{(l(\boldsymbol{\theta}_0) - l(\boldsymbol{\theta}_*))LN^\alpha}{Tn\nu} + \frac{2\kappa\mu^2\sigma^2}{N^\alpha m\nu}$$

*where $\boldsymbol{\theta_\pi}$ is sampled uniformly from $\{\{\boldsymbol{\theta}_t^s\}_{t=0}^{m-1}\}_{s=0}^{T-1}$, $N$ denotes the data size, $n$ denotes the mini-batch size, $m$ denotes the epoch length, $\boldsymbol{\theta}_0$ is the initial point and $\boldsymbol{\theta}_*$ is the optimal solution, $\nu, \mu, \kappa, \alpha > 0$ are constants depending on $\eta_t$ and $c_t$, $N$ and $n$.*

In regard to Theorem 1, we make the following remarks to justify the acceleration from our AGENT:

**Remark 1.** *(Faster Gradient Change Speed) An influential difference between sparse and dense training is the gradient change speed, which is reflected in Assumption 1 (L-smooth). Typically, $L$ in sparse training will be larger than $L$ in dense training.*

**Remark 2.** *(First Term Analysis) In Theorem 1, the first term in the bound of our AGENT measures the error introduced by deviations from the optimal parameters, which goes to zero when the number of epochs $T$ reaches infinity. However, in real sparse training applications, $T$ is finite and this term is expanded due to the increase of $L$ in sparse training, which implies that the optimization under sparse constraints is more challenging.*

**Remark 3.** *(Second Term Analysis) In Theorem 1, the second term measures the error introduced by the noisy gradient and the finite data during the optimization. Since $\sigma^2$ is relatively small and $N$ is usually large in our DNNs training, the second term is negligible or much smaller compared to the first term when $T$ is assumed to be finite.*

From the above analysis, we can compare the bounds of AGENT and SVRG and find that in the case of sparse training, an appropriate choice of $c_t$ can make the bound for our AGENT tighter than the bound for SVRG by well-corrected gradients.

**Remark 4.** *(Comparison with SVRG) Under Assumptions 1-2, the gradient $\mathbb{E}[||g(\boldsymbol{\theta_\pi})||^2]$ using SVRG after $T$ training epochs can be bounded by (Reddi et al., 2016):*

$$\mathbb{E}[||g(\boldsymbol{\theta_\pi})||^2] \leq \frac{(l(\boldsymbol{\theta}_0) - l(\boldsymbol{\theta}_*))LN^\alpha}{Tn\nu^*}.$$

*This bound is of a similar form to the first term in Theorem 1. Since the second term of Theorem 1 is negligible or much smaller than the first one, we only need to compare the first term. With a proper choice of $c_t$, the variance of $\hat{\boldsymbol{g}}(\boldsymbol{\theta}_t)$ will decrease, which leads to a smaller $\nu$ for AGENT than $\nu^*$ for SVRG (detailed proof is included in Appendix A Remark 6). Thus, AGENT can bring a smaller first term compared to SVRG, which indicates that AGENT effectively reduces the error due to the deviations and has a tighter bound compared to SVRG.*

## 6 Experiments

We add our AGENT to four recent sparse training pipelines, namely SET (Mocanu et al., 2018), RigL (Evci et al., 2020), BSR-Net (Özdenizci & Legenstein, 2021) and ITOP (Liu et al., 2021). SET is a broadly-used sparse training method that prunes and regrows connections by examining the magnitude of the weights. RigL is another popular dynamic sparse training method which uses weight and gradient magnitudes to learn the connections. BSR-Net is a recent sparse training method that updates connections by Bayesian sampling and also includes adversarial setups for model robustness. ITOP is another recent pipeline for

dynamic sparse training, which uses sufficient and reliable parameter exploration to achieve in-time over-parameterization and find well-performing sparse models. Detailed information about the dataset, model architectures, and other training and evaluation setups is provided below.

**Datasets & Model Architectures:** The datasets we use include CIFAR-10, CIFAR-100 (Krizhevsky et al., 2009), SVHN (Netzer et al., 2011), and ImageNet-2012 (Russakovsky et al., 2015). For model architectures, we use VGG-16 (Simonyan & Zisserman, 2015), ResNet-18, ResNet-50 (He et al., 2016), and Wide-ResNet-28-4 (Zagoruyko & Komodakis, 2016).

**Training Settings:** For sparse training, we choose two sparsity levels, namely 90% and 99%. For BSR-Net, we consider both standard and adversarial setups. In RigL and ITOP, we focus on standard training. In standard training, we only use the original data to update the parameters instead of using perturbed samples. For adversarial part, we use the perturbed data with two popular objective (AT and TRADES) (Madry et al., 2018; Zhang et al., 2019). Following Özdenizci & Legenstein (2021), we evaluate robust accuracy against PGD attacks with random starts using 50 iterations ($PGD^{50}$) (Madry et al., 2018; Brendel et al., 2019).

**Implementations:** Aligned with the choice of Evci et al. (2020); Sundar & Dwaraknath (2021); Özdenizci & Legenstein (2021), the parameters of the model are optimized by SGD with momentum. Thus, the comparison between the popular sparse training pipelines can be viewed as a comparison between AGENT and momentum-based SGD.

## 6.1 Convergence Speed & Stability Comparisons

We compare the convergence speed by two criteria, including (a) **the test accuracy** at the same number of pass data (epoch) and (b) **the number of pass data (epoch)** required to achieve the same test accuracy, which is widely used to compare the speed of optimization algorithms (Allen-Zhu & Hazan, 2016; Chatterji et al., 2018; Zou et al., 2018; Cutkosky & Orabona, 2019).

**For BSR-Net-based results using test accuracy at the same number of pass data (epoch)**, Tables 1-2 list the accuracies on clean and adversarial samples after 20, 40, 70, 90, 140, and 200 epochs of training, where the higher accuracies are bolded. Sparse VGG-16 are learned on CIFAR-10 in both standard and adversarial setups. For the standard setup, we only present the clean accuracy. As we can see, our method maintains higher clean and robust accuracies for almost all training epochs

Table 1: Testing accuracy (%) of BSR-Net-based models. Sparse VGG-16 are learned in standard setups. For the same number of training epochs, our method has higher accuracy compared to BSR-Net in almost all cases.

| | EPOCH | 90% SPARSITY | | 99% SPARSITY | |
|---|---|---|---|---|---|
| | | BSR-NET | OURS | BSR-NET | OURS |
| AT | 20 | 55.0 (1.59) | **63.6 (1.31)** | 49.8 (1.46) | **56.4 (1.39)** |
| | 40 | 62.2 (1.88) | **64.9 (0.81)** | 54.1 (1.72) | **57.7 (0.39)** |
| | 70 | 73.1 (0.39) | **75.1 (0.27)** | 64.7 (0.30) | **66.0 (0.23)** |
| | 90 | 73.2 (0.29) | **74.1 (0.25)** | 63.7 (0.25) | **65.8 (0.24)** |
| | 140 | 76.7 (0.27) | **77.4 (0.26)** | 68.4 (0.20) | **69.8 (0.14)** |
| | 200 | 76.6 (0.25) | **78.1 (0.24)** | 69.0 (0.15) | **70.7 (0.06)** |
| TRADES | 20 | 62.0 (0.82) | **65.0 (0.61)** | 55.7 (0.76) | **57.6 (0.45)** |
| | 40 | 65.4 (0.97) | **66.0 (0.34)** | **60.6 (0.69)** | 58.4 (0.34) |
| | 70 | 73.4 (0.52) | **73.5 (0.33)** | 66.3 (0.35) | **67.3 (0.30)** |
| | 90 | 73.0 (0.36) | **73.6 (0.28)** | 66.2 (0.33) | **67.5 (0.24)** |
| | 140 | 76.4 (0.25) | **76.8 (0.25)** | **70.0 (0.29)** | 69.9 (0.21) |
| | 200 | 75.6 (0.23) | **77.0 (0.24)** | 70.8 (0.19) | **70.9 (0.25)** |
| STANDARD | 20 | 70.4 (2.50) | **81.8 (0.62)** | 60.6 (1.26) | **69.8 (1.45)** |
| | 40 | 77.6 (1.39) | **82.4 (0.47)** | 62.6 (2.47) | **73.7 (0.36)** |
| | 70 | 86.8 (0.78) | **89.7 (0.38)** | 79.7 (0.72) | **83.7 (0.24)** |
| | 90 | 87.6 (0.63) | **89.3 (0.22)** | 80.5 (0.55) | **83.9 (0.42)** |
| | 140 | 91.7 (0.44) | **92.5 (0.06)** | 85.7 (0.42) | **86.9 (0.07)** |
| | 200 | 91.8 (0.23) | **92.6 (0.12)** | 85.8 (0.12) | **87.1 (0.25)** |

and setups which demonstrates the successful acceleration from our method. In particular, for limited time periods like 20 epochs, our A-BSR-Net usually shows dramatic improvements with clean accuracy as high as

Table 2: Testing accuracy (%) of BSR-Net-based models (BSR). Sparse VGG-16 are learned in adversarial setups. For the same number of training epochs, our method has higher accuracy compared to BSR-Net in almost all cases.

| Epoch | | 90% Sparse | | 99% Sparse | | | 90% Sparse | | 99% Sparse | |
| --- | --- | --- | --- | --- | --- | --- | --- | --- | --- | --- |
| | | BSR | Ours | BSR | Ours | | BSR | Ours | BSR | Ours |
| 20 | | **38.2** | 37.3 | 31.0 | **31.4** | | 33.3 | **37.6** | 25.5 | **31.6** |
| 40 | | **39.2** | 37.9 | 33.9 | **34.5** | | 35.3 | **37.2** | 28.9 | **33.4** |
| 70 | AT | 37.8 | **45.2** | 34.9 | **39.4** | TRADES | 34.8 | **45.4** | 33.5 | **39.0** |
| 90 | | 33.6 | **44.8** | 35.8 | **39.8** | | 36.8 | **44.8** | 31.7 | **39.1** |
| 140 | | **46.5** | 43.8 | 40.8 | **41.2** | | 45.1 | **46.3** | 38.2 | **41.5** |
| 200 | | 43.3 | **44.6** | **42.2** | 42.0 | | **47.2** | 46.2 | 39.3 | **41.2** |

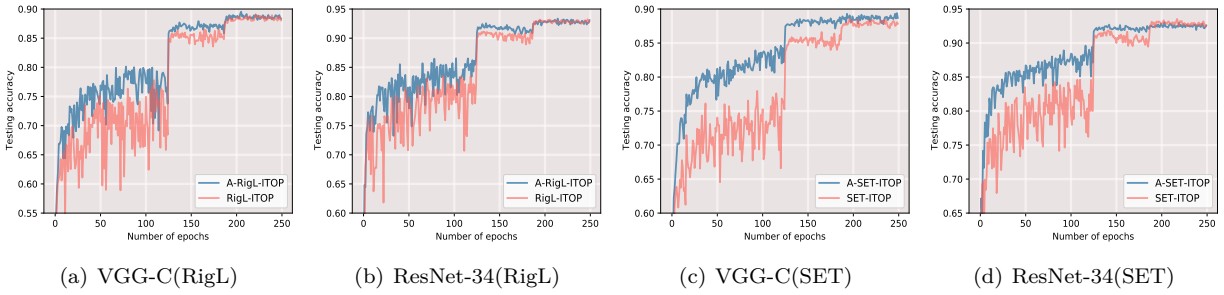

(a) VGG-C(RigL)  (b) ResNet-34(RigL)  (c) VGG-C(SET)  (d) ResNet-34(SET)

Figure 3: Testing accuracy for ITOP-based models at 99% sparsity on CIFAR-10. A-RigL-ITOP and A-SET-ITOP (blue curves) converge faster than RigL-ITOP and SET-ITOP (pink curves).

**11.4%**, indicating a significant reduction in early search time. In addition, considering the average accuracy improvement over the 6 time budgets, our method outperforms BSR-Net in accuracy by upto **5.0%**.

**For ITOP-based results using criterion test accuracy at the same number of pass data (epoch)**, as shown in Figure 3, the blue curves (A-RigL-ITOP and A-SET-ITOP) are always higher than the pink curves (RigL-ITOP and SET-ITOP), indicating faster training when using our AGENT. In addition, we can see that the pink curves experience severe up and down fluctuations, especially in the early stages of training. In contrast, the blue curves are more stable all the settings, which indicates AGENT is effective in stabilizing the sparse training.

We also show ITOP-based results on ImageNet-2012. As shown in Figure 4, the red and blue curve represent AGENT + RigL-ITOP and RigL-ITOP on 80% and 90% sparse ResNet-50, respectively. For 80% sparsity, the red curve is above the blue curve, demonstrating the acceleration effect of our AGENT, especially in the early stages. For

Table 3: Final accuracy (%) of RigL-based models at 0% (dense), 90% and 99% sparsity. AGENT + RigL (A-RigL) maintains or even improves the accuracy compared to that of RigL.

| | | Dense | 90% | 99% |
| --- | --- | --- | --- | --- |
| CIFAR-10 | A-RigL | **95.2 (0.24)** | **95.0 (0.21)** | **93.1 (0.25)** |
| | RigL | 95.0 (0.26) | 94.2 (0.22) | 92.5 (0.33) |
| CIFAR-100 | A-RigL | 72.9 (0.19) | **72.1 (0.20)** | **66.4 (0.14)** |
| | RigL | **73.1 (0.17)** | 71.6 (0.26) | 66.0 (0.19) |

90% sparity, we can see that the red curve is more stable than the blue curve, which shows the stable effect of our AGENT on large data sets. If we use SVRG in this case, we will not only fail to train stably, but also slow down the training speed. In contrast, our AGENT can solve the limitation of SVRG.

**For BSR-Net-based results using the number of pass data (epoch) required to achieve the same test accuracy**, Figure 5 depicts the number of training epochs required to achieve certain accuracy. We can see that the blue curves (A-BSR-Net) are always lower than the pink curves (BSR-Net), and on

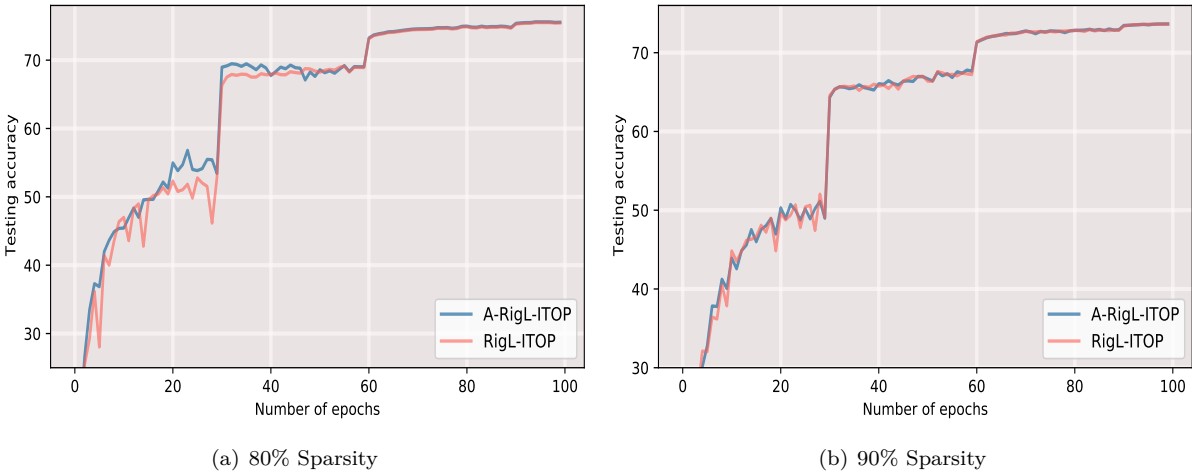

(a) 80% Sparsity

(b) 90% Sparsity

Figure 4: Testing accuracy for ITOP-based models at 80% and 90% sparsity on ImageNet-2012. A-RigL-ITOP (blue curves) converge faster than RigL-ITOP(pink curves).

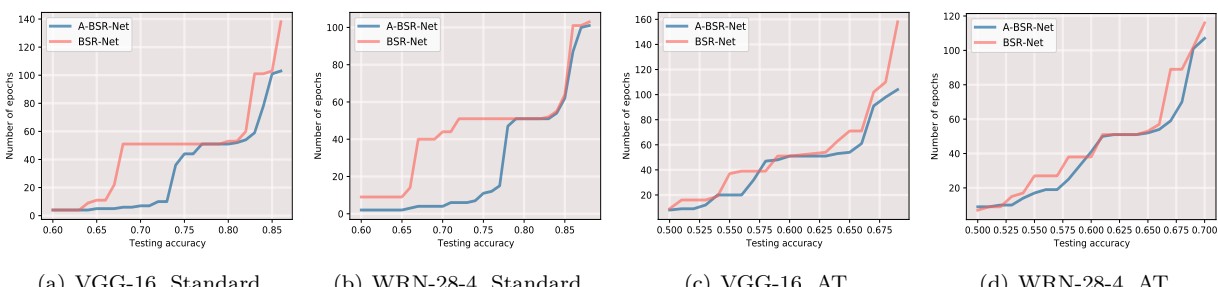

(a) VGG-16, Standard     (b) WRN-28-4, Standard     (c) VGG-16, AT     (d) WRN-28-4, AT

Figure 5: Number of training epochs required to achieve the accuracy at 99% sparsity. Our A-BSR-Net (blue curves) need less time to achieve the accuracy compared to BSR-Net (pink curves).

average our method reduces the number of training epochs by up to **52.1%**, indicating faster training when using our proposed A-BSR-Net.

## 6.2 Final Accuracy Comparisons

In addition, we compare the final accuracy after sufficient training. RigL-based results on CIFAR-10/100 are shown in Table 3. Our method A-RigL tends to be the best in almost all the scenarios. For ITOP-based results in Table 4, we compare our A-RigL-ITOP with RigL-ITOP on ImageNet-12 using ResNet-50, and our method always maintain the final accuracy. For BSR-Net-based results in Table 5, we compare our A-BSR-Net with BSR-Net on SVHN using VGG-16 and WideResNet-28-4 (WRN-28-4), and our method is often the best again. This shows that our AGENT can accelerate sparse training while maintaining or even improving the accuracy.

Table 4: Final accuracy (%) of ITOP-based ResNet-50 at 80% and 90% sparsity on ImageNet-2012. AGENT + RigL-ITOP (A-RigL-ITOP) maintains the accuracy compared to that of RigL-ITOP.

| Sparsity | RigL-ITOP | A-RigL-ITOP |
|----------|-----------|-------------|
| 80% | 75.5 (0.10) | **75.6 (0.12)** |
| 90% | 73.6 (0.12) | **73.4 (0.11)** |

## 6.3 Comparison with Other Gradient Correction Methods

We also compare our AGENT with SVRG (Baker et al., 2019), a popular gradient correction method in the non-sparse case. The presented ITOP-based results are based on sparse (99%) VGG-C and ResNet-34 on CIFAR-10. Figure 6 (a)-(b) show the testing accuracy of A-RigL-ITOP (blue), RigL-ITOP (pink), and RigL-ITOP+SVRG (green) at different epochs. We can see that the green curve for RigL-ITOP+SVRG is

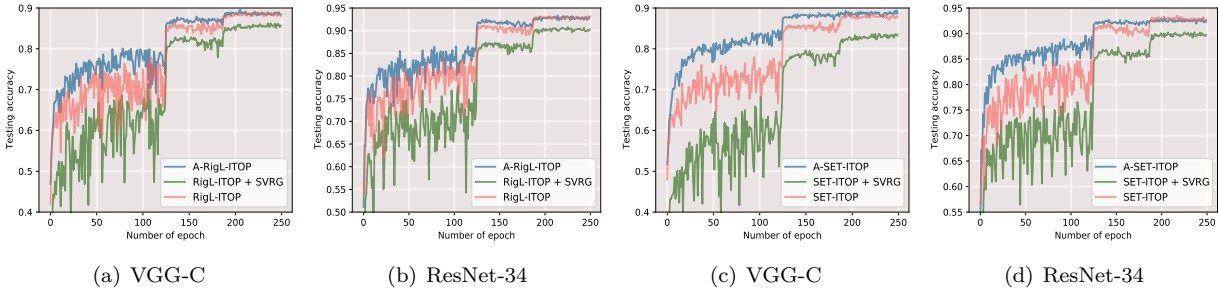

(a) VGG-C       (b) ResNet-34       (c) VGG-C       (d) ResNet-34

Figure 6: Testing accuracy for ITOP-based models at 99% sparsity on CIFAR-10. SVRG (green curves) slows down the training compared to SGD (pink curves). Our AGENT (blue curves) accelerates the training.

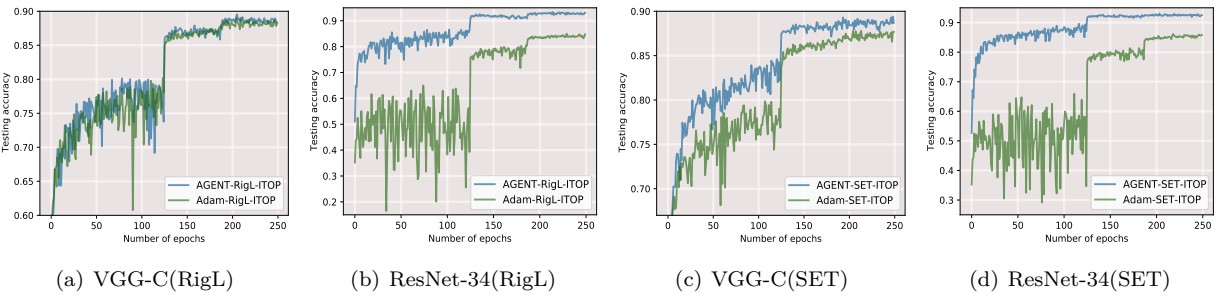

(a) VGG-C(RigL)      (b) ResNet-34(RigL)      (c) VGG-C(SET)      (d) ResNet-34(SET)

Figure 7: Testing accuracy for ITOP-based models at 99% sparsity on CIFAR-10. AGENT-based training (blue curves) converge faster than Adam-based training (pink curves).

often lower than the other two curves for A-RigL-ITOP and RigL-ITOP, indicating that model convergence is slowed down by SVRG. As for the blue curve for our A-RigL-ITOP, it is always on the top of the pink curve for RigL-ITOP and also smoother than the green curve for RigL-ITOP+SVRG, indicating a successful acceleration and stabilization. The SET-ITOP-based results depicted in Figure 6 (c)-(d) show a similar pattern. The green curve (SET-ITOP+SVRG) is often lower than the blue (A-SET-ITOP) and pink (SET-ITOP) curves. This demonstrates that SVRG does not work for sparse training, while our AGENT overcomes its limitations, leading to accelerated and stabilized sparse training.

### 6.4 Comparison with Other Adaptive Gradient Methods

We also compare our AGENT with other adaptive gradient methods, where we take Adam (Kingma & Ba, 2014) as an example. As shown in Figure 7, AGENT-RigL-ITOP and AGENT-SET-ITOP (blue curves) are usually above Adam-RigL-ITOP and Adam-SET-ITOP (green curves), indicating that our AGENT converges faster compared to Adam. This demonstrates the importance of using correlation in sparse training to balance old and new information.

### 6.5 Combination with Other Gradient Correction Methods

Table 5: Final accuracy (%) of BSR-Net-based models at 90% and 99% sparsity on SVHN with adversarial training objectives (TRADES). Our AGENT maintains or even improves the accuracy.

|     |         | BSR-NET | OURS |
| --- | ------- | ------- | ---- |
| 90% | VGG-16  | 89.4 (0.29) | **94.4 (0.25)** |
|     | WRN-28-4 | 92.8 (0.24) | **95.5 (0.23)** |
| 99% | VGG-16  | 86.4 (0.25) | **90.9 (0.26)** |
|     | WRN-28-4 | 89.5 (0.22) | **92.2 (0.19)** |

In addition to working with SVRG, our AGENT can be combined with other gradient correction methods to achieve sparse training acceleration, such as the momentum-based variance reduction method (MVR) (Cutkosky & Orabona, 2019). We train CIFAR-10 on 99% SET-ITOP-based sparse VGG-C using MVR and MVR+AGENT, respectively. As shown in Table 6, MVR+AGENT usually achieves higher

test accuracy than MVR for different number of training epochs (20, 40, 70, 90, 140, and 200), which demonstrates the acceleration effect of AGENT and the generality of our AGENT.

## 6.6 Ablation Studies

We demonstrate the importance of each component in our method AGENT by removing them one by one and comparing the results. Specifically, we consider examining the contribution of the time-varying weight $c_t$ of the old gradients and the scaling parameter $\gamma$. The term "Fixed

Table 6: Testing accuracy (%) comparisons between MVR and AGENT+MVR. AGENT can accelerate MVR in sparse training.

|  | 20-TH | 40-TH | 70-TH | 90-TH | 140-TH | 200-TH |
|---|---|---|---|---|---|---|
| MVR | 62.6 | 66.8 | 69.8 | 71.2 | 73.5 | 74.4 |
| AGENT+MVR | **71.6** | **75.7** | **77.9** | **79.1** | **82.3** | **82.3** |

$c_t$" corresponds to fixing weight $c_t = 0.1$ during training, and "No $\gamma$" represents a direct use of $\hat{c}_t^*$ in Eq. (4) and the momentum scheme without adding the scaling parameter $\gamma$.

Table 7 shows the clean and robust accuracies of standard and adversarial (AT or TRADES) training at 90% and 99% sparsity on CIFAR-10 using VGG-16 under different number of training epoch budgets. In the adversarial training (AT and TRADES), we can see that "No $\gamma$" is poorly learned and has the worst results. While our method outperforms "Fix $c_t$" and "No $\gamma$" in almost all cases, especially in highly sparse tasks (i.e., 99% sparsity). For standard training, "No $\gamma$" can learn some information, but still performs worse than the other two methods. For "Fix $c_t$", it provides similar convergence speed as our method, while ours tends to have a better final score.

From the above discussion, both the adaptive update of $c_t$ and the multiplication of the scaling parameter $\gamma$ are important for the acceleration. On the one hand, the traditional way of setting $c_t = 1$ is not desirable in sparse training and can cause model divergence under sparsity constraints. Fixing it as a smaller value, such as 0.1, sometimes can work in standard training. But updating $c_t$ adaptively with loss-dependent information usually provides some benefits, such as a better final score. These benefits become more significant in sparse and adversarial training which are more challenging and of great value. On the other hand, we recommend adding a scaling parameter $\gamma$ (e.g., $\gamma = 0.1$) to $c_t$ to avoid increasing the variance and reduce the potential bias in adversarial training, which helps the balance and further accelerates the convergence.

## 6.7 Scaling Parameter Setting

The scaling parameter $\gamma$ is to avoid introducing large variance due to error in approximating $c_t^*$ and bias due to the adversarial training. The choice of $\gamma$ is important and can be seen as a hyper-parameter tuning process. Our results are based on $\gamma = 0.1$ and the best value for $\gamma$ depends on many factors such as the dataset, architecture, and sparsity. Therefore, if we tune the value of $\gamma$ according to the gradient correlation of different settings, it is possible to obtain a faster convergence rate than the reported results.

We check different values from 0 to 1 and find that it is generally better not to set $\gamma$ to close to 1 or 0. If setting $\gamma$ close to 1, we will not be able to completely avoid the increase in variance, which leads to performance drop, similar to "No $\gamma$" in Table 7. If $\gamma$ is set too small, such as 0.01, the weight of the old gradients will be too small and the old gradients will have limited influence on the model update, which will return to SGD's slowdown and training instability. More detailed experimental results using different scaling parameters $\gamma$ are included in the Appendix B.3.

## 7 Discussion and Conclusion

We develop an adaptive gradient correction (AGENT) method for sparse training to achieve the time efficiency and reduce training instability from an optimization perspective, which can be incorporated into any SGD-based sparse training pipeline and work in both standard and adversarial setups. To achieve a fine-grained control over the balance of current and previous gradients, we use loss information to analyze gradient changes, and add an adaptive weight to the old gradients. In addition, we design a scaling parameter to reduce the bias of the gradient estimator introduced by the adversarial samples and improve the worst case

Table 7: Ablation Studies: testing accuracy (%) comparisons with Fixed $c$ and No $\gamma$ on sparse VGG-16. Results are presented as clean/robust accuracy (%). For the same number of training epochs, our method has higher accuracy compared to Fixed $c$ and No $\gamma$ in almost all cases.

| | | 90% SPARSITY | | | 99% SPARSITY | | |
| | | FIXED $c_t$ | No $\gamma$ | OURS | FIXED $c_t$ | No $\gamma$ | OURS |
|---|---|---|---|---|---|---|---|
| AT | 20-TH | 54.1/36.2 | 28.6/20.1 | **63.6/37.3** | 10.0/10.0 | 10.0/10.0 | **56.4/31.4** |
| | 40-TH | 58.9/37.1 | 20.4/13.0 | **64.9/37.9** | 10.0/10.0 | 10.0/10.0 | **57.7/34.5** |
| | 70-TH | 66.8/41.6 | 19.9/14.7 | **75.1/45.2** | 10.0/10.0 | 10.0/10.0 | **66.0/39.4** |
| | 90-TH | 67.7/43.3 | 21.8/15.6 | **74.1/44.8** | 10.0/10.0 | 10.0/10.0 | **65.8/39.8** |
| | 140-TH | 71.4/43.4 | 20.0/12.1 | **77.4/43.8** | 10.0/10.0 | 10.0/10.0 | **69.8/41.2** |
| | 200-TH | 71.7/43.0 | 20.5/9.5 | **78.1/44.6** | 10.0/10.0 | 10.0/10.0 | **70.7/42.0** |
| TRADES | 20-TH | 62.6/35.2 | 38.5/21.8 | **65.0/37.6** | 54.5/31.2 | 35.2/21.6 | **57.6/31.6** |
| | 40-TH | 65.0/**38.0** | 34.7/20.2 | **66.0**/37.2 | 56.0/30.5 | 21.5/10.0 | **58.4/33.4** |
| | 70-TH | **73.9**/44.5 | 28.8/18.4 | 73.5/**45.4** | 62.5/36.8 | 18.8/16.2 | **67.3/39.0** |
| | 90-TH | **75.1**/44.4 | 25.8/15.9 | 73.6/**44.8** | 63.9/37.4 | 16.9/15.9 | **67.5/39.1** |
| | 140-TH | 76.7/**46.5** | 28.6/14.1 | **76.8**/46.3 | 65.5/39.0 | 19.7/14.4 | **69.9/41.5** |
| | 200-TH | 76.8/46.1 | 30.7/12.7 | **77.0**/46.2 | 70.3/38.5 | 20.1/13.2 | **70.9/41.2** |
| STANDARD | 20-TH | 80.9/0.0 | 70.6/0.0 | **81.8**/0.0 | **73.7**/0.0 | 51.8/0.0 | 69.8/0.0 |
| | 40-TH | **83.3**/0.0 | 68.0/0.0 | 82.4/0.0 | **74.9**/0.0 | 55.2/0.0 | 73.7/0.0 |
| | 70-TH | **90.2**/0.0 | 77.3/0.0 | 89.7/0.0 | **84.1**/0.0 | 65.9/0.0 | 83.7/0.0 |
| | 90-TH | **89.8**/0.0 | 77.8/0.0 | 89.3/0.0 | 80.5/0.0 | 67.8/0.0 | **83.9**/0.0 |
| | 140-TH | 92.4/0.0 | 80.7/0.0 | **92.5**/0.0 | **87.2**/0.0 | 71.9/0.0 | 86.9/0.0 |
| | 200-TH | 92.1/0.0 | 78.6/0.0 | **92.6**/0.0 | 86.4/0.0 | 70.0/0.0 | **87.1**/0.0 |

of the adaptive weight estimate. In theory, we show that our AGENT can accelerate the convergence rate of sparse training. Experiment results on multiple datasets, model architectures, and sparsities demonstrate that our method outperforms state-of-the-art sparse training methods in terms of accuracy by up to **5.0%** and reduces the number of training epochs by up to **52.1%** for the same accuracy achieved.

A number of methods can be employed to reduce the FLOPs in our AGENT. Similar to SVRG, our AGENT increases the training FLOPs in each iteration due to the extra forward and backward used to compute the old gradients. To reduce the FLOPs, the first method is to use sparse gradients (Elibol et al., 2020), which effectively reduces the cost of backward in sparse training and can be easily applied to our method. The second method is parallel computing Allen-Zhu & Hazan (2016). Since the additional forward and backward over the old model parameters are fully parallelizable, we can view it as doubling the mini-batch size. Third, we can follow the idea of SAGA (Defazio et al., 2014) by storing gradients for each single sample. By this way, we do not need extra forward and backward steps, saving the computation. However, it requires extra memory to store the gradients.

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

# A  Appendix: Theoretical Proof of Convergence Rate

In this section, we provide a detailed proof for the convergence rate of our AGENT method. We start with some assumptions on which we will give some useful lemmas. Then, we will establish the convergence rate of our AGENT method based on these lemmas.

## A.1  Algorithm Reformulation

We reformulate our Adaptive Gradient Correction (AGENT) into a math-friendly version that is shown in Algorithm 2.

---

**Algorithm 2** Adaptive Gradient Correction

**Input:** Initialize $\boldsymbol{\theta}_0^0$ and $c_{-1} = 0$, set the number of epochs $S$, epoch length $m$, step sizes $\eta_t$, scaling parameter $\gamma$, and smoothing factor $\alpha$

**for** $s = 0$ **to** $S - 1$ **do**

    $\widetilde{\boldsymbol{\theta}} = \boldsymbol{\theta}_0^s$

    $\widetilde{g} = (\sum_{i=1}^N \nabla l(\mathbf{x}_i; \widetilde{\boldsymbol{\theta}}))/N$

    Calculate $\widehat{c}_s^*$ via Eq. (4)

    $\widetilde{c}_s = (1 - \alpha)\widetilde{c}_{s-1} + \alpha\widehat{c}_s^*$

    $c_s = \gamma\widetilde{c}_s$

    **for** $t = 0$ **to** $m - 1$ **do**

        Sample a mini-batch data $\mathbf{B}_t$ with size $n$

        $\boldsymbol{\theta}_{t+1}^s = \boldsymbol{\theta}_t^s - \eta_t \left( \frac{1}{n} \sum_{i \in \mathbf{B}_t} \left( g_i(\boldsymbol{\theta}_t^s) - c_s \cdot g_i(\widetilde{\boldsymbol{\theta}}) \right) + c_s \cdot \widetilde{g} \right)$

    **end for**

    $\boldsymbol{\theta}_0^{s+1} = \boldsymbol{\theta}_m^s$

**end for**

**Output:** Iterates $\boldsymbol{\theta}_\pi$ chosen uniformly random from $\{\{\boldsymbol{\theta}_t^s\}_{t=0}^{m-1}\}_{s=0}^{S-1}$

---

## A.2  Assumptions

**L-smooth**: A differentiable function $l : \mathbb{R}^n \to \mathbb{R}$ is said to be L-smooth if for all $\boldsymbol{x}, \boldsymbol{y} \in \mathbb{R}^n$ is satisfies $||\nabla l(\boldsymbol{x}) - \nabla l(\boldsymbol{y})|| \leq L||\boldsymbol{x} - \boldsymbol{y}||$. And an equivalent definition is for. all $\boldsymbol{x}, \boldsymbol{y} \in \mathbb{R}^n$:

$$-\frac{L}{2}||\boldsymbol{x} - \boldsymbol{y}||^2 \leq l(x) - l(y) - \langle \nabla l(\boldsymbol{x}), \boldsymbol{x} - \boldsymbol{y} \rangle \leq \frac{L}{2}||\boldsymbol{x} - \boldsymbol{y}||^2$$

**$\boldsymbol{\sigma}$-bounded**: We say function $l$ has a $\sigma$-bounded gradient if $||\nabla l_i(\boldsymbol{x})|| \leq \sigma$ for all $i \in [N]$ and $\boldsymbol{x} \in \mathbb{R}^n$

## A.3  Analysis framework

Under the above assumptions, we are ready to analyze the convergence rate of AGENT in **Algorithm 2**. To introduce the convergence analysis more clearly, we provide a brief analytical framework for our proof.

- First, we need to show that the variance of our gradient estimator is smaller than that of minibatch SVRG under proper choice of $c_s$. Since the gradient estimator of both AGENT and minibatch SVRG are unbiased estimators in standard training, we only need to show that our bound $E[||\mathbf{u}_t||^2]$ is smaller than minibatch SVRG. (See in Lemma 1)

- Based on above fact, we next apply the Lyapunov function to prove the convergence rate of AGENT in one arbitrary epoch. (See in Lemma 3)

- Then, we extend our previous results to the entire epoch (from 0 to $S$-th epoch) and derive the convergence rate of the output $\theta_\pi$ of **Algorithm 2**. (See in Lemma 4)

- Finally, we compare the convergence rate of our AGENT with that of minibatch SVRG. Setting the parameters in Lemma 4 according to the actual situation of sparse learning, we obtain a bound that is more stringent than minibatch SVRG.

## A.4 Lemma

We first denote step length $\eta_t = N \cdot h_t$. Since we mainly focus on a single epoch, we drop the superscript $s$ and denote $\mathbf{u}_t = \frac{1}{n} \sum_{i \in \mathbf{B}_t} \left( g_i(\boldsymbol{\theta}_t) - c \cdot g_i(\widetilde{\boldsymbol{\theta}}) \right) + c \cdot \widetilde{g}$ which is the gradient estimator in our algorithm and $\boldsymbol{\tau}_t = \frac{1}{n} \sum_{i \in \mathbf{B}_t} \left( g_i(\boldsymbol{\theta}_t) - c \cdot g_i(\widetilde{\boldsymbol{\theta}}) \right)$, then lines the update procedure in **Algorithm 2** can be replaced with $\boldsymbol{\theta}_{t+1} = \boldsymbol{\theta}_t - \eta_t \cdot \mathbf{u}_t$

### A.4.1 Lemma 1

For the $\mathbf{u}_t$ defined above and function $l$ is a L-smooth, $\lambda$ - strongly convex function with $\sigma$-bounded gradient, then we have the following results:

$$\mathbb{E}\left[||\mathbf{u}_t||^2\right] \leq 2\mathbb{E}\left[||g(\boldsymbol{\theta}_t)||^2\right] + \frac{4c^2 L^2}{n}\mathbb{E}\left[||\boldsymbol{\theta}_t - \tilde{\boldsymbol{\theta}}||^2\right] + \frac{4(1-c)^2}{n}\sigma^2 \tag{6}$$

*Proof*:

$$\begin{aligned}
\mathbb{E}\left[||\mathbf{u}_t||^2\right] &= \mathbb{E}\left[||\boldsymbol{\tau}_t + c \cdot \widetilde{g}||^2\right] = \mathbb{E}\left[||\boldsymbol{\tau}_t + c \cdot \widetilde{g} - g(\boldsymbol{\theta}_t) + g(\boldsymbol{\theta}_t)||^2\right] \\
&\leq 2\mathbb{E}\left[||g(\boldsymbol{\theta}_t)||^2\right] + 2\mathbb{E}\left[||\boldsymbol{\tau}_t - \mathbb{E}(\boldsymbol{\tau}_t)||^2\right] \leq 2\mathbb{E}\left[||g(\boldsymbol{\theta}_t)||^2\right] + \frac{2}{n}\mathbb{E}\left[\boldsymbol{\tau}_t^2\right] \\
&= 2\mathbb{E}\left[||g(\boldsymbol{\theta})||^2\right] + \frac{2}{n}\mathbb{E}\left[||c(g_i(\boldsymbol{\theta}_t) - g_i(\tilde{\boldsymbol{\theta}})) + (1-c)g_i(\boldsymbol{\theta}_t)||^2\right] \\
&\leq 2\mathbb{E}\left[||g(\boldsymbol{\theta})||^2\right] + \frac{4}{n}\mathbb{E}\left[||c(g_i(\boldsymbol{\theta}_t) - g_i(\tilde{\boldsymbol{\theta}}))||^2\right] + \frac{4(1-c)^2}{n}\mathbb{E}\left[||g_i(\boldsymbol{\theta}_t)||^2\right] \\
&\leq 2\mathbb{E}\left[||g(\boldsymbol{\theta})||^2\right] + \frac{4c^2 L^2}{n}\mathbb{E}\left[||\boldsymbol{\theta}_t - \tilde{\boldsymbol{\theta}}||^2\right] + \frac{4(1-c)^2}{n}\sigma^2
\end{aligned}$$

The first and third inequality are because $||a + b||^2 \leq 2||a||^2 + 2||b||^2$ , the second inequality follows the $\mathbb{E}\left[||\boldsymbol{\tau} - E\left[\boldsymbol{\tau}\right]||^2\right] \leq \mathbb{E}\left[||\boldsymbol{\tau}||^2\right]$ and the last inequality follows the L-smoothness and $\sigma$-bounded of function $l_i$.

**Remark 5.** *Compared with the gradient estimator of minibatch SVRG, the bound of $\mathbb{E}[||\boldsymbol{u}_t||^2]$ is smaller when L is large, $\sigma$ is relative small and c is properly chosen.*

### A.4.2 Lemma 2

$$\mathbb{E}\left[l(\boldsymbol{\theta}_{t+1})\right] \leq \mathbb{E}\left[l(\boldsymbol{\theta}_t) + \eta_t||g(\boldsymbol{\theta}_t)||^2 + \frac{L\eta^2}{2}||\mathbf{u}_t||^2\right] \tag{7}$$

*Proof*:

By the L-smoothness of function $l$, we have

$$\mathbb{E}\left[l(\boldsymbol{\theta}_{t+1})\right] \leq \mathbb{E}\left[l(\boldsymbol{\theta}_t) + \langle g(\boldsymbol{\theta}_t), \boldsymbol{\theta}_{t+1} - \boldsymbol{\theta}_t\rangle + \frac{L}{2}||\boldsymbol{\theta}_{t+1} - \boldsymbol{\theta}_t||^2\right]$$

By the update procedure in algorithm 2 and unbiasedness, the right hand side can further upper bounded by

$$\mathbb{E}\left[l(\boldsymbol{\theta}_t) + \eta_t||g(\boldsymbol{\theta}_t)||^2 + \frac{L\eta_t^2}{2}||\mathbf{u}_t||^2\right]$$

### A.4.3 Lemma 3

For $b_t, b_{t+1}, \zeta_t > 0$ and $b_t$ and $b_{t+1}$ have the following relationship

$$b_t = b_{t+1}(1 + \eta_t\zeta_t + \frac{4c^2\eta_t^2L^2}{n}) + 2\frac{c^2\eta_t^2L^3}{n}$$

and define

$$\Phi_t := \eta_t - \frac{b_{t+1}\eta_t}{\zeta_t} - \eta_t^2 L - 2b_{t+1}\eta_t^2$$

$$\Psi_t := \mathbb{E}\left[l(\boldsymbol{\theta}_t) + b_t||\boldsymbol{\theta}_t - \tilde{\boldsymbol{\theta}}||^2\right] \tag{8}$$

$\eta_t$, $\zeta_t$ and $b_{t+1}$ can be chosen such that $\Phi_t > 0$. Then the $x_t$ in Algorithm 1 have the bound:

$$\mathbb{E}[||g(\boldsymbol{\theta}_t)||^2] \leq \frac{\Psi_t - \Psi_{t+1} + \frac{2(L\eta_t^2 + 2b_{t+1}\eta_t^2)(1-c)^2}{n}\sigma^2}{\Phi_t}$$

*Proof*:
We apply Lyapunov function

$$\Psi_t = \mathbb{E}\left[l(\boldsymbol{\theta}_t) + b_t||\boldsymbol{\theta}_t - \tilde{\boldsymbol{\theta}}||^2\right]$$

Then we need to bound $||\boldsymbol{\theta}_t - \tilde{\boldsymbol{\theta}}||$

$$\begin{aligned}
\mathbb{E}\left[||\boldsymbol{\theta}_{t+1} - \tilde{\boldsymbol{\theta}}||^2\right] &= \mathbb{E}\left[||\boldsymbol{\theta}_{t+1} - \boldsymbol{\theta}_t + \boldsymbol{\theta}_t - \tilde{\boldsymbol{\theta}}||^2\right] \\
&= \mathbb{E}\left[||\boldsymbol{\theta}_{t+1} - \boldsymbol{\theta}_t||^2 + ||\boldsymbol{\theta}_t - \tilde{\boldsymbol{\theta}}||^2 + 2\langle\boldsymbol{\theta}_{t+1} - \boldsymbol{\theta}_t, \boldsymbol{\theta}_t - \tilde{\boldsymbol{\theta}}\rangle\right] \\
&= \mathbb{E}\left[\eta_t^2||\mathbf{u}_t||^2 + ||\boldsymbol{\theta}_t - \tilde{\boldsymbol{\theta}}||^2\right] - 2\eta_t\mathbb{E}\left[\langle g(\boldsymbol{\theta}_t), \boldsymbol{\theta}_t - \tilde{\boldsymbol{\theta}}\rangle\right] \\
&\leq \mathbb{E}[\eta_t^2||\mathbf{u}_t^{s+1}||^2 + ||\boldsymbol{\theta}_t - \tilde{\boldsymbol{\theta}}||^2] + 2\eta_t\mathbb{E}\left[\frac{1}{2\zeta_t}||g(\boldsymbol{\theta}_t)|| + \frac{\zeta_t}{2}||\boldsymbol{\theta}_t - \tilde{\boldsymbol{\theta}}||^2\right]
\end{aligned} \tag{9}$$

The third equality due to the unbiasedness of the update and the last inequality follows Cauchy-Schwarz and Young's inequality. Plugging Equation (6), Equation (7) and Equation (9) into Equation (8), we can get the following bound:

$$\begin{aligned}
\Psi_{t+1} &\leq \mathbb{E}\left[l(\boldsymbol{\theta}_t)\right] + \left(b_{t+1}(1 + \eta_t\zeta_t + \frac{4c^2\eta_t^2L^2}{n}) + \frac{2c^2\eta_t^2L^3}{n}\right)\mathbb{E}[||\boldsymbol{\theta}_t - \tilde{\boldsymbol{\theta}}||^2] \\
&\quad - (\eta_t - \frac{b_{t+1}\eta_t}{\zeta_t} - L\eta_t^2 - 2b_{t+1}\eta_t^2)\mathbb{E}\left[||g(\boldsymbol{\theta}_t)||^2\right] + 4(\frac{L\eta_t^2}{2} + b_{t+1}\eta_t^2)\frac{(1-c)^2}{n}\sigma^2 \\
&= \Psi_t - (\eta_t - \frac{b_{t+1}\eta_t}{\zeta_t} - L\eta_t^2 - 2b_{t+1}\eta_t^2)\mathbb{E}\left[||g(\boldsymbol{\theta}_t)||^2\right] + 4(\frac{L\eta_t^2}{2} + b_{t+1}\eta_t^2)\frac{(1-c)^2}{n}\sigma^2
\end{aligned}$$

### A.4.4 Lemma 4

Now we consider the effect of epoch and use $s$ to denote the epoch number. Let $b_m^s = 0$, $\eta_t^s = \eta$, $\zeta_t^s = \zeta$ and $b_t^s = b_{t+1}^s(1 + \eta\zeta + \frac{4c_s^2\eta L^2}{n}) + 2\frac{c_s^2\eta^2 L^2}{n}$, $\Phi_t^s = \eta - \frac{b_{t+1}^s \eta}{\zeta_t} - \eta^2 L - 2b_{t+1}^s\eta^2$ Define $\phi := \min_{t,s} \Phi_t^s$. Then we can conclude that:

$$\mathbb{E}[||g(\boldsymbol{\theta_\pi})||^2] \leq \frac{l(\theta_0) - l(\theta_*)}{T\phi} + \sum_{s=0}^{S-1}\sum_{t=0}^{m-1}\frac{2(L + 2b_{t+1}^s)(1 - c_s)^2\eta^2\sigma^2}{Tn\phi}$$

*Proof*:

Under the condition of $\eta_t^s = \eta$, we apply telescoping sum on **Lemma 3**, then we will get:

$$\sum_{t=1}^{m-1}\mathbb{E}[||g(\boldsymbol{\theta}_t^s)||^2] \leq \frac{\Psi_0^s - \Psi_m^s}{\phi} + \sum_{t=0}^{m-1}\frac{2(L + 2b_{t+1}^s)(1 - c_s)^2\eta^2\sigma^2}{n\phi}$$

From previous definition, we know $\Psi_0^s = l(\tilde{\boldsymbol{\theta}}^s)$, $\Psi_m^s = l(\tilde{\boldsymbol{\theta}}^{s+1})$ and plugging into previous equation, we obtain:

$$\sum_{t=1}^{m-1}\mathbb{E}[||g(\boldsymbol{\theta}_t^s)||^2] \leq \frac{l(\tilde{\boldsymbol{\theta}}^s) - l(\tilde{\boldsymbol{\theta}}^{s+1})}{\phi} + \sum_{t=0}^{m-1}\frac{2(L + 2b_{t+1}^s)(1 - c_s)^2\eta^2\sigma^2}{n\phi}$$

Take summation over all the epochs and using the fact that $\tilde{\boldsymbol{\theta}}^0 = \theta_0$, $l(\tilde{\boldsymbol{\theta}}^S) \leq l(\boldsymbol{\theta}_*)$ we immediately obtain:

$$\frac{1}{T}\sum_{s=0}^{S-1}\sum_{t=1}^{m-1}\mathbb{E}[||g(\boldsymbol{\theta}_t^s)||^2] \leq \frac{l(\boldsymbol{\theta}_0) - l(\boldsymbol{\theta}_*)}{\phi} + \sum_{s=0}^{S-1}\sum_{t=0}^{m-1}\frac{2(L + 2b_{t+1}^s)(1 - c_s)^2\eta^2\sigma^2}{Tn\phi} \tag{10}$$

## A.5 Theorem

### A.5.1 Theorem 1

Define $\xi_s = \sum_{t=0}^{m-1}(L + 2b_{t+1}^s)$ and $\xi := \min_s \xi_s$. Let $\eta = \frac{\mu n}{LN^\alpha}$ $(0 < \mu < 1)$ *and* $(0 < \alpha \leq 1)$, $\zeta = \frac{L}{N^{\alpha/2}}$ and $m = \frac{N^{\frac{3\alpha}{2}}}{\mu n}$. Then there exists constant $\nu, \mu, \alpha, \kappa > 0$ such that $\phi \geq \frac{n\nu}{LN^\alpha}$ and $\xi \leq \kappa L$. Then $\mathbb{E}[||g(\boldsymbol{\theta_\pi})||^2]$ can be future bounded by:

$$\mathbb{E}[||g(\boldsymbol{\theta_\pi})||^2] \leq \frac{(l(\boldsymbol{\theta}_0) - l(\boldsymbol{\theta}_*))LN^\alpha}{Tn\nu} + \frac{2\kappa\mu^2\sigma^2}{N^\alpha\nu m}$$

*Proof*:

By applying summation formula of geometric progression on the relation $b_t^s = b_{t+1}^s(1 + \eta_t\zeta_t + \frac{4c_s^2\eta_t^2 L^2}{n}) + 2\frac{c_s^2\eta_t^2 L^3}{n}$, we have $b_t^s = \frac{2c_s^2\eta_t^2 L^3}{n}\frac{(1+\omega_s)^{m-t} - 1}{\omega_s}$ where:

$$\omega_s = \eta\zeta + \frac{4c_s^2\eta^2 L}{n} = \frac{\mu n}{N^{\frac{3\alpha}{2}}} + \frac{4c_s^2\mu^2 n}{N^{2\alpha}} \leq \frac{(4c_s^2 + 1)\mu n}{N^{\frac{3\alpha}{2}}}$$

This bound holds because $\mu \leq 1$ and $N \geq 1$ and thus $\frac{4c_s^2\mu^2 n}{N^{2\alpha}} = \frac{4c_s^2\mu n}{N^{\frac{3\alpha}{2}}} \times \frac{\mu}{N^{\frac{\alpha}{2}}} \leq \frac{4c_s^2\mu n}{N^{\frac{3\alpha}{2}}}$. And using this bound, we obtain:

$$b_0^s = \frac{2\eta^2 c_s^2 L^3}{n}\frac{(1+\omega_s)^m - 1}{\omega_s} = \frac{2\mu^2 n c_s^2 L}{N^{2\alpha}}\frac{(1+\omega_s)^m - 1}{\omega_s}$$

$$\leq \frac{2\mu n c_s^2 L((1+\omega_s)^m - 1)}{N^{\frac{\alpha}{2}}(4c_s + 1)} \leq \frac{2\mu n c_s^2 L((1+\frac{(4c_s^2+1)\mu n}{N^{\frac{3\alpha}{2}}})^{\frac{N^{\frac{3\alpha}{2}}}{\mu n}} - 1)}{N^{\frac{\alpha}{2}}(4c_s^2 + 1)}$$

$$\leq \frac{2\mu n c_s^2 L(e^{\frac{1}{4c_s^2+1}} - 1)}{N^{\frac{\alpha}{2}}(4c_s^2 + 1)}$$

The last inequality holds because $(1 + \frac{1}{x})^x$ is a monotone increasing function of $x$ when $x > 0$. Thus $(1 + \frac{(4c_s^2+1)\mu n}{N^{\frac{3\alpha}{2}}})^{\frac{N^{\frac{3\alpha}{2}}}{\mu n}} \leq e^{\frac{1}{4c_s^2+1}}$ in the third inequality. And we can obtain the lower bound for $\phi$

$$\phi = \min_{t,s}\Phi_t^s \geq \min_s(\eta - \frac{b_0^s\eta}{\zeta} - \eta^2 L - 2b_0^s\eta^2) \geq \frac{n\nu}{LN^\alpha}$$

The first inequality holds since $b_s^t$ is a decrease function of $t$. Meanwhile, the second inequality holds because there exist uniform constant $\nu$ such that $\nu \geq \mu(1 - \frac{b_0^s\eta}{\zeta} - L\eta - b_0^s\eta)$.

**Remark 6.** *In practice, $b_0^s \approx 0$ because both $\gamma$ and $c_s$ is both smaller than 0.1 which leads to $\mu(1 - \frac{b_0^s}{\zeta} - L\eta - b_0^s\eta) \approx \mu(1 - L\eta)$ and this value is usually much bigger than the $\nu^*$ in the bound of minibatch SVRG.*

We need to find the upper bound for $\xi$

$$\xi_s = \sum_{t=0}^{m-1}(L + 2b_{t+1}^s) = mL + 2\sum_{t=0}^{m-1} b_{t+1}^s$$

$$= mL + 2\sum_{t=0}^{m-1}\frac{2c_s^2\eta^2 L^3}{n}\frac{(1+\omega_s)^{m-t} - 1}{\omega_s}$$

$$= mL + \frac{2c_s^2\eta^2 L^3}{n\omega_s}[\frac{(1+\omega_s)^{m+1} - (1+\omega_s)}{\omega_s} - m]$$

$$\leq mL + \frac{2c_s^2\eta^2 L^3}{n}[\frac{1+\omega_s}{\omega_s^2}(e^{\frac{1}{4c_s^2+1}} - 1) - m]$$

$$\leq mL + \frac{2c_s^2 LN^\alpha}{n}(1 + \frac{\mu n}{N^{3\alpha/2}})(e^{\frac{1}{4c_s^2+1}} - 1) - \frac{2c_s^2\mu^2 nmL}{N^{2\alpha}}$$

$$= L[(1 - \frac{2c_s^2\mu^2 nL}{N^{2\alpha}})m + \frac{2c_s^2 N^\alpha}{n}(1 + \frac{\mu n}{N^{3\alpha/2}})(e^{\frac{1}{4c_s^2+1}} - 1)]$$

The reason why the first inequality holds is explained before and the second inequality holds because $\frac{1+x}{x^2}$ is a monotone decreasing function of $x$ when $x > 0$, $\omega_s = \frac{\mu n}{N^{\frac{3\alpha}{2}}} + \frac{4c_s^2\mu^2 n}{N^{2\alpha}} \leq \frac{\mu n}{N^{\frac{3\alpha}{2}}}$ and $\eta = \frac{\mu n}{LN^\alpha}$. Then $\xi = \max_s \xi_s \leq \kappa L$ where $\kappa \geq \max_s((1 - \frac{2c_s^2\mu^2 nL}{N^{2\alpha}})m + \frac{2c_s^2 N^\alpha}{n}(1 + \frac{\mu n}{N^{3\alpha/2}})(e^{\frac{1}{4c_s^2+1}} - 1))$. When $c_s \approx 0$, $(1 - \frac{2c_s^2\mu^2 nL}{N^{2\alpha}})m + \frac{2c_s^2 N^\alpha}{n}(1 + \frac{\mu n}{N^{3\alpha/2}})(e^{\frac{1}{4c_s^2+1}} - 1) \approx m$.

Now we obtain the lower bound for $\phi$ and upper bound for $\xi$, plugging them into equation (10), we will have:

$$\mathbb{E}[||g(\boldsymbol{\theta_\pi})||^2] \leq \frac{l(\boldsymbol{\theta}_0) - l(\boldsymbol{\theta}_*)}{\phi} + \sum_{s=0}^{S-1}\sum_{t=0}^{m-1} \frac{2(L + 2b_{t+1}^s)(1 - c_s)^2\eta^2\sigma^2}{Tn\phi}$$

$$\leq \frac{(l(\boldsymbol{\theta}_0) - l(\boldsymbol{\theta}_*))LN^\alpha}{Tn\nu} + \sum_{s=0}^{S-1}\sum_{t=0}^{m-1} \frac{2(L + 2b_{t+1}^s)\eta^2\sigma^2}{Tn\phi}$$

$$\leq \frac{(l(\boldsymbol{\theta}_0) - l(\boldsymbol{\theta}_*))LN^\alpha}{Tn\nu} + \sum_{s=0}^{S-1}(\frac{2\eta^2\sigma^2}{Tn\phi})\sum_{t=0}^{m-1}(L + 2b_{t+1}^s)$$

$$\leq \frac{(l(\boldsymbol{\theta}_0) - l(\boldsymbol{\theta}_*))LN^\alpha}{Tn\nu} + \frac{2\kappa\mu^2\sigma^2}{N^\alpha\nu m}$$

**Remark 7.** *In our theoretical analysis above, we consider c as a constant in each epoch, which is still consistent with our practical algorithm for the following reasons.*

(i) In our Algorithm 1, $\widehat{c}_t^*$ is actually a fixed constant within each epoch, which can be different in different epochs. Since it is too expensive to compute the exact $\widehat{c}_t^*$ in each iteration, we compute it at the beginning of each epoch and use it as an approximation in the following epoch.

(ii) As for our proof, we first show the convergence rate of one arbitrary training epoch. In this step, treating $c$ as a constant is aligned with our practical algorithm.

(iii) Then, when we extend the results of one epoch to the whole epoch, we establish an upper bound for different $c$ in each epoch. Thus, the bound can be applied when $c$ differs across epochs, which enables our theoretical analysis consistent with our practical algorithm.

### A.6 Real Case Analysis for Sparse Training

#### A.6.1 CIFAR-10/100 dataset

In our experiments, we apply both SVRG and AGENT on CIFAR-10 and CIFAR-100 dataset with $\eta = 0.1$, $\gamma = 0.1$, batch size $m = 128$ and in total 50000 training sample. Under this parameter setting, $\nu$ and $\nu^*$ in **Theorem 1** and **Remark 4** are about 0.1 and 0.06, respectively. While $\frac{2\kappa\mu^2\sigma^2}{N^\alpha\nu m}$ is around $10^{-5}$ which is negligible so we know AGENT should have a tighter bound than SVRG in this situation which matches with the experimental results show in *Figure* 6.

#### A.6.2 svhn dataset

Meanwhile, in SVHN dataset, we train our model with parameters: $\eta = 0.1$, $\gamma = 0.1$, batch size $m = 573$ and sample size $N = 73257$. $\nu$, $\nu^*$ equal 0.4 and 0.06 respectively and $\frac{2\kappa\mu^2\sigma^2}{N^\alpha\nu m}$ is around $10^{-4}$. Although the second term in **Theorem 1** is bigger. Since $\nu$ here is a lot bigger than $\nu^*$ which lead to the first term in **Theorem 1** much smaller than that of **Remark 4**. So we still obtain a more stringent bound compared with SVRG which also meets with the outcome presented in *Figure* 9.

# B    Additional Experimental Results

We summarize additional experimental results for the BSR-Net-based Özdenizci & Legenstein (2021), RigL-based Evci et al. (2020), and ITOP-based Liu et al. (2021) models.

## B.1    Accuracy Comparisons in Different Epochs

Aligned with the main manuscript, we compare the accuracy for a given number of epochs to compare both the speed of convergence and training stability. We first show BSR-Net-based results in this section. Since our approach has faster convergence and does not require a long warm-up period, the dividing points for the decay scheduler are set to the 50th and 100th epochs. In the manuscript, we also use this schedule for BSR-Net for an accurate comparison. In the Appendix, we include the results using its original schedule. BSR-Net and BSR-Net (ori) represent the results learned using our learning rate schedule and original schedule in Özdenizci & Legenstein (2021), respectively. As shown in Figures 8, 9, 10, 11, 12, 13, 14, the blue curves (A-BSR-Net) are always higher than the yellow curves and also much smoother than yellow curves (BSR-Net and BSR-Net (ori)), indicating faster and more stable training when using our proposed A-BSR-Net.

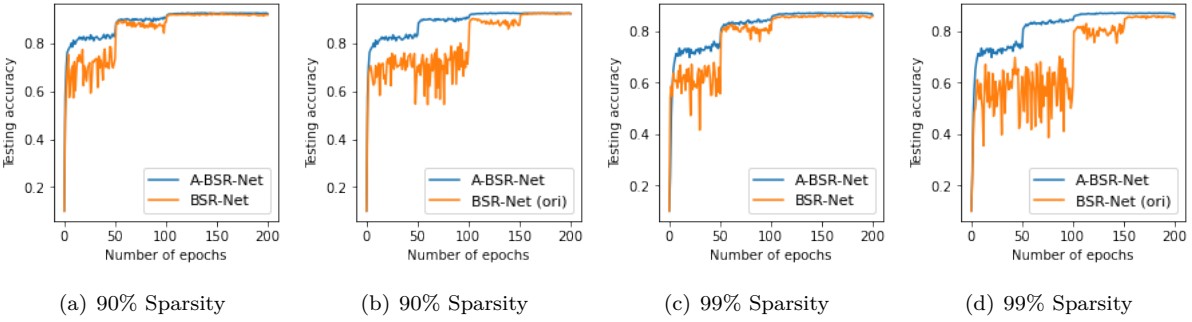

(a) 90% Sparsity     (b) 90% Sparsity     (c) 99% Sparsity     (d) 99% Sparsity

Figure 8: Comparisons (accuracy given the number of epochs) with BSR-Net Özdenizci & Legenstein (2021). We evaluate sparse networks (99% or 90%) learned with natural training on CIFAR-10 using VGG-16.

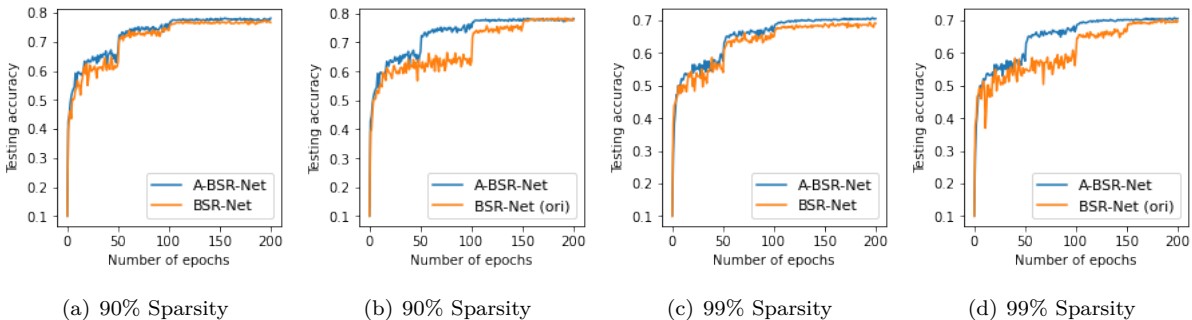

(a) 90% Sparsity     (b) 90% Sparsity     (c) 99% Sparsity     (d) 99% Sparsity

Figure 9: Comparisons (accuracy given the number of epochs) with BSR-Net Özdenizci & Legenstein (2021). We evaluate sparse networks (99% or 90%) learned with adversarial training (objective: AT) on CIFAR-10 using VGG-16.

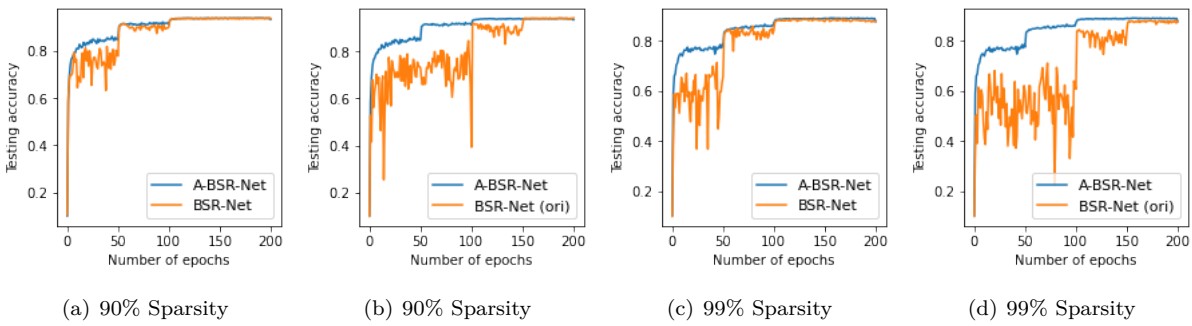

(a) 90% Sparsity      (b) 90% Sparsity      (c) 99% Sparsity      (d) 99% Sparsity

Figure 10: Comparisons (accuracy given the number of epochs) with BSR-Net Özdenizci & Legenstein (2021). We evaluate sparse networks (99% or 90%) learned with natural training on CIFAR-10 using Wide-ResNet-28-4.

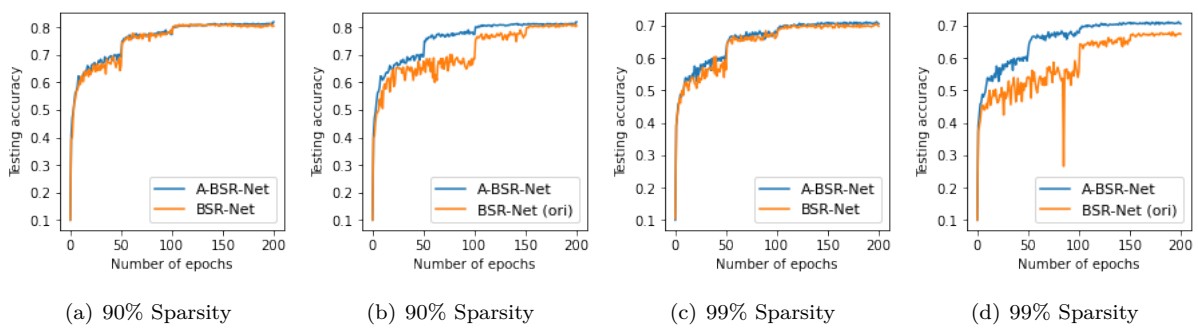

(a) 90% Sparsity      (b) 90% Sparsity      (c) 99% Sparsity      (d) 99% Sparsity

Figure 11: Comparisons (accuracy given the number of epochs) with BSR-Net Özdenizci & Legenstein (2021). We evaluate sparse networks (99% or 90%) learned with adversarial training (objective: AT) on CIFAR-10 using Wide-ResNet-28-4.

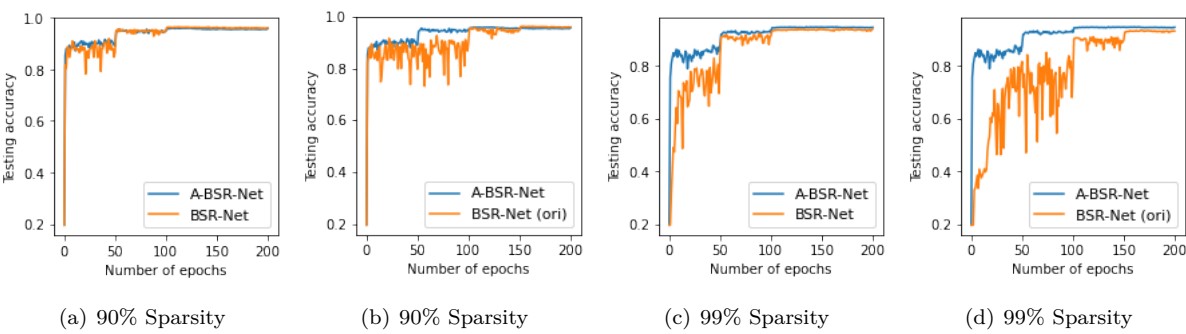

(a) 90% Sparsity      (b) 90% Sparsity      (c) 99% Sparsity      (d) 99% Sparsity

Figure 12: Comparisons (accuracy given the number of epochs) with BSR-Net Özdenizci & Legenstein (2021). We evaluate sparse networks (99% or 90%) learned with natural training on SVHN using VGG-16.

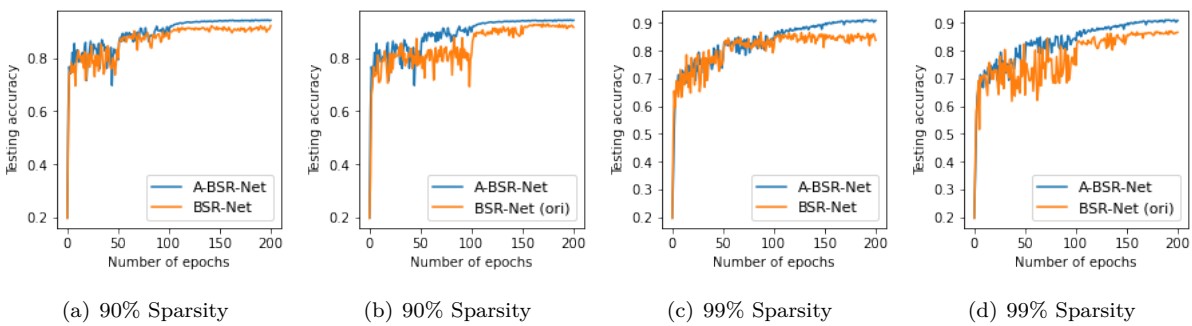

(a) 90% Sparsity     (b) 90% Sparsity     (c) 99% Sparsity     (d) 99% Sparsity

Figure 13: Comparisons (accuracy given the number of epochs) with BSR-Net Özdenizci & Legenstein (2021). We evaluate sparse networks (99% or 90%) learned with adversarial training (objective: TRADES) on SVHN using VGG-16.

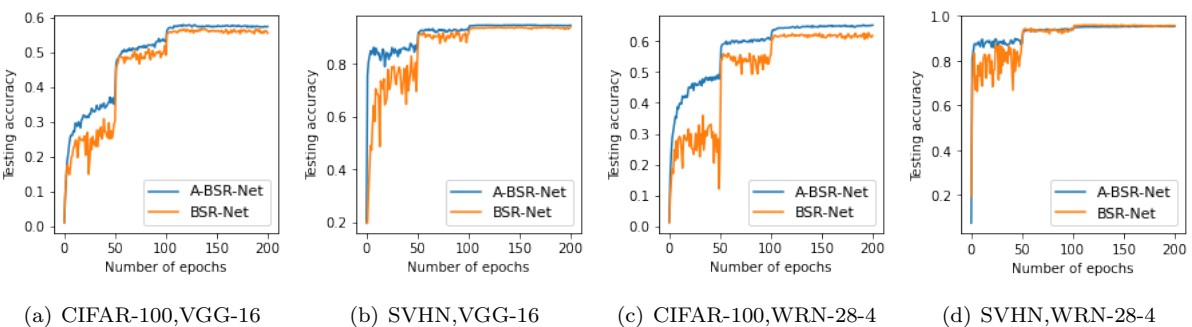

(a) CIFAR-100,VGG-16    (b) SVHN,VGG-16    (c) CIFAR-100,WRN-28-4    (d) SVHN,WRN-28-4

Figure 14: Training curve (accuracy given number of epochs) of BSR-Net-based models (Özdenizci & Legenstein, 2021). Sparse networks (99%) are learned in standard setups on (a) CIFAR-100 using VGG-16, (b) SVHN using VGG-16, (c) CIFAR-100 using WRN-28-4, (d) SVHN using WRN-28-4.

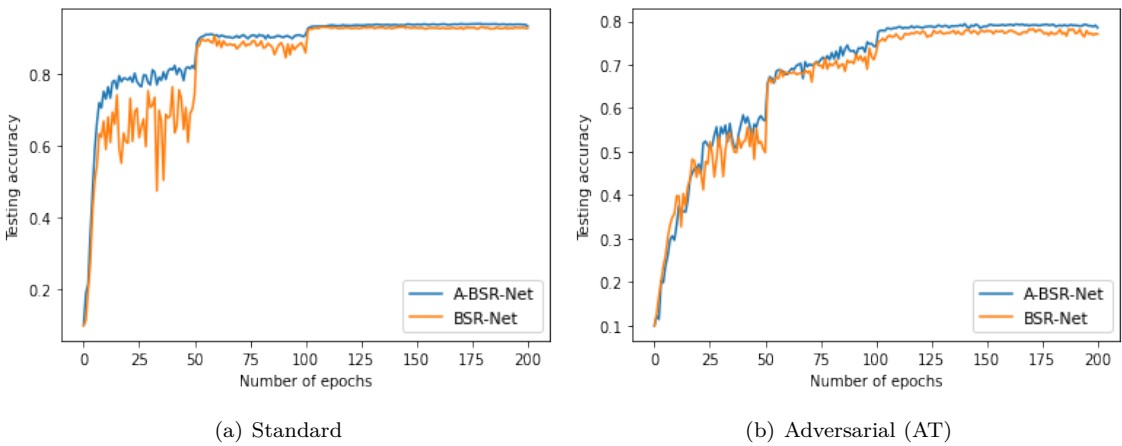

(a) Standard                 (b) Adversarial (AT)

Figure 15: Training curve (required epochs to reach given accuracy) of BSR-Net-based models (Özdenizci & Legenstein, 2021). Dense networks are learned in standard and adversarial setups on CIFAR-10 using VGG-16.

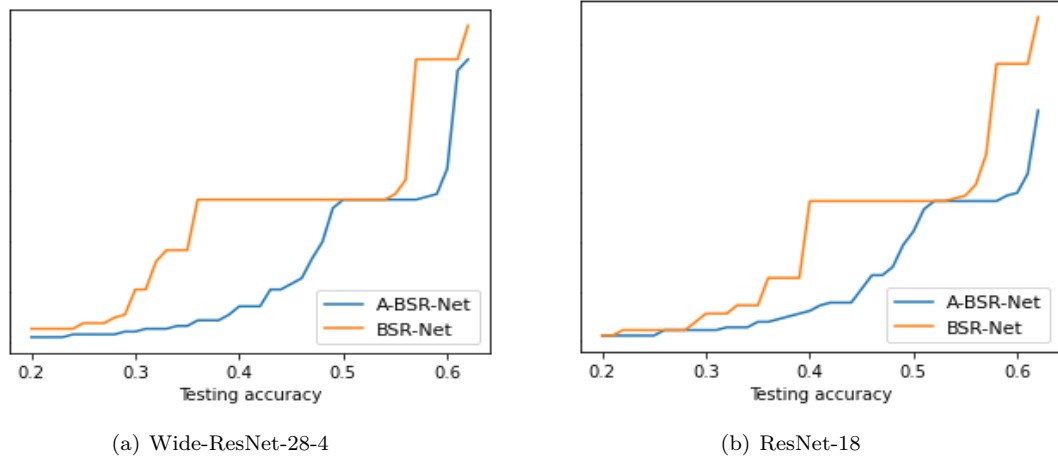

(a) Wide-ResNet-28-4                    (b) ResNet-18

Figure 16: Comparisons (required hours to reach given accuracy. We evaluate sparse networks (99%) learned with natural training on CIFAR-100 using (a) Wide-ResNet-28-4, and (b) ResNet-18.

In Figure 15, we also compare the convergence speed without sparsity. We show a BSR-Net-based result, where dense network is learned by adversarial training (AT) and standard training on CIFAR-10 using VGG-16. The blue curve of our A-BSR-Net tends to be above the yellow curve of BSR-Net, indicating successful acceleration. This demonstrates the broad applicability of our method.

## B.2   Number of Training Epoch Comparisons

We also compare the number of training epochs required to reach the same accuracy in BSR-Net-based results. In Figures 16, 17, 18, 19, 20, 21, 22, the blue curves (A-BSR-Net) are always lower than yellow curves (BSR-Net and BSR-Net (ori)), indicating faster convergence of A-BSR-Net.

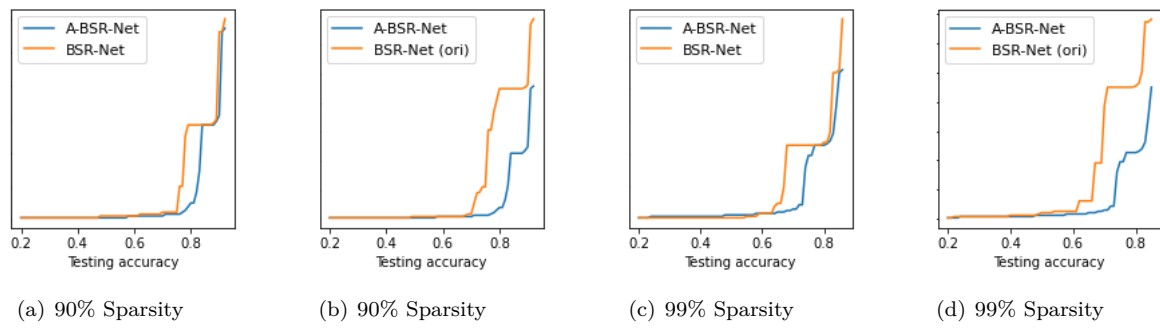

(a) 90% Sparsity          (b) 90% Sparsity          (c) 99% Sparsity          (d) 99% Sparsity

Figure 17: Comparisons (required hours to reach given accuracy. We evaluate sparse networks (99% or 90%) learned with natural training on CIFAR-10 using VGG-16.

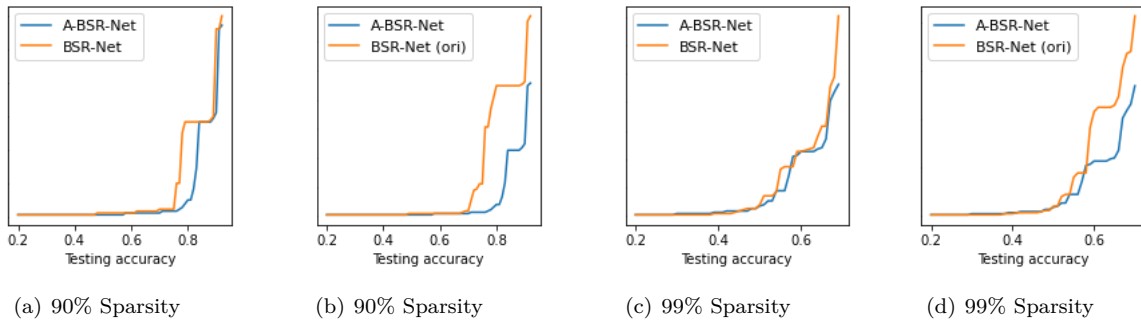

(a) 90% Sparsity     (b) 90% Sparsity     (c) 99% Sparsity     (d) 99% Sparsity

Figure 18: Comparisons (required hours to reach given accuracy). We evaluate sparse networks (99% or 90%) learned with adversarial training (objective: AT) on CIFAR-10 using VGG-16.

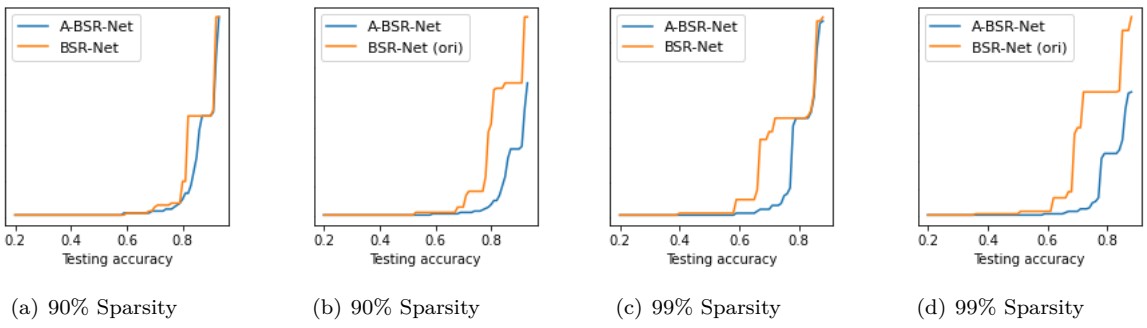

(a) 90% Sparsity     (b) 90% Sparsity     (c) 99% Sparsity     (d) 99% Sparsity

Figure 19: Comparisons (required hours to reach given accuracy). We evaluate sparse networks (99% or 90%) learned with natural training on CIFAR-10 using Wide-ResNet-28-4.

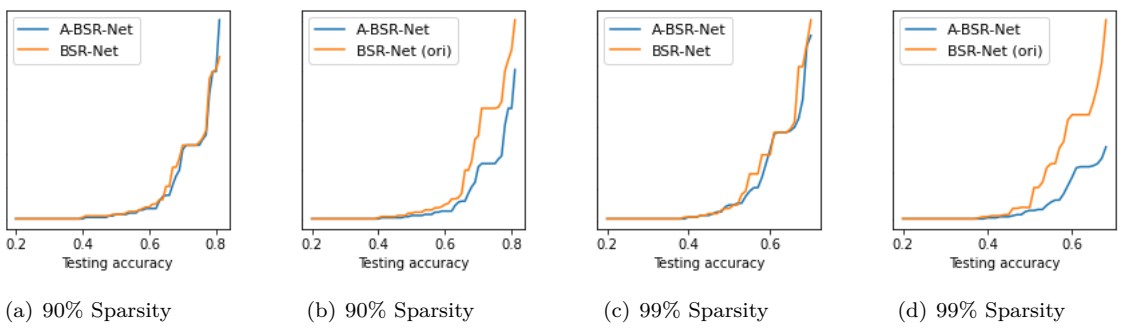

(a) 90% Sparsity     (b) 90% Sparsity     (c) 99% Sparsity     (d) 99% Sparsity

Figure 20: Comparisons (required hours to reach given accuracy). We evaluate sparse networks (99% or 90%) learned with adversarial training (objective: AT) on CIFAR-10 using Wide-ResNet-28-4.

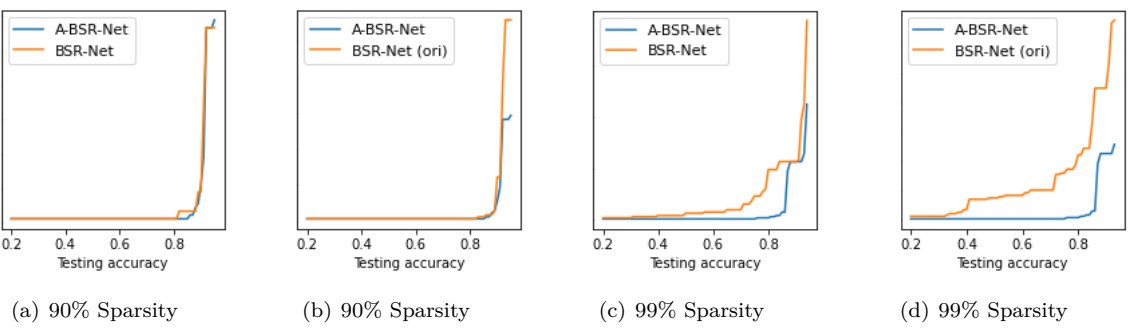

(a) 90% Sparsity     (b) 90% Sparsity     (c) 99% Sparsity     (d) 99% Sparsity

Figure 21: Comparisons (required hours to reach given accuracy). We evaluate sparse networks (99% or 90%) learned with natural training on SVHN using VGG-16.

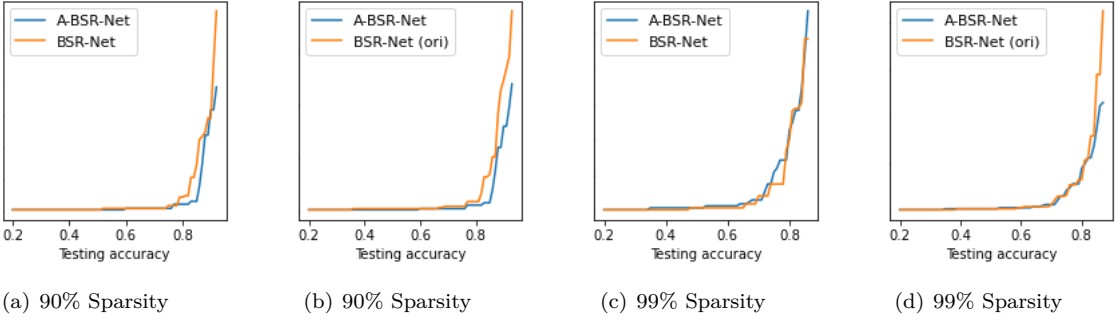

(a) 90% Sparsity        (b) 90% Sparsity        (c) 99% Sparsity        (d) 99% Sparsity

Figure 22: Comparisons (required hours to reach given accuracy). We evaluate sparse networks (99% or 90%) learned with adversarial training (objective: TRADES) on SVHN using VGG-16.

### B.3 Scaling Parameter Setting

The choice of the scaling parameter $\gamma$ is important to the acceleration and can be seen as a hyper-parameter tuning process. We experiment with different values of $\gamma$ and find that setting $\gamma = 0.1$ is a good choice for effective acceleration of training. We first present results which are based on sparse networks (99%) learned with adversarial training (objective: AT) on CIFAR-10 using VGG-16. The sparse training method is BSR-Net.

$\gamma = 0.1$ vs $\gamma = 0.5$: As shown in Figure 23 (a), we compare the training curves (testing accuracy at different epochs) A-BSR-Net ($\gamma = 0.1$), A-BSR-Net ($\gamma = 0.5$), and BSR-Net. The yellow curve for A-BSR-Net ($\gamma = 0.5$) collapses after around 40 epochs training, indicating a model divergence. The reason is that if setting $\gamma$ close to 1, e.g., like 0.5, we will not be able to completely avoid the increase in variance. The increase in variance will lead to a decrease in performance, which is similar to "No $\gamma$" in section 5.4 of the manuscript.

$\gamma = 0.1$ vs $\gamma = 0.01$: As shown in Figure 23 (b), we compare the training curves (testing accuracy at different epochs) A-BSR-Net ($\gamma = 0.1$), A-BSR-Net ($\gamma = 0.01$), and BSR-Net. The yellow curve for A-BSR-Net ($\gamma = 0.01$) is below the blue curve for A-BSR-Net ($\gamma = 0.1$), indicating a slower convergence speed. The reason is that if $\gamma$ is set small, such as 0.01, the weight of the old gradients will be small. Thus, the old gradients will have limited influence on the updated direction of the model, which tends to slow down the convergence and sometimes can lead to more training instability.

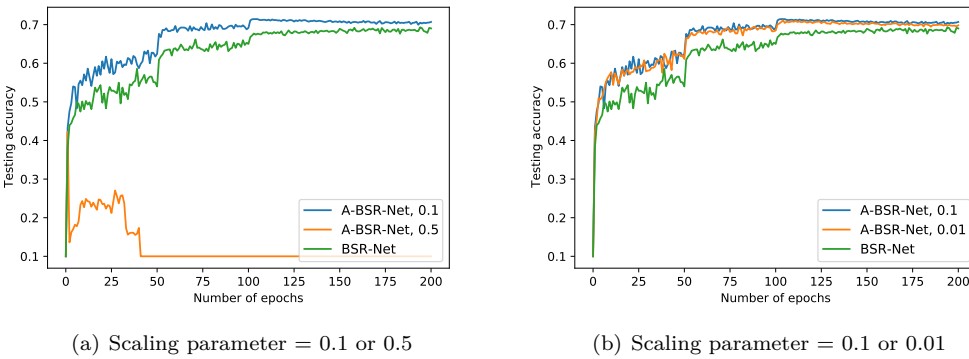

(a) Scaling parameter = 0.1 or 0.5        (b) Scaling parameter = 0.1 or 0.01

Figure 23: Comparisons (validation accuracy given the number of epochs) with different scaling parameters in BSR-Net-based models Özdenizci & Legenstein (2021). We evaluate sparse networks (99%) learned with adversarial training (objective: AT) on CIFAR-10 using VGG-16. (a) scaling parameter = 0.1 or 0.5, (b) scaling parameter = 0.1 or 0.01.

We also present results which are based on sparse networks (99%) learned with standard training on CIFAR-10 using VGG-C. The sparse training method is SET-ITOP. As shown in Figure 24, the results of setting $\gamma = 0.01, 0.5$ are similar to that of $\gamma = 0.1$, and the results of setting $\gamma = 0.9$ are worse than that of $\gamma = 0.1$. This may be due to the fact that 0.9 is too large for the relatively low gradient correlation.

### B.4 Other Variance Reduction Method Comparisons

We also include more results about comparison between our ADSVRG and stochastic variance reduced gradient (SVRG) Baker et al. (2019); Chen et al. (2019); Zou et al. (2018), a popular variance reduction method in non-sparse case, to show the limitations of previous methods.

### B.4.1 BSR-Net-based Results

The presented results are based on sparse networks (99%) learned with adversarial training (objective: AT) on CIFAR-10 using VGG-16. As presented in Figure 25, we show the training curves (testing accuracy

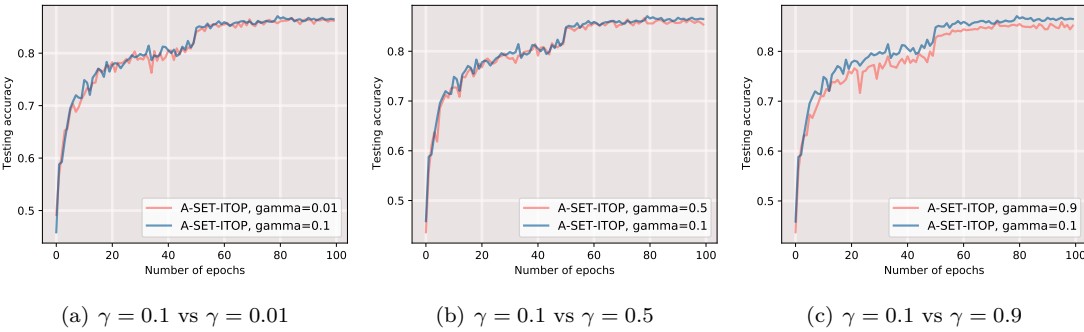

(a) $\gamma = 0.1$ vs $\gamma = 0.01$         (b) $\gamma = 0.1$ vs $\gamma = 0.5$         (c) $\gamma = 0.1$ vs $\gamma = 0.9$

Figure 24: Comparisons (validation accuracy given the number of epochs) between different scaling factors $\gamma$. We evaluate sparse networks (99%) learned with standard training on CIFAR-10 using VGG-C where we set (a) $\gamma = 0.1$ vs $\gamma = 0.01$, (b) $\gamma = 0.1$ vs $\gamma = 0.5$, and (c) $\gamma = 0.1$ vs $\gamma = 0.9$.

at different epochs)of A-BSR-Net, BSR-Net, and BSR-Net using SVRG. The yellow curve for BSR-Net using SVRG rises to around 0.4 and then rapidly decreases to a small value around 0.1, indicating a model divergence. This demonstrates that SVRG does not work for sparse training. As for the blue curve for our A-BSR-Net, it is always above the green curve for BSR-Net, indicating a successful acceleration.

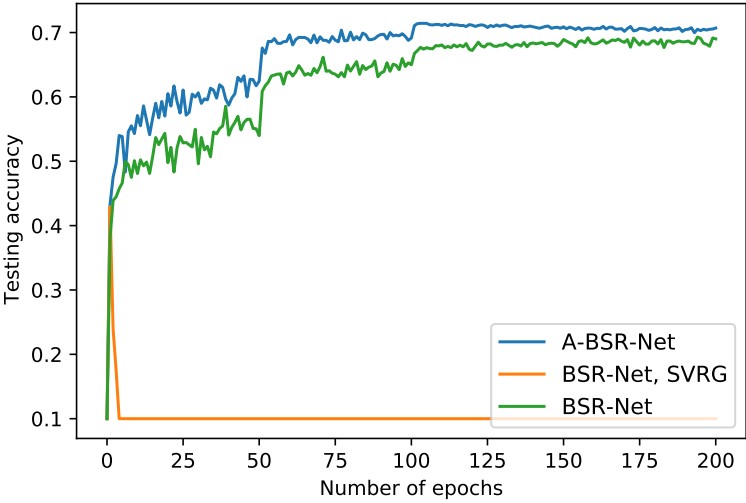

Figure 25: Comparisons (testing accuracy given the number of epochs) with different variance reduction methods in BSR-Net-based models Özdenizci & Legenstein (2021). We evaluate sparse networks (99%) learned with adversarial training (objective: AT) on CIFAR-10 using VGG-16.

### B.4.2 RigL-based Results

The presented results are based on sparse networks (90%) learned with standard training on CIFAR-100 using ResNet-50. As presented in Figure 26, we show the training curves (testing accuracy at different epochs) of A-RigL, RigL, and RigL using SVRG. The yellow curve for RigL using SVRG is always below the other two curves, indicating a slower model convergence. This demonstrate that SVRG does not work for sparse training. As for the blue curve for our A-RigL, it is always on the top of the green curve for RigL, indicating that the speedup is successful.

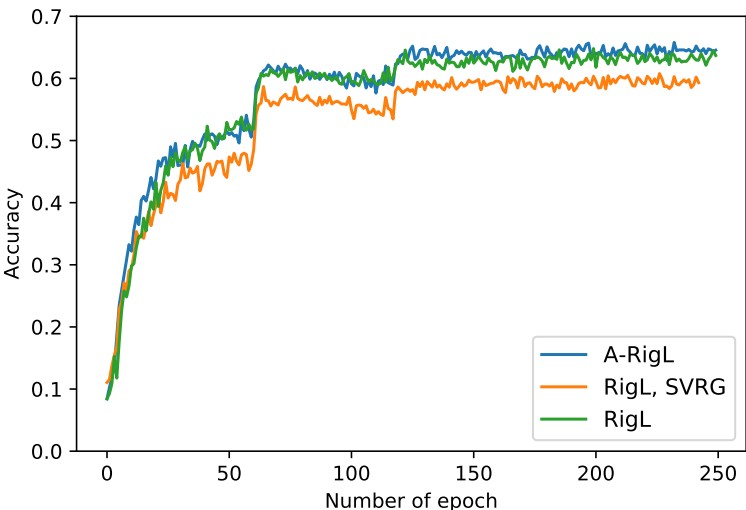

Figure 26: Comparisons (testing accuracy given the number of epochs) with different variance reduction methods in RigL-based models Evci et al. (2020). We evaluate sparse networks (90%) learned with standard training on CIFAR-100 using ResNet-50.

Table 8: Comparisons the BSR-Net Özdenizci & Legenstein (2021) and HYDRA Sehwag et al. (2020). Evaluations of sparse networks learned with robust training objectives (TRADES) on SVHN using VGG-16 and WideResNet-28-4. Evaluations are after full training (200 epochs) and presented as clean/robust accuracy (%). Robust accuracy is evaluated via $PGD^{50}$ with 10 restarts $\epsilon = 8/255$.

|  |  | BSR-Net | HYDRA | Ours |
|---|---|---|---|---|
| 90% Sparsity | VGG-16 | 89.4/**53.7** | 89.2/52.8 | **94.4**/51.9 |
|  | WRN-28-4 | 92.8/**55.6** | 94.4/43.9 | **95.5**/46.2 |
| 99% Sparsity | VGG-16 | 86.4/**48.7** | 84.4/47.8 | **90.9**/47.9 |
|  | WRN-28-4 | 89.5/**52.7** | 88.9/39.1 | **92.2**/51.1 |

## B.5 Final Accuracy Comparisons

We also provide additional BSR-Net-based results for the final accuracy comparison. In addition to the BSR-Net and A-BSR-Net in the manuscript, we also include HYDRA in the appendix, which is also a SOTA sparse and adversarial training pipeline. The results are trained on SVHN using VGG-16 and WideResNet-28-4 (WRN-28-4). The final results for BSR-Net and HYDRA are obtained from Özdenizci & Legenstein (2021) using their original learning rate schedules. As shown in Table 8, it is encouraging to note that our method tends to be the best in all cases when given clean test samples. In terms of the robustness, our A-BSR-Net beats HYDRA in most cases, while experience a performance degradation compared to BSR-Net.

## B.6 Gradient Change Speed & Sparsity Level

In sparse training, when there is a small change in the weights, the gradient changes faster than in dense training, and this phenomenon can be expressed as a low correlation between the current and previous gradients, making the existing variance reduction methods ineffective.

**Intuitive point of view**: Considering the weights on which the current and previous gradients were calculated, there are three cases to be discussed in sparse training when the masks of current and previous

gradients are different. First, if current weights are pruned, we do not need to consider their correlation because we do not need to update the current weights using the corresponding previous weights. Second, if current weights are not pruned but previous weights are pruned, the previous weights are zero and the difference between two weights is relatively large, leading to a lower relevance. Third, if neither the current nor the previous weights are pruned, which weights are pruned can still change significantly, leading to large changes in the current and previous models. Thus, the correlation between the current and previous gradients of the weights will be relatively small. Thus, it is not a good idea to set $c = 1$ directly in sparse training which can even increase the variance and slow down the convergence.

When the masks of the current and previous gradients are the same, the correlation still tends to be weaker. As we know, $c_t^* = \frac{\text{Cov}(g_{\text{new}}, g_{\text{old}})}{\text{Var}(g_{\text{old}})}$. Even if $\text{Cov}(g_{\text{new}}, g_{\text{old}})$ does not decrease, the variance $\text{Var}(g_{\text{old}})$ increases in sparse training, leading to a decrease in $c_t^*$.

Apart from the analysis above, we also do some experiments to demonstrate that the gradient changes faster as the sparsity increases. To measure the rate of change, our experiments are described below.

**Correlation over the course of training:** We analyze the gradient correlation during the standard training of sparse VGG-C on CIFAR-10 using SET-ITOP. The results are sumarized in Figure 27, where the blue curves represent the gradient correlation of dense training (0% sparsity) and the pink curves denote correlation of sparse training, i.e., SET-ITOP. As we can see, for 50% sparsity, the correlation between dense and sparse training are close. For 80% sparsity, sparse training tend to have lower correlation compared to the dense training, especially in late training stages. For 90% and 95%, sparse training also gives lower relevance than dense training, and the differences become larger with increasing sparsity.

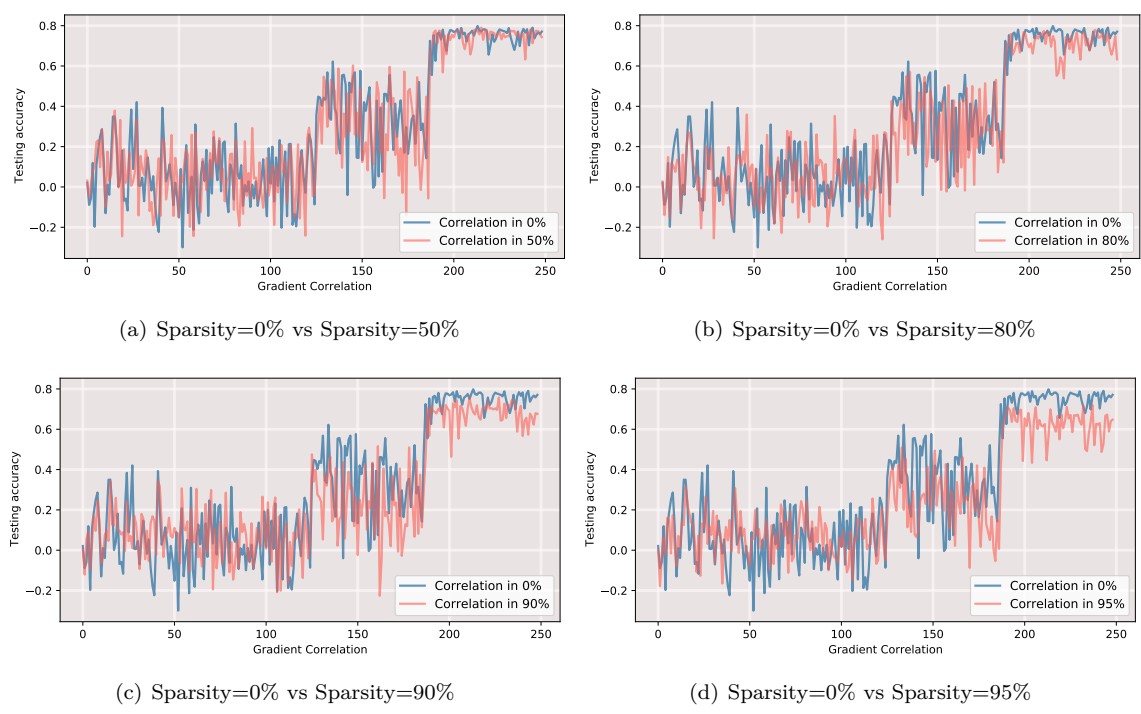

(a) Sparsity=0% vs Sparsity=50%

(b) Sparsity=0% vs Sparsity=80%

(c) Sparsity=0% vs Sparsity=90%

(d) Sparsity=0% vs Sparsity=95%

Figure 27: Gradient correlation in dense training and SET-ITOP. We evaluate sparse networks learned with standard training on CIFAR-10 using VGG-C. We compare the correlation between dense training (sparsity=0%) and sparse training in sparsity (a) 50%, (b) 80%, (c) 90%, and (d) 95%.

We also analyze the gradient correlation during the standard training of sparse ResNet-50 on CIFAR-100 using RigL. The results are sumarized in Figure 28, where the blue curves represent the gradient correlation of dense training (0% sparsity) and the pink curves denote correlation of sparse training, i.e., RigL. As we can see, the pattern is similar with those of SET-ITOP. For 50% sparsity, the correlation between dense and sparse training are close. For 80% sparsity, sparse training tend to have lower correlation compared to the

dense training, especially in late training stages. For 90% and 95%, sparse training also gives lower relevance than dense training, and the differences become larger with increasing sparsity.

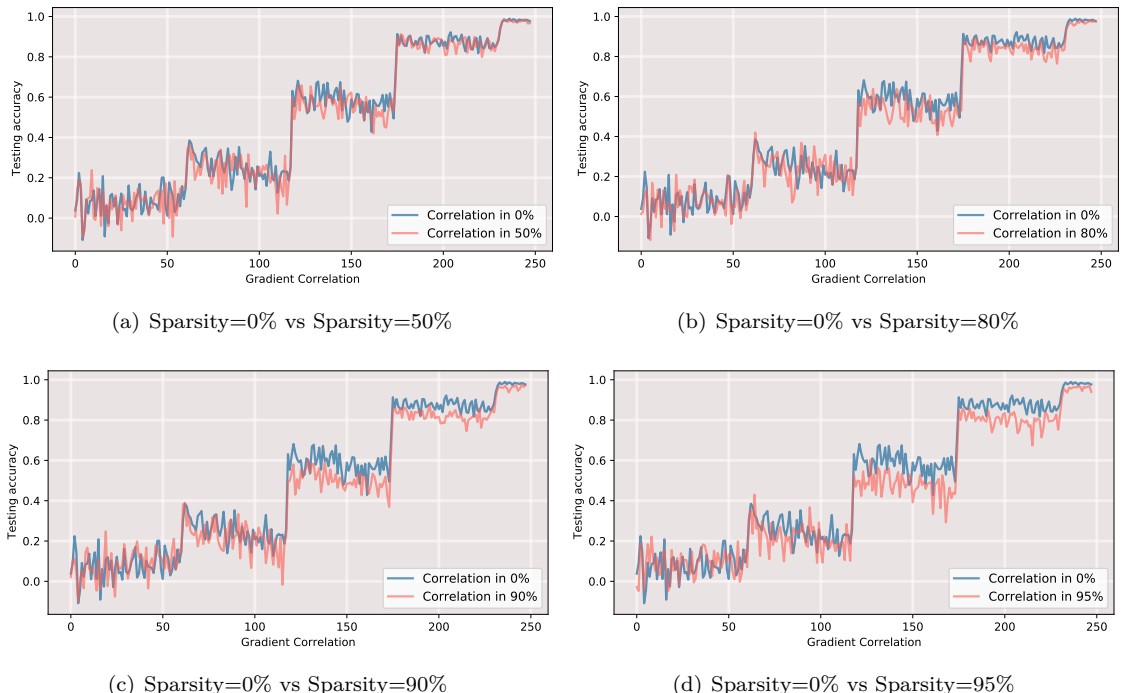

(a) Sparsity=0% vs Sparsity=50%

(b) Sparsity=0% vs Sparsity=80%

(c) Sparsity=0% vs Sparsity=90%

(d) Sparsity=0% vs Sparsity=95%

Figure 28: Gradient correlation in dense training and SET-ITOP. We evaluate sparse networks learned with standard training on CIFAR-10 using VGG-C. We compare the correlation between dense training (sparsity=0%) and sparse training in sparsity (a) 50%, (b) 80%, (c) 90%, and (d) 95%.

**Correlation of the fully-trained model:** We begin with fully-trained checkpoints from ResNet-50 on CIFAR-100 with RigL and SET at 0%, 50%, 80%, 90%, and 95% sparsity. We calculate and store the gradient of each weight on all training data. Then, we add Gaussian perturbations (std = 0.015) to all the weights and calculate the gradients again. Lastly, we calculate the correlation between the gradient of the new perturbed weights and the old original weights.

As we know, there is always a difference between the old and new weights. If the gradients become very different after adding some small noise to the weights, the new and old gradients will tend to have smaller correlations. If the gradients do not change a lot after adding some small noise, the old and new gradients will have a higher correlation. Thus, we add Gaussian noise to the weights to simulate the difference between the new and old gradients. As shown in Table 9, the correlation decreases with increasing sparsity, which indicates a weaker correlation in sparse training and supports our claim.

Table 9: Correlation between the gradient of the new perturbed weights and the old original weights from ResNet-50 on CIFAR-100 produced by RigL and SET at different sparsity including 0%, 50%, 80%, 90%, 95%, 99%.

| SPARSITY | 0% | 50% | 80% | 90% | 95% |
|---|---|---|---|---|---|
| RESNET-50, CIFAR-100 (RIGL) | 0.6005 | 0.4564 | 0.3217 | 0.1886 | 0.1590 |
| RESNET-50, CIFAR-100 (SET) | 0.6005 | 0.4535 | 0.2528 | 0.1763 | 0.1195 |

### B.7 Comparison between True Correlation & Our Approximation

In this section, to test how well our approximation estimates the true optimum c, we empirically compare the approximation $c^*$ in Eq. (4) (in the main manuscript) and the correlation between gradient of current weights and gradient of previous epoch weights. As shown in Figure 29, the yellow and blue curves represent the approximation $c^*$ and the correlation, respectively. The two curves tend to have similar up-and-down patterns, and the yellow curves usually have a larger magnitude. This suggests that our c approximation captures the dynamic patterns of the correlation. For the larger magnitude, it can be matched by our scaling parameter.

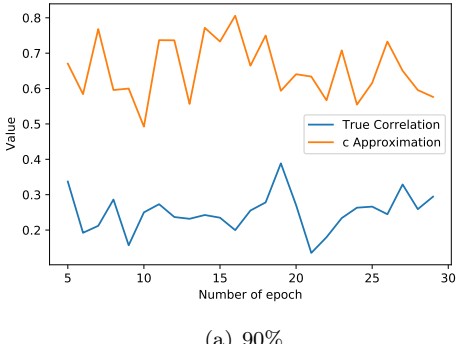
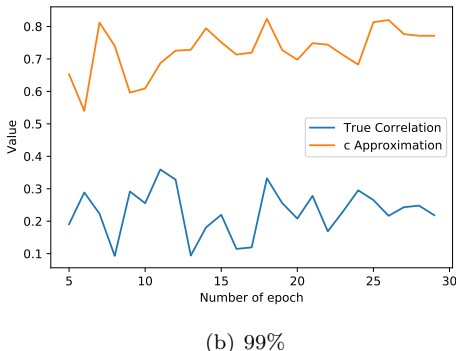

(a) 90%

(b) 99%

Figure 29: Comparisons between the approximation $c^*$ and correlation between gradient of current weights and gradient of previous epoch weights. We evaluate sparse networks learned with RigL-based standard training on CIFAR-10 using ResNet-50 with (a) 90% sparsity and (b) 99% sparsity.

### B.8 Variants of RigL

RigL is one of the most popular dynamic sparse training pipeline which uses weight magnitude for pruning and gradient magnitude for growing. Our method adaptively updates the new batch gradient using the old storage gradient which usually has less noise. As a result, the variance of the new batch gradient is reduced, leading to fast convergence. Currently, we only use gradients with corrected variance in weight updates. A natural question is how does it perform if we also use this variance-corrected gradient for weight growth in RigL.

We do some experiments in RigL-based models trained on CIFAR-10. As shown in Figure 30, the blue curves (RigL-ITOP-G) and yellow curves (RigL-ITOP) correspond to the weight growth with and without the variance-corrected gradient, respectively. We can see that in the initial stage, the blue curves are higher than the yellow curves. But after the first learning rate decay, they tend to be lower than the yellow curves. This suggests that weight growth using a variance-corrected gradient at the beginning of training can help the model improve accuracy faster. However, this may lead to a slight decrease in accuracy in the later training stages. This may be due to the fact that some variance in the gradient can help the model explore local regions better and find better masks as the model approaches its optimal point.

### B.9 Comparison with Reducing Learning Rate

To demonstrate the design of the scaling parameter $\gamma$, we compare our AGENT with "Reduce LR", where we remove the scaling parameter $\gamma$ from AGENT and set the learning rate to 0.1 times the original one. As shown in Table 10, reducing the learning rate can lead to a comparable convergence rate in the early stage. However, it slows down the later stages of training and leads to sub-optimal final accuracy. The reason is that it reduces both signal and noise, and therefore does not improve the signal-to-noise ratio or speed up the sparse training.

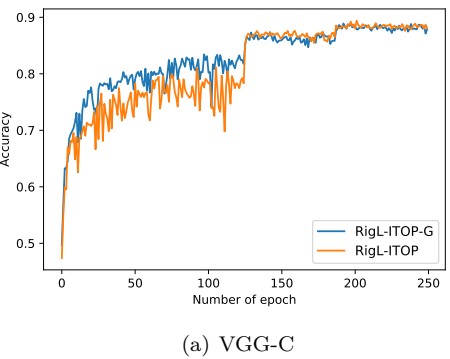
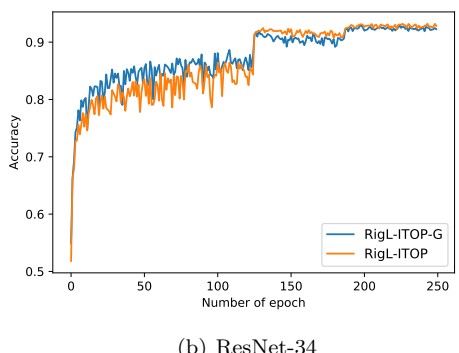

(a) VGG-C

(b) ResNet-34

Figure 30: Comparisons (testing accuracy given the number of epochs) between weight growth with (RigL-ITOP-G) and without (RigL-ITOP) variance-corrected gradient Liu et al. (2021). We evaluate sparse networks (99%) learned with standard training on CIFAR-10 using (a) VGG-C and (b) ResNet-34.

The motivation of $\gamma$ is to avoid introducing large variance due to error in approximating $c_t$ and bias due to the adversarial training. The true correlation depends on many factors such as the dataset, architecture, and sparsity. In some cases, it can be greater or smaller than 10%. For the value of $\gamma$, it is a hyperparameter and we can choose different values for different settings. In our case, for simplicity, we choose $\gamma = 0.1$ for all the settings, and find that it works well and accelerates the convergence. If we tune the value of $\gamma$ for different settings according to their corresponding correlations, it is possible to obtain faster convergence rates.

Table 10: Testing accuracy (%) of SET-ITOP-based models for AGENT (ours) and "Reduce LR". Sparse VGG-C and ResNet-34 are learned in standard setups.

| Epoch | 20 | 80 | 130 | 180 | 240 |
|---|---|---|---|---|---|
| Reduce LR (VGG-C, SET-ITOP) | **76.5** | 81.3 | 84.6 | 85.5 | 85.5 |
| AGENT (VGG-C, SET-ITOP) | 76.1 | **81.5** | **87.6** | **87.1** | **88.6** |
| Reduce LR (ResNet-34, SET-ITOP) | 81.4 | **85.9** | 89.3 | 89.5 | 89.8 |
| AGENT (ResNet-34, SET-ITOP) | **83.0** | 85.6 | **92.0** | **92.3** | **92.5** |

### B.10 Comparison with Momentum-based Methods

The momentum-based approach works well in general, but it still suffers from optimization difficulties due to sparsity constraints. For example, in our baseline SGD, following the original code base, we have also added momentum to the optimizer. However, as shown in the pink curves in Figure 2, it still has training instability and convergence problems. The reason is that they do not take into account the sparse and adversarial training characteristics and cannot provide an adaptive balance between old and new information.

Our method AGENT is designed for sparse and adversarial training and can establish a finer control over how much information we should get from the old to help the new. To demonstrate the importance of this fine-grained adaptive balance, we do ablation studies in Section 6.4. In "Fixed $c_t$", we set $c_t = 0.1$ and test the convergence rate without the adaptive control. We find that the adaptive balance (ours) outperforms "Fixed $c_t$" in almost all cases, especially in adversarial training. For standard training, "Fix $c_t$" provides similar convergence rates to our method, while ours tends to have better final scores.

### B.11 Different Total Number of Training Epochs

In this section, we show that our method can achieve acceleration over different training budgets (i.e., number of training epochs), rather than being a pseudo-proposition of better early performance compared to the

baseline method. To demonstrate this, we add experiments under different total number of training epochs and change the learning rate scheduler accordingly to allow convergence.

Take the SET-ITOP as an example. In the main paper, we follow the baseline paper where the epoch number is 250 and the learning rate scheduler is set as the stepwise learning rate with decay points 125 (i.e., $0.5 \times 250$) and 187 (i.e., $0.75 \times 250$). To reduce the epoch number and allow convergence, we set epoch number as 50 and 100 where the decay points is set as $\{25, 37\}$ and $\{50, 75\}$, respectively. As shown in Figure 31, blue curves (our A-SET-ITOP) are usually on top of pink curves (SET-ITOP), implying acceleration from our AGENT.

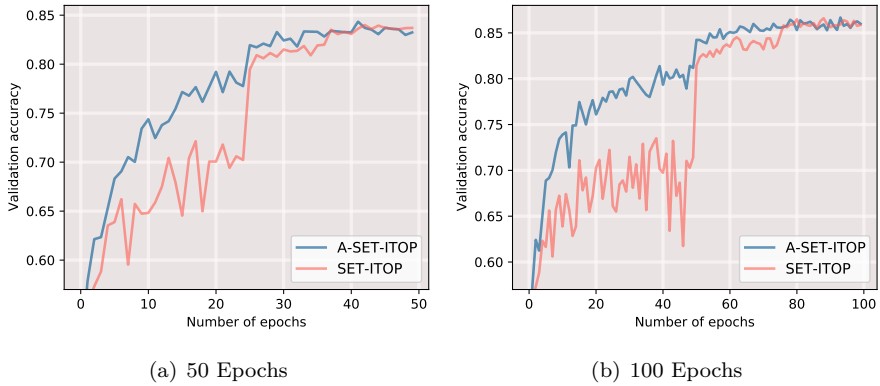

(a) 50 Epochs  (b) 100 Epochs

Figure 31: Comparisons (validation accuracy given the number of epochs) between A-SET-ITOP and SET-ITOP. We evaluate sparse networks (99%) learned with standard training on CIFAR-10 using VGG-C under (a) 50 training epochs, and (b) 100 training epochs.

## B.12 Smoothing Factor Tuning

For smoothing factor $\alpha$, we follow the default value in Deng et al. (2020) which is set as 0.3. We add some experiments to test the influence of $\alpha$. We further compare the validation accuracy across different smoothing factors $\alpha$. As shown in Figure 32, the results of setting $\alpha = 0.05, 0.5, 0.9$ are similar to that of $\alpha = 0.3$. Thus, our method is not sensitive to the choice of $\alpha$, and we can follow the default 0.3.

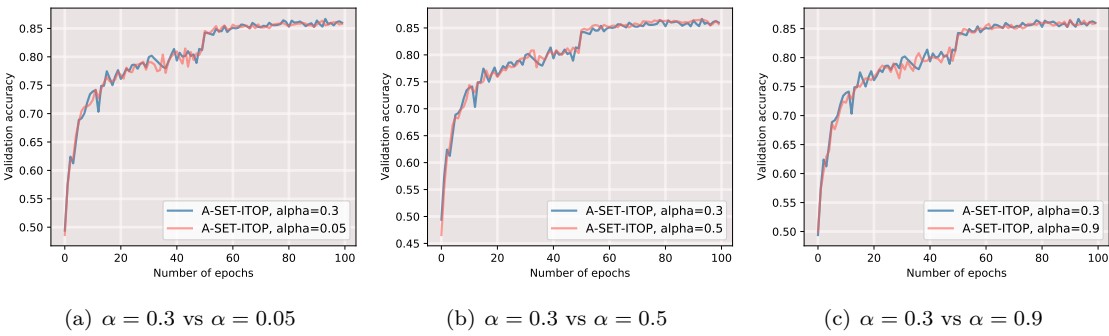

(a) $\alpha = 0.3$ vs $\alpha = 0.05$  (b) $\alpha = 0.3$ vs $\alpha = 0.5$  (c) $\alpha = 0.3$ vs $\alpha = 0.9$

Figure 32: Comparisons (validation accuracy given the number of epochs) between different smoothing factors $\alpha$. We evaluate sparse networks (99%) learned with standard training on CIFAR-10 using VGG-C where we set (a) $\alpha = 0.3$ vs $\alpha = 0.05$, (b) $\alpha = 0.3$ vs $\alpha = 0.5$, and (c) $\alpha = 0.3$ vs $\alpha = 0.9$.

## B.13 Fixed $c_t$ Tuning

We add a more realistic baseline of "Fixed $c_t$" with good hyperparameter tuning to show that adaptive re-weighting is crucial. The term "Fixed $c_t$" corresponds to fixing weight $c_t = 0.1$ during training, which is mentioned in our ablation studies in Section 6.6. Specifically, we further check different $c_t$ in "Fixed $c_t$" and

compare their validation accuracy with our A-SET-ITOP. As shown in Figure 33, when $c_t$ is fixed as 0.001, 0.001, and 0.1, the pink curves ( "Fixed $c_t$") are lower than the blue curve (A-SET-ITOP) in the early stages, indicating slower early convergence in "Fixed $c_t$". When fixing $c_t$ as 0.5, 0.8, and 1.0, the whole pink curves ( "Fixed $c_t$") are below the blue curves (A-SET-ITOP), implying slower convergence in "Fixed $c_t$".

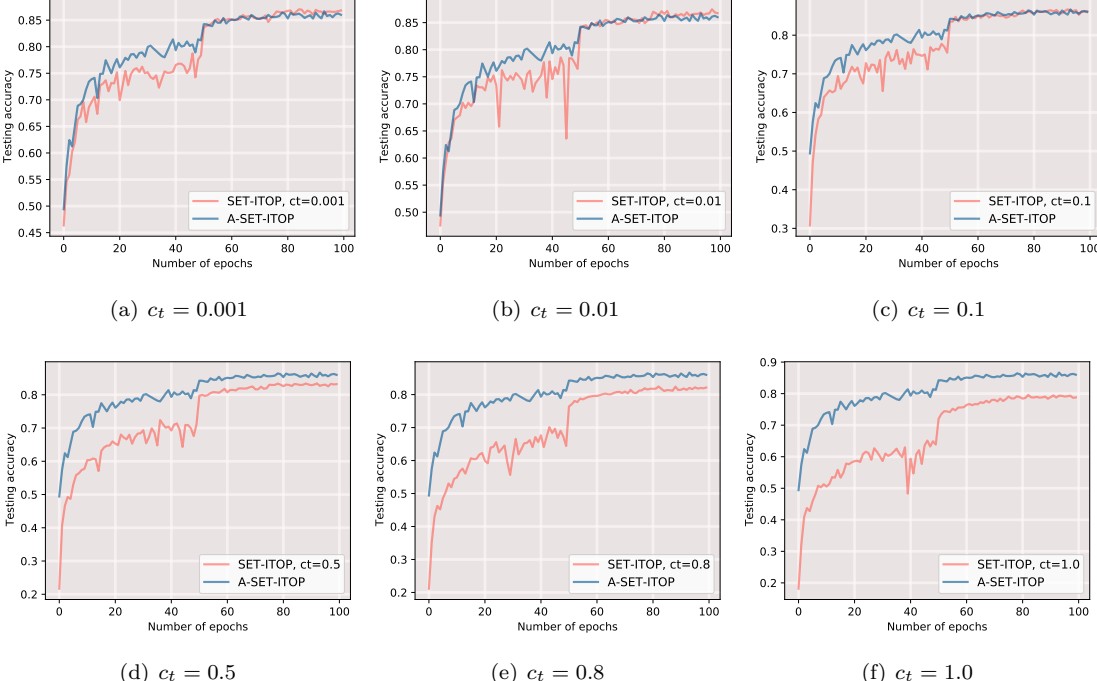

(a) $c_t = 0.001$            (b) $c_t = 0.01$           (c) $c_t = 0.1$

(d) $c_t = 0.5$            (e) $c_t = 0.8$           (f) $c_t = 1.0$

Figure 33: Comparisons (validation accuracy given the number of epochs) between different $c_t$ in "Fixed $c_t$". We evaluate sparse networks (99%) learned with standard training on CIFAR-10 using VGG-C where we compare A-SET-ITOP with (a) $c_t = 0.001$, (b) $c_t = 0.01$, (c) $c_t = 0.1$, (d) $c_t = 0.5$, (e) $c_t = 0.8$, (f) $c_t = 1.0$

### B.14 More Comparison between $\hat{c}_1$ and Gradient Correlation

Our surrogate estimate $\hat{c}_1$ can achieve acceleration although it is not optimal. The idea is illustrated in Figure 2 of Section 4.2. Gradient variance is a quadratic function of $c_t$. Although our $\hat{c}_1$ is not equal to the optimal $c^*$, it can still reduce the variance, leading to acceleration.

We also empirically compare $\hat{c}_1$ and the gradient correlation, where $\gamma = 0.1$. As shown in Figures 34 (a)-(d), $\hat{c}_1$ and gradient correlation during training are compared, which shows a similar up-and-down pattern with different magnitudes. The different magnitudes can be matched via the scaling parameter $\gamma$.

### B.15 Loss Value Comparisons

Apart from accuracy, we also include loss comparison to demonstrate the acceleration. As shown in Figure 35, the blue curves for our A-SET-ITOP are usually below the pink curves for SET-ITOP, implying successful acceleration.

### B.16 More Baseline Comparison

We add more results where ADAM and SVGR are compared together. As shown in Figure 36, the blue curves, pink curves, and green curves represent our AGENT, Adam, and SVRG, respectively. The blue curves of our AGENT are usually higher than the pink and green curves, indicating faster convergence using our AGENT compared to the other two methods.

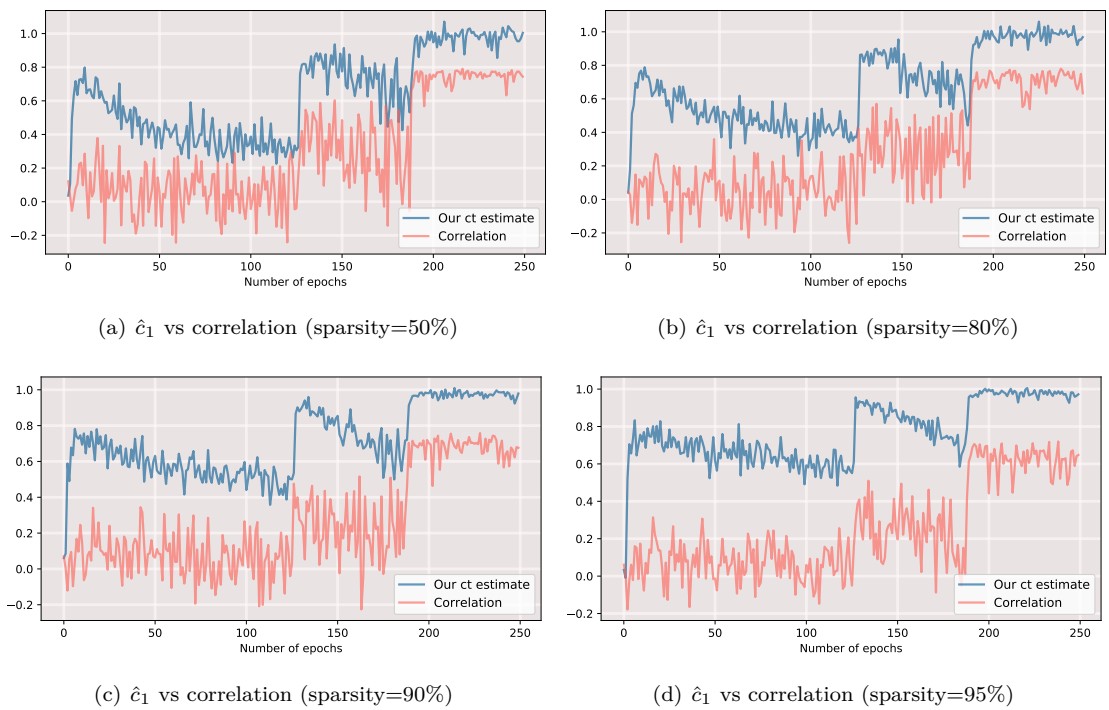

(a) $\hat{c}_1$ vs correlation (sparsity=50%)

(b) $\hat{c}_1$ vs correlation (sparsity=80%)

(c) $\hat{c}_1$ vs correlation (sparsity=90%)

(d) $\hat{c}_1$ vs correlation (sparsity=95%)

Figure 34: Comparison between $\hat{c}_1$ and the gradient correlation. We evaluate sparse networks learned with standard training on CIFAR-10 using VGG-C. (a) $\hat{c}_1$ vs correlation (sparsity=50%), (b) $\hat{c}_1$ vs correlation (sparsity=80%), (c) $\hat{c}_1$ vs correlation (sparsity=90%), and (d) $\hat{c}_1$ vs correlation (sparsity=95%).

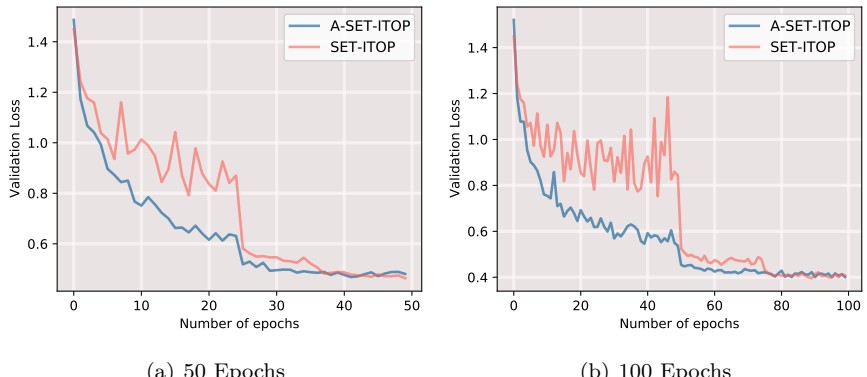

(a) 50 Epochs

(b) 100 Epochs

Figure 35: Comparisons (validation loss given the number of epochs) between A-SET-ITOP and SET-ITOP. We evaluate sparse networks (99%) learned with standard training on CIFAR-10 using VGG-C under (a) 50 training epochs, and (b) 100 training epochs.

### B.17 More Training Time Comparison

We check the training time of our method and baseline methods. For ITOP-based results, the training time ratio between our A-SET-ITOP and SET-ITOP is 5:3, and the training time ratio between our A-RigL-ITOP and RigL-ITOP is 2:1. For BSR-Net based results, the training time ratio between our A-BSR-Net and BSR-Net is 5:4. Despite our current training time does not have advantages over baseline methods, our training time can be easily reduced by the following ways.

- We can use sparse gradients in sparse training, which effectively reduces the cost of backward in sparse training and can be easily applied to our method (Elibol et al., 2020).

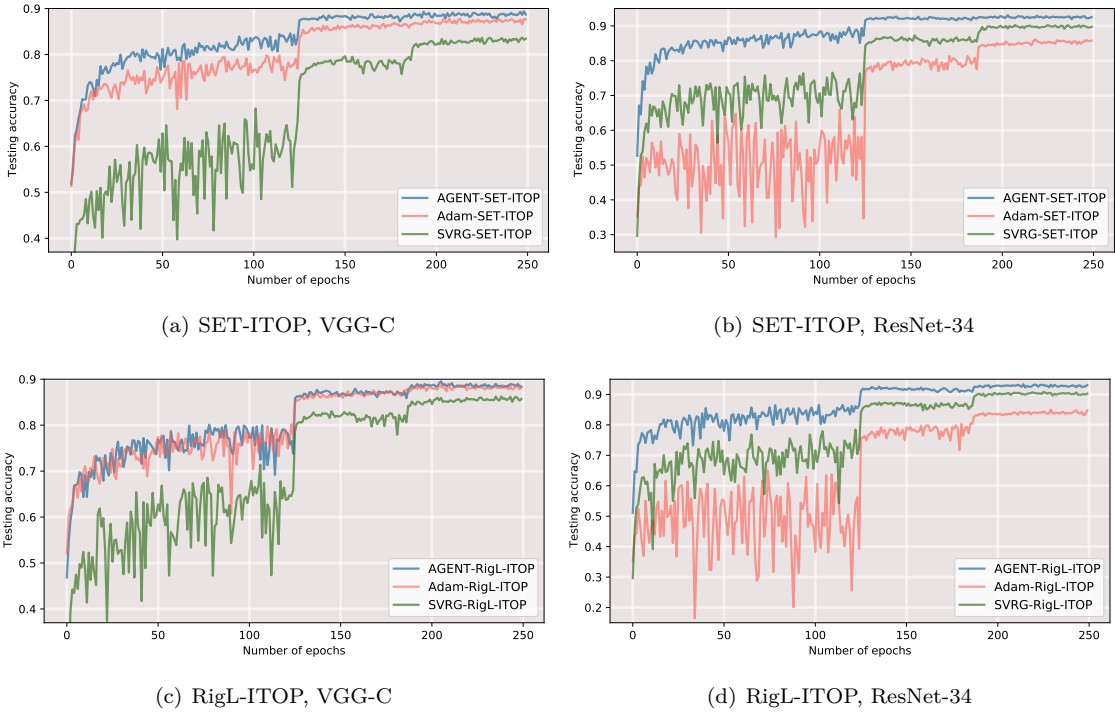

(a) SET-ITOP, VGG-C

(b) SET-ITOP, ResNet-34

(c) RigL-ITOP, VGG-C

(d) RigL-ITOP, ResNet-34

Figure 36: Comparison between our AGENT, Adam, and SVRG. We evaluate sparse networks learned with standard training on CIFAR-10. (a) SET-ITOP, VGG-C, (b) SET-ITOP, ResNet-34, (c) RigL-ITOP, VGG-C, and (d) RigL-ITOP, ResNet-34.

- We can use parallel computing. Since the additional forward and backward over the old model parameters are fully parallelizable, we can view it as doubling the mini-batch size (Allen-Zhu & Hazan, 2016).

- We can follow the idea of SAGA and store gradients for each sample. Then, we do not need extra forward and backward steps, saving the wall-clock time (Defazio et al., 2014).

## C  Additional Details about Experiment Settings

### C.1  Gradient Variance and Correlation Calculation

We calculate the gradient variance and correlation of the ResNet-50 on CIFAR-100 from RigL (Evci et al., 2020) and SET (Mocanu et al., 2018) at different sparsities including 0%, 50%, 80%, 90%, and 95%. The calculation is based on the checkpoints from Sundar & Dwaraknath (2021).

**Gradient variance**: We first load fully trained checkpoints for the 0%, 50%, 80%, 90%, and 95% sparse models. Then, to see the gradient variance around the converged optimum, we add small perturbations to the weights and compute the mean of the gradient variance. For each checkpoint, we do three replicates.

**Gradient correlation**: We begin with fully-trained checkpoints at 0%, 50%, 80%, 90%, and 95% sparsity. We calculate and store the gradient of each weight on all training data. Then, we add Gaussian perturbations to all the weights and calculate the gradients again. Lastly, we calculate the correlation between the gradient of the new perturbed weights and the old original weights. For each checkpoint, we do three replicates.

### C.2 Implementations

**In BSR-Net-based results**, aligned with the choice of Özdenizci & Legenstein (2021), the gradients for all models are calculated by SGD with momentum and decoupled weight decay (Loshchilov & Hutter, 2019). All models are trained for 200 epochs with a batch size of 128.

**In RigL-based results**, we follow the settings in Evci et al. (2020); Sundar & Dwaraknath (2021). We train all the models for 250 epochs with a batch size of 128, and parameters are optimized by SGD with momentum.

**In ITOP-based results**, we follow the settings in Liu et al. (2021). For CIFAR-10 and CIFAR-100, we train all the models for 250 epochs with a batch size of 128. For ImageNet-2012, we train all the models for 100 epochs with a batch size of 64. Parameters are optimized by SGD with momentum.

### C.3 Learning Rate

Aligned with popular sparse training methods (Evci et al., 2020; Özdenizci & Legenstein, 2021; Liu et al., 2021), we choose piecewise constant decay schedulers for learning rate and weight decay. In our A-BSR-Net, we use the 50th and 100th epochs as the dividing points of our learning rate decay scheduler. The reason is that our approach has faster convergence and doesn't require a long warm-up period. In the evaluation shown in the manuscript, we also use this scheduler for BSR-Net for a more accurate and fair comparison.

### C.4 Initialization (BSR-Net-based results)

Consistent with Özdenizci & Legenstein (2021), we also choose Kaiming initialization to initialize the network weights He et al. (2015)

### C.5 Benchmark Datasets (BSR-Net-based results)

For a fair comparison, we choose the same benchmark datasets as Özdenizci & Legenstein (2021). Specifically, we use CIFAR-10 and CIFAR-100 Krizhevsky et al. (2009) and SVHN Netzer et al. (2011) in our experiments. Both CIFAR-10 and CIFAR-100 datasets include 50, 000 training and 10, 000 test images. SVHN dataset includes 73, 257 training and 26, 032 test samples.

### C.6 Data Augmentation

We follow a popular data augmentation method used in Özdenizci & Legenstein (2021); He et al. (2016). In particular, we randomly shift the images to the left or right, crop them back to their original size, and flip them in the horizontal direction. In addition, all the pixel values are normalized in the range of [0, 1].

## D Sparse Training Method Description

### D.1 Sparse Training

Sparse training is a popular method to achieve resource efficiency in deep neural networks (DNNs). Specifically, to obtain a 90% sparse DNN, we randomly initialize a 90% sparse DNN. Then, we maintain sparse weights throughout the training process, pruning and regrowing a certain number of weights every $m$ iterations. Thus, we can save training memory and produce sparse models with dense performance levels.

### D.2 Bayesian Sparse Robust Training

Bayesian Sparse Robust Training (BSR-Net) Özdenizci & Legenstein (2021) is a Bayesian Sparse and Robust training pipeline. Based on a Bayesian posterior sampling principle, a network rewiring process simultaneously learns the sparse connectivity structure and the robustness-accuracy trade-off based on the adversarial

learning objective. More specifically, regarding its mask update, it prunes all negative weights and grows new weights randomly.

# E   Limitations of Our Adaptive Gradient Correction Method

## E.1   Extra FLOPs

Similar to SVRG, our ADSVRG increases the training FLOPs in each iteration due to the extra forward and backward used to compute the old gradients.

However, the true computation difference can be smaller and the GPU-based runining time of SVRG will not be affected that much. For example, in the adversarial setting, we need additional computations to generate the adversarial samples, which is time-consuming and only needs to be done once in each iteration of our AVR and SGD. For BSR-Net, we empirically find that the ratio of time required for each iteration of our AVR and SGD is about 1.2.

There are also several methods to reduce the extra computation caused by SVRG. The first approach is to use the sparse gradients proposed by M Elibol (2020) Elibol et al. (2020). It can effectively reduce the computational cost of SVRG and can be easily applied to our method. The second approach is suggested by Allen-Zhu and Hazan (2016) Allen-Zhu & Hazan (2016). The extra cost on computing batch gradient on old model parameters is totally parallelizable. Thus, we can view SVRG as doubling the mini-batch size. Third, we can follow the idea of SAGA Defazio et al. (2014) and store gradients for individual samples. By this way, we do not need the extra forward and backward step and save the computation. But it requires extra memory to store the gradients.

In the main manuscript, we choose to compare the convergence speed of our ADSVRG and SGD for the same number of pass data (epoch), which is widely used as a criterion to compare SVRG-based optimization and SGD (Allen-Zhu & Hazan, 2016; Chatterji et al., 2018; Zou et al., 2018; Cutkosky & Orabona, 2019). A comparison in this way in this way can demonstrate the accelerating effect of the optimization method and provide inspiration for future work.

## E.2   Scaling parameter tuning

In our adaptive variance reduction method (AVR), we add an additional scaling parameter $\gamma$ which need to be adjusted. We find that setting $\gamma = 0.1$ is a good choice for BSR-Net, RigL, and ITOP. However, it can be different for other different sparse training pipelines.

## E.3   Robust Accuracy Degradation

For the final accuracy results of BSR-Net-based models, there is a small decrease in the robustness accuracy after using our AVR. It is still an open question how to further improve the robust accuracy when using adaptive variance reduction in sparse and adversarial training.

