# OpenReview forum: "Balance is Essence: Accelerating Sparse Training via Adaptive Gradient Correction"
_TMLR — Rejected by TMLR_

### Review · Reviewer_phez · 2023-02-02

**Summary Of Contributions:**

The paper studies the problem of sparse neural network training. A modified SVRG-style estimator is suggested instead of the standard mini-batch stochastic gradient to reduce the variance and thus accelerate and stabilize convergence. The authors argue that the suggested method is theoretically better than standard SVRG in the non-convex setting. Moreover, the benefits of the proposed algorithm, named AGENT (in combination with several sparse training approaches), are extensively experimentally examined for a variety of vision datasets. The method is also shown to perform better than vanilla SVRG, ADAM, and momentum-based variance reduction.

**Audience:**

Yes

**Broader Impact Concerns:**

No concerns.

**Claims And Evidence:**

No

**Requested Changes:**

1. The phrase
> *“Theoretically, we prove that our method can accelerate the convergence rate of sparse training.”*

is unfortunately unjustified and may even be incorrect. As convergence rate (complexity) is typically obtained by equating the convergence upper bound to certain precision $\varepsilon$ and calculating the number of iterations needed to reach arbitrary accuracy $\varepsilon$. It is clear that basic analysis will show that result from Theorem 1 will not lead to better complexity due to the second term. It is not completely clear if it can even be eliminated completely in contrast to the first error term, which goes down to zero as the number of iterations grows.

  - Why is it said that $\sigma^2$ is relatively small? It uniformly upper bounds the norm of the gradient and thus can be potentially arbitrarily large for a quadratic loss function.

  - There are a lot of constants $\nu, \mu, \kappa, \alpha$ mentioned in Theorem 1. It is said that they depend on $\eta_t, c_t, N, n$, some of which are crucial constants needed for accurate comparison of the results and adequate understanding of the convergence rate. More details on this are needed in the main text.

  - The statement
> *“Typically, $L$ in sparse training will be larger than L in dense training.”*

needs at least some justification

  - In total, the analysis is not complete. For making such a statement, it is necessary to show for what kind of regimes (relation between problem constants) this may hold. Potentially it may require some numerical simulations to compute the constants from the upper bounds.

2. The main part misses discussion on the computational trade-offs of using control variates as they create storage and computational overhead. Especially due to the required full (on the whole dataset) gradient computation for $\tilde{g}$. How was it incorporated into the experimental comparison to make it fair against methods which do not need this kind of additional work? For now, the results do not look convincing.

3. I would like the authors to define explicitly and accurately the source of randomness in section 4.1 and what sampling is used to select a random subset $B_t$. As in some instances, this randomness can lead to complete independence, and thus $\mathrm{Cov}$ will be equal to 0.


### Minor comments

- Literature references on variance-reduced optimization are mixed with works on MCMC which may be confusing for some readers. In my view, it would be better to separate them, as the original idea was first proposed in the optimization community for finite-sum minimization problems. SAG and SAGA papers are missing in section 2.2. In general, I recommend having a look at the paper [2] for references on variance-reduced methods.

- The phrase “They usually assume that current and previous gradients are highly correlated” is not accurate as the works analyzing variance-reduced methods, such as (Gorbunov et al., 2020), do not make this kind of assumption.

- The idea of using factor $c_t$, as it is not novel, should be credited to Monte Carlo variance reduction methods. It can be found in classical books such as [3].

- The wording choice “approximation algorithm” (in the last paragraph on page 4) seems a bit misleading as there is no theoretical justification for that.

- It is not very clear how AGENT is combined with MVR, as both of them are variance-reduced methods.

- Some issue in the second line of the A.4.1 lemma 1 proof. Probably missing squared norm for vector $\tau_t$.

- Not capitalized letters for MCMC, SGD, Monte Carlo, Langevin, and SAGA in the references.

- Duplicated bibliography entry: Guillaume Bellec, David Kappel, Wolfgang Maass, and Robert Legenstein. Deep rewiring: Training very sparse deep networks.


#### Typos
- Page 8, line 2 from the bottom: “sutups” -> “setups”
- Page 4, paragraph 5:  “closed” -> “close”

----

[2] Gower, Robert M., et al. "Variance-reduced methods for machine learning." Proceedings of the IEEE 108.11 (2020): 1968-1983.

[3] Asmussen, Søren, et al. "Stochastic Optimization." Stochastic Simulation: Algorithms and Analysis (2007): 242-258.

**Strengths And Weaknesses:**

# Strengths

**S1.** An important and interesting problem is considered with an approach that seems to be not very well explored.

**S2.** The suggested approach to variance reduction looks creative in the sparse training context, while it was previously considered not useful in Deep Learning settings [2].

**S3.** Numerical simulations are extensive and include a comparison to other methods on various tasks. An ablation study for the parameters of the method is also presented.

**S4.** The method is supported by some theoretical analysis.


# Weaknesses

**W1.** Some of the claims in the paper are not well-supported or not very accurately stated. Evidence is lacking in multiple places (more details on it later).

**W2.** Writing is not very clear in some parts.

For instance, some of the terms are not formulated/defined: it is unclear (without an explicit formula) how *"Gradient Correlation"* was calculated for Figure 1(b).

Besides, more details on sparse training would also be helpful as, for now, it is not understandable (at least for me) what problem and approach are exactly considered (short descriptions at the beginning of Section 6 do not seem to be enough). E.g., is the network initially sparsified, and then the resulting submodel is trained without changes? Or the model is also adjusted during optimization so that sparsifying masks change throughout iterations.

Considering both convergence speed and robustness seems intriguing but may be confusing for some of the readers, in my view. The authors touch upon adversarial training, but slightly. So, it does not bring novel insights but complicates and distracts from the main message.

**W3.**  There is a serious issue related to replacing (empirical) covariance between stochastic gradients with losses. The statement “loss is intuitively related to gradients“ is questionable as optimal methods for non-convex optimization (in some sense designed to minimize the expected norm of the gradient as fast as possible) do not typically work well (do not minimize loss so efficiently) for neural networks in comparison to ADAM and momentum SGD. Moreover, there are situations in deep learning optimization when the loss goes down to zero, but the norm of the gradient is not. In addition, the authors state that “$\hat{c_t^*}$ and the correlation have similar up-and-down patterns”, which can not serve as a piece of evidence for the proposed hypothesis/assumption. For example, for me, the plots in the Appendix just do not follow similar patterns. It was not tested qualitatively.

**W4.** The non-convex analysis of SVRG seems quite outdated as it relies on bounded gradient assumption, which is very restrictive (e.g., it may not hold even for simple quadratic functions) and is not needed to obtain this kind of convergence guarantee. Ideally it would be revisited with the help of a more recent approach.

----

[1] Defazio, Aaron, and Léon Bottou. "On the ineffectiveness of variance reduced optimization for deep learning." Advances in Neural Information Processing Systems 32 (2019).

---

> ### Author Response · Authors · 2023-03-04
> **Response to reviewer phez (Q1-Q4)**
>
> Thank you for the review and for bringing up these questions. We have revised the paper accordingly and the main changes are highlighted in blue. Please see our responses below to clarify the main concerns.
>
> **Q1: For instance, some of the terms are not formulated/defined: it is unclear (without an explicit formula) how "Gradient Correlation" was calculated for Figure 1(b).**
>
> A1: The details of the calculation are included in section C.1.
> * We calculate the gradient correlation of the ResNet-50 on CIFAR-100 from RigL and SET at different sparsities including 0%, 50%, 80%, 90%, and 95%.
> *  We begin with fully-trained checkpoints at 0%, 50%, 80%, 90%, and 95% sparsity.
> *  We calculate and store the gradient of each weight on all training data. Then, we add Gaussian perturbations to all the weights and calculate the gradients again.
> *  Lastly, we calculate the correlation between the gradient of the new perturbed weights and the old original weights. For each checkpoint, we do three replicates.
>
> **Q2: Besides, more details on sparse training would also be helpful as, for now, it is not understandable (at least for me) what problem and approach are exactly considered (short descriptions at the beginning of Section 6 do not seem to be enough). E.g., is the network initially sparsified, and then the resulting submodel is trained without changes? Or the model is also adjusted during optimization so that sparsifying masks change throughout iterations.**
>
> A2: We add more details on sparse training in Section D.1. In our study, we focus on the case where the network is initially sparsified and the sparse mask is also dynamically adjusted every $m$ iteration.
> * Sparse training is a popular method to achieve resource efficiency in deep neural networks (DNNs).
> * Specifically, to obtain a 90% sparse DNN, we randomly initialize a 90% sparse DNN.
> * Then, we maintain sparse weights throughout the training process, pruning and regrowing a certain number of weights every $m$ iteration.
> * Thus, we can save training memory and produce sparse models with dense performance levels.
>
> **Q3: Considering both convergence speed and robustness seems intriguing but may be confusing for some of the readers, in my view. The authors touch upon adversarial training, but slightly. So, it does not bring novel insights but complicates and distracts from the main message.**
>
> A3: Our goal is to accelerate sparse training under both standard and adversarial setups.
>
> (i) Considering the adversarial setup is important [R1].
> * Vulnerability to adversarial samples is one of the weaknesses of DNNs
> * Sparse training can exacerbate models' vulnerability to adversarial samples.
>
> (ii) Considering simultaneously sparse and adversarial setups is not a trivial task [R2].
> * Adversarial training can add additional bias in the gradient estimate.
> * The bias makes the gradient more inaccurate and slows down the convergence rate.
>
> [R1] Ozan Özdenizci, et al. Training adversarially robust sparse networks via bayesian connectivity sampling, 2021 ICML.
>
> [R2] Yan Li, et al. Implicit bias of gradient descent based adversarial training on separable data, ICLR 2020.
>
> **Q4: There is a serious issue related to replacing (empirical) covariance between stochastic gradients with losses. The statement “loss is intuitively related to gradients“ is questionable as optimal methods for non-convex optimization (in some sense designed to minimize the expected norm of the gradient as fast as possible) do not typically work well (do not minimize loss so efficiently) for neural networks in comparison to ADAM and momentum SGD.**
>
> A4: Our surrogate estimate can achieve acceleration although it is not optimal.
>
> (i) The idea is illustrated in Figure 2 of Section 4.2.
> * Gradient variance is a quadratic function of $c_t$.
> * Although our $\gamma\hat{c}_1$ is not equal to the optimal $c^*$, it can still reduce the variance, leading to acceleration.
>
> Our surrogate estimate is aligned with our theoretical analysis.
>
> (i) Our estimate leads to a better bound in our theory which is illustrated in Section A.6.
>
> (ii) We also empirically compare $\hat{c}_1$ and gradient correlation. The results are summarized in Section B.14.
> * As shown in Figure 34, $\hat{c}_1$ and gradient correlation during training are compared, which shows a similar up-and-down pattern with different magnitudes.
> * The different magnitudes can be matched via the scaling parameter $\gamma$.
>
> Adam and momentum SGD are designed for dense training, while sparse training has its own characteristics, which are taken into account in our method.

---

> > ### Author Response · Authors · 2023-03-04
> > **Response to reviewer phez (Q5-Q10)**
> >
> > **Q5: Moreover, there are situations in deep learning optimization when the loss goes down to zero, but the norm of the gradient is not. In addition, the authors state that “$\hat{c}_t^{*}$ and the correlation have similar up-and-down patterns”, which can not serve as a piece of evidence for the proposed hypothesis/assumption. For example, for me, the plots in the Appendix just do not follow similar patterns. It was not tested qualitatively.**
> >
> > A5: (i) We think the loss correlation can help estimate the gradient correlation.
> >
> > (ii) The absolute values of the losses and gradients can be very different.
> >
> > (iii) We add more experiments in Section B.14 shows the similar up-and-down pattern between $\hat{c}_1$ and gradient correlation.
> >
> > **Q6: The non-convex analysis of SVRG seems quite outdated as it relies on bounded gradient assumption, which is very restrictive (e.g., it may not hold even for simple quadratic functions) and is not needed to obtain this kind of convergence guarantee. Ideally, it would be revisited with the help of a more recent approach.**
> >
> > A6: Our $\sigma$-bounded assumption follows the literature on gradient correction methods [R3, R4]. We will investigate how to remove this assumption in future studies.
> >
> > [R3] Sashank J. Reddi, et al. Stochastic variance reduction for nonconvex optimization, ICML 2016.
> >
> > [R4] Ashok Cutkosky, et al. Momentum-based variance reduction in non-convex sgd. NeurIPS 2019.
> >
> > **Q7: The phrase "Theoretically, we prove that our method can accelerate the convergence rate of sparse training." is unfortunately unjustified and may even be incorrect. As convergence rate (complexity) is typically obtained by equating the convergence upper bound to certain precision $\epsilon$ and calculating the number of iterations needed to reach arbitrary accuracy $\epsilon$. It is clear that basic analysis will show that results from Theorem 1 will not lead to better complexity due to the second term. It is not completely clear if it can even be eliminated completely in contrast to the first error term, which goes down to zero as the number of iterations grows.**
> >
> > A7: (i) Our analysis of the convergence rate follows [R3].
> >
> > (ii) We calculate the second term in Section A.6 and prove that the second term is negligible.
> >
> > [R3] Sashank J. Reddi, et al. Stochastic variance reduction for nonconvex optimization, ICML 2016.
> >
> > **Q8: Why is it said that $\sigma^2$ is relatively small? It uniformly upper bounds the norm of the gradient and thus can be potentially arbitrarily large for a quadratic loss function.**
> >
> > A8: (i) This assumption follows [R3].
> >
> > (ii) In the sparse training of deep neural networks, we usually design a good initialization strategy with not too large losses.
> >
> > (iii) During the training process, the loss decreases rapidly in the early stages and then slowly decreases to the optimal value. Therefore, the losses are not too large.
> >
> > [R3] Sashank J. Reddi, et al. Stochastic variance reduction for nonconvex optimization, ICML 2016.
> >
> > **Q9: There are a lot of constants mentioned in Theorem 1, and some of which are crucial constants needed for accurate comparison of the results and adequate understanding of the convergence rate. It is necessary to show for what kind of regimes (relation between problem constants) this may hold. Potentially it may require some numerical simulations to compute the constants from the upper bounds.**
> >
> > A9: (i) Our estimate $c$ leads to a better bound in our theory which is illustrated in Section A.6.
> >
> > (ii) We investigate the values of these constants in real data scenarios.
> > * The exact conditions rely on the specific dataset and training hyperparameters.
> > * In widelu-used dataset, e.g., CIFAR-10/100 and SVHN, the term $\frac{2 \kappa \mu^2 \sigma^2}{ N^\alpha \nu m}$ is small and negligible. Specifally, $\frac{2 \kappa \mu^2 \sigma^2}{ N^\alpha \nu m}$ is around $10^{-5}$ and $10^{-4}$ for CIFAR-10/100 and SVHN, respectively.
> >
> > **Q10: The statement “Typically, $L$ in sparse training will be larger than L in dense training.” needs at least some justification.**
> >
> > A10: The value of $L$ is related to the gradient correlation.
> >
> > (i) $L$ denotes the degree of change in the gradient when there is a small change in the input.
> >
> > (ii) Gradient correlation is the correlation between gradients at current parameters and previous epoch parameters.
> >
> > (ii) Thus, $L$ tends to be larger if the gradient correlation is smaller.
> >
> > (iv) We observe a lower gradient correlation during sparse training in Section B.6, implying larger $L$ in sparse training.

---

> > > ### Author Response · Authors · 2023-03-04
> > > **Response to reviewer phez (Q11-Q16)**
> > >
> > > **Q11: The main part misses discussion on the computational trade-offs of using control variates as they create storage and computational overhead. Especially due to the required full (on the whole dataset) gradient computation for $\hat{g}$. How was it incorporated into the experimental comparison to make it fair against methods which do not need this kind of additional work? For now, the results do not look convincing.**
> > >
> > > A11: (i) Our current training time does not have advantages over baseline methods.
> > >
> > > (ii) Our training time can be easily reduced by several methods.
> > > * We can use sparse gradients in sparse training, which effectively reduces the cost of backward in sparse training and can be easily applied to our method [R4].
> > > * We can use parallel computing. Since the additional forward and backward over the old model parameters are fully parallelizable, we can view it as doubling the mini-batch size [R5].
> > > * We can follow the idea of SAGA and store gradients for each sample. Then, we do not need extra forward and backward steps, saving the wall-clock time [R6].
> > >
> > > [R4] Zeyuan Allen-Zhu, et al. Variance reduction for faster non-convex optimization. ICML 2016.
> > >
> > > [R5] Melih Elibol, et al. Variance reduction with sparse gradients. arXiv preprint arXiv:2001.09623, 2020.
> > >
> > > [R6] Aaron Defazio, et al. Saga: A fast incremental gradient method with support for non-strongly convex composite objectives. NeurIPS 2014.
> > >
> > > **Q12: I would like the authors to define explicitly and accurately the source of randomness in section 4.1 and what sampling is used to select a random subset $B_t$. As in some instances, this randomness can lead to complete independence, and thus $Cov$ will be equal to 0.**
> > >
> > > A12: (i) The random batch $B_t$ is randomly selected from the full data.
> > >
> > > (ii) Correlation is between new and previous model parameters, not between different batches of data. Thus, $Cov$ will not be equal to 0.
> > >
> > > **Q13: Literature references on variance-reduced optimization are mixed with works on MCMC which may be confusing for some readers. In my view, it would be better to separate them, as the original idea was first proposed in the optimization community for finite-sum minimization problems. SAG and SAGA papers are missing in section 2.2. In general, I recommend having a look at the paper [R7] for references on variance-reduced methods.**
> > >
> > > A13: (i) The key idea of using variance reduction in SGLD and SGD is similar. Therefore, our approach can be applied to both MCMC and optimization processes.
> > >
> > > (ii) We follow the reviewer's suggestion and add SAG, SAGA, and [R5] in Section 2.2.
> > >
> > > [R7] Gower, Robert M., et al. "Variance-reduced methods for machine learning." Proceedings of the IEEE 108.11 (2020): 1968-1983.
> > >
> > > **Q14: The phrase “They usually assume that current and previous gradients are highly correlated” is not accurate as the works analyzing variance-reduced methods, such as (Gorbunov et al., 2020), do not make this kind of assumption.**
> > >
> > > A14: (i) Existing work usually uses $c=1$, which is optimal when the current and previous gradients are highly correlated.
> > >
> > > (ii) Thus, existing work will perform well when current and previous gradients are highly correlated.
> > >
> > > **Q15： The idea of using factor, as it is not novel, should be credited to Monte Carlo variance reduction methods. It can be found in classical books such as [R8].**
> > >
> > > A15: (i) Our method takes into account the characteristics of sparse training and thus is designed for accelerated sparse training.
> > >
> > > (ii) [R8] consider the case of dense training, which is different from our method.
> > >
> > > [R8] Asmussen, Søren, et al. "Stochastic Optimization." Stochastic Simulation: Algorithms and Analysis (2007): 242-258.
> > >
> > > **Q16: The wording choice “approximation algorithm” (in the last paragraph on page 4) seems a bit misleading as there is no theoretical justification for that.**
> > >
> > > A16: Our surrogate estimate can achieve acceleration although it is not optimal.
> > >
> > > (i) The idea is illustrated in Figure 2 of Section 4.2.
> > > * Gradient variance is a quadratic function of $c_t$.
> > > * Although our $\gamma\hat{c}_1$ is not equal to the optimal $c^*$, it can still reduce the variance, leading to acceleration.
> > >
> > > Our surrogate estimate is aligned with our theoretical analysis.
> > >
> > > (i) Our estimate leads to a better bound in our theory which is illustrated in Section A.6.
> > >
> > > (ii) We also empirically compare $\hat{c}_1$ and gradient correlation. The results are summarized in Section B.14.
> > > * As shown in Figure 34 (a)-(d), $\hat{c}_1$ and gradient correlation during training are compared, which shows a similar up-and-down pattern with different magnitudes.
> > > * The different magnitudes can be matched via the scaling parameter $\gamma$.

---

> > > > ### Author Response · Authors · 2023-03-04
> > > > **Response to reviewer phez (Q17-Q19)**
> > > >
> > > > **Q17: It is not very clear how AGENT is combined with MVR, as both of them are variance-reduced methods..**
> > > >
> > > > A17: (i) The key parts of our method are the proposed adaptive coefficient and scaling parameter.
> > > >
> > > > (ii) Thus, we can combine our method with MVR by changing the gradient update to $d_t=(1-a_t)d_{t-1} + a_t \nabla f(x_t, \xi_t) + (1-a_t)(\nabla f(x_t,\xi_t) - \nabla f(x_{t-1},\xi_t))$, where $a_t=\frac{k}{(w+\sum_{i=1}^t G_t^2)^{1/3}}$.
> > > >
> > > > **Q18: Not capitalized letters for MCMC, SGD, Monte Carlo, Langevin, and SAGA in the references.**
> > > >
> > > > A18: We update the references and capitalize MCMC, SGD, Monte Carlo, Langevin, and SAGA.
> > > >
> > > > **Q19: Duplicated bibliography entry: Guillaume Bellec, David Kappel, Wolfgang Maass, and Robert Legenstein. Deep rewiring: Training very sparse deep networks.**
> > > >
> > > > A19: We have solved the issue in the new version.

---

> > > ### Comment · Reviewer_phez · 2023-03-20
> > > **Further question 2**
> > >
> > > Could you please clarify how did you compute the term $\frac{2 \kappa \mu^{2} \sigma^{2}}{N^{\alpha} \nu m}$ as it includes a uniform upper bound $\sigma$ on the norm of the gradient of the loss functions over the whole space $\mathbb{R}^d$? The same question arises regarding the smoothness constant $L$.

---

> > > > ### Author Response · Authors · 2023-03-20
> > > > **Response to further question 2**
> > > >
> > > > Thank you very much for your reply and the valuable comment!
> > > >
> > > > (i) We clarify the calculation of the term $\frac{2 \kappa \mu^2 \sigma^2}{ N^\alpha \nu m}$ as below.
> > > >
> > > > For CIFAR10/100 dataset,
> > > > * We have $\eta = 0.1$, $\gamma = 0.1$, $m = 391$, $\mu=0.1$, $n=128$, and in total $N = 50000$ training samples.
> > > > * A uniform upper bound $\sigma$ is set as 0.1 based on empirical study.
> > > > * Since $ m = \frac{N^{\frac{3\alpha}{2}}}{\mu n} $, we can get $\alpha = 0.52$. Since $\kappa$ can be bounded by $m$, we use $\kappa=m$ to obtain an upper bound of the term.
> > > > * Under this parameter setting, $\nu$ and $\nu^{*}$in Theorem 1 and Remark 4 are about $0.1$ and $0.06$, respectively.
> > > > * Then, we can calculate $\frac{2 \kappa \mu^2 \sigma^2}{ N^\alpha \nu m}$, which is less than $10^{-5}$ and is negligible.
> > > >
> > > > For SVHN dataset,
> > > > * We have $\eta = 0.1$, $\gamma = 0.1$, $m = 573$, $\mu=0.1$, $n=128$, and sample size $N = 73257$.
> > > > * A uniform upper bound $\sigma$ is set as 0.1 based on empirical study.
> > > > * Since $ m = \frac{N^{\frac{3\alpha}{2}}}{\mu n} $, we can get $\alpha = 0.53$. Since $\kappa$ can be bounded by $m$, we use $\kappa=m$ to obtain an upper bound of the term.
> > > > * Under this parameter setting, $\nu$, $\nu^{*}$ equal 0.4 and 0.06, respectively.
> > > > * Then, we can compute $\frac{2 \kappa \mu^2 \sigma^2}{ N^\alpha \nu m}$, which is smaller than $10^{-4}$ and is negligible.
> > > >
> > > > (ii) For the smoothness constant $L$, it represents gradient change speed and is related to the gradient correlation.
> > > > * We do not assume its exact value, which depends on a variety of factors.
> > > > * $L$ tends to be larger if the gradient correlation is smaller.
> > > > * In Section B.6, we discuss the gradient change speed and correlation from both an intuitive perspective and empirical aspects.
> > > > * We find that in the same setup, correlations tend to be smaller and $L$ tends to be larger in sparse training compared to dense training.

---

> > ### Comment · Reviewer_phez · 2023-03-20
> > **Further question**
> >
> > Thank you for the response and clarifications.
> >
> > Regarding **A1**, could you please write the mathematical expressions you use to compute the correlation between gradients in the paper?

---

> > > ### Author Response · Authors · 2023-03-20
> > > **Response to further question**
> > >
> > > Thank you very much for your reply and the valuable comment!
> > >
> > > The mathematical expressions for the correlation between gradients in A1 are as below.
> > > * We begin with weights $\theta=\\{\theta_1,\cdots, \theta_L\\}$ of the DNN which has $L$ layers.
> > > * We calculate the gradient of weights in each layer, i.e., $g=\\{ g_1,\cdots, g_L \\}$.
> > > * We add Gaussian perturbations to all the weights and get $\theta^*=\\{ \theta_1^*,\cdots, \theta_L^* \\}$ where $\theta_l^*=\theta_l+\epsilon$ and $\epsilon \sim N(0, \sigma_0^2)$.
> > > * We calculate the gradient of the perturbed weights $\theta^*$ in each layer, i.e., $g^*=\\{ g_1^*,\cdots, g_L^* \\}$.
> > > * Then, we calculate the correlation between $g$ and $g^*$ in each layer, i.e., $c_l = cor(g_l, g_l^*), l=1,\cdots,L$.
> > > * Therefore, we can calculate the average correlation of each layer and use it as the output correlation: $c_o = \frac{1}{L} \sum_{l=1}^L c_l$.

---

### Review · Reviewer_h9qB · 2023-02-13

**Summary Of Contributions:**

The paper proposes a modified gradient descent method for optimization, with the aim of reducing the variances of the stochastic gradients based on the estimates of the correlations between old and new gradient values. The paper, therefore, introduces an extra parameter to balance the tradeoff between the old and the new gradient. Under certain assumptions, the paper shows that with properly chosen balance parameters, the proposed method could have a better convergence bound compared to the bound of SVRG. The paper shows improved empirical performance on sparse training tasks on some datasets.

**Audience:**

Yes

**Claims And Evidence:**

Yes

**Requested Changes:**

1. More clear theoretical justification on the chosen c value as stated in the weakness part.
2. Add loss values as additional metrics for some of the experiments to demonstrate improved convergence.
3. Show comparisons to other optimization methods in multiple settings and compare them in the same figure.

Minor:
1. Using G to denote the loss function could be confusing.
2. Section 4.1, second para: "variance of updated gradient" -> "variance of the updated gradient"; "is not closed to 1" -> "is not close to 1"
3. Statement of Assumption 1 contains some grammar errors and I cannot understand it in its current form.
4. Table 1 is confusing. Recommend reorganizing it, e.g., separate the robust training v.s. standard training results into two tables.

**Strengths And Weaknesses:**

Strengths:
1. The proposed method is simple and easy to implement.
2. The paper provides some theoretical justifications for the proposed method.
3. The method shows improved performance on sparse training tasks for both standard and robust training objectives.

Weaknesses:
1. There is a gap between the theory and the algorithm. Firstly, it is hard to estimate the actual correlation between the gradients and a surrogate estimate is utilized. Secondly, the theory only covers properly chosen c values, and we do not know how that is connected to the c and \gamma values used in the algorithm.
2. The theory on the chosen c values requires more justification. I think more details should be given in Remarks 4 and 5 to precisely define under what conditions, and for which exact c values, the proposed method gives a better bound.
3. For experiments, as the author claims to have faster convergence, it is also critical to compare the loss values in addition to the accuracies.
4. Comparisons with other optimization methods are not shown comprehensively and clearly. E.g., the comparison to ADAM and SVGR are shown in two separate figures and only under the 99% sparse training setting.
5. The writing of the paper requires improvement (see some typos in the Minor part). There are occasionally unnecessary replicated statements.

---

> ### Author Response · Authors · 2023-03-04
> **Response to reviewer h9qB (Q1-Q8)**
>
> Thank you for the valuable comments and for bringing up these questions. We have revised the paper accordingly and the main changes are highlighted in blue. Please see our responses below to clarify the main concerns.
>
> **Q1: There is a gap between the theory and the algorithm. Firstly, it is hard to estimate the actual correlation between the gradients and a surrogate estimate is utilized. Secondly, the theory only covers properly chosen c values, and we do not know how that is connected to the c and $\gamma$ values used in the algorithm.**
>
> A1: Our surrogate estimate can achieve acceleration although it is not optimal.
>
> (i) The idea is illustrated in Figure 2 of Section 4.2.
> * Gradient variance is a quadratic function of $c_t$.
> * Although our $\gamma\hat{c}_1$ is not equal to the optimal $c^*$, it can still reduce the variance, leading to acceleration.
>
> Our surrogate estimate is aligned with our theoretical analysis.
>
> (i) Our estimate leads to a better bound in our theory which is illustrated in Section A.6.
>
> (ii) We also empirically compare $\hat{c}_1$ and gradient correlation. The results are summarized in Section B.14.
> * As shown in Figure 34, $\hat{c}_1$ and gradient correlation during training are compared, which shows a similar up-and-down pattern with different magnitudes.
> * The different magnitudes can be matched via the scaling parameter $\gamma$.
>
> **Q2: The theory on the chosen c values requires more justification. I think more details should be given in Remarks 4 and 5 to precisely define under what conditions, and for which exact c values, the proposed method gives a better bound.**
>
> A2: Our estimate $c$ leads to a better bound in our theory which is illustrated in Section A.6.
> * The exact conditions rely on the specific dataset and training hyperparameters.
> * In widelu-used dataset, e.g., CIFAR-10/100 and SVHN, the term $\frac{2 \kappa \mu^2 \sigma^2}{ N^\alpha \nu m}$ is small and negligible. Specifally, $\frac{2 \kappa \mu^2 \sigma^2}{ N^\alpha \nu m}$ is around $10^{-5}$ and $10^{-4}$ for CIFAR-10/100 and SVHN, respectively.
>
> **Q3: For experiments, as the author claims to have faster convergence, it is also critical to compare the loss values in addition to the accuracies.**
>
> A3: We have added more results of loss comparisons in Section B.15. As shown in Figure 35, the blue curves for our A-SET-ITOP are usually below the pink curves for SET-ITOP, implying successful acceleration.
>
> **Q4: Comparisons with other optimization methods are not shown comprehensively and clearly. E.g., the comparison to ADAM and SVGR are shown in two separate figures.**
>
> A4: We add more results in Section B.16, where ADAM and SVGR are compared together.
> * As shown in Figure 36, the blue curves, pink curves, and green curves represent our AGENT, Adam, and SVRG, respectively.
> *  The blue curves of our AGENT are usually higher than the pink and green curves, indicating faster convergence using our AGENT compared to the other two methods.
>
> **Q5: Using G to denote the loss function could be confusing.**
>
> A5: To avoid confusion, we use $l$ instead of $G$ to denote the loss function.
>
> **Q6: Section 4.1, second para: "variance of updated gradient" -> "variance of the updated gradient"; "is not closed to 1" -> "is not close to 1"**
>
> A6: We have updated the manuscript following the reviewer's suggestions.
>
> **Q7: Statement of Assumption 1 contains some grammar errors and I cannot understand it in its current form.**
>
> A7: We have updated Assumption 1 in the new version.
>
> **Q8: Table 1 is confusing. Recommend reorganizing it, e.g., separate the robust training v.s. standard training results into two tables.**
>
> A8: We have separated Table 1 into two Table 1 and Table 2 following the reviewer's suggestions.
> * Table 1 shows the accuracy of standard training.
> * Table 2 shows the accuracy of adversarial training.

---

### Review · Reviewer_th9n · 2023-02-21

**Summary Of Contributions:**

This paper proposes a modification of the SVRG algorithm to re-weight the epoch gradient and the mini-batch gradient in an adaptive manner.
The authors apply their new method, called AGENT, to sparse training where the learned model is constrained to have few non-zero weights (1 to 10%).
In this setup, they motivate the use of their method by showing that gradients exhibit less correlation which explains why previous variance reduction fails at improving convergence.

The new method estimates the adaptive weight $c_t$ with an approximation using two new hyper-parameters: $\alpha$ a smoothing factor acting as momentum to estimate $c_t$, and a scaling factor $\gamma$. They prove that this method induces an improvement of the constant of the main term of the convergence rate compared to standard SVRG and introduces only a negligible term.

**Audience:**

No

**Broader Impact Concerns:**

 No concerns for this submission.

**Claims And Evidence:**

No

**Requested Changes:**

I would like to propose the following changes to improve the paper

1. Provide extensive explanations on your methodology to tune the learning rate, the smoothing factor $\alpha$ and scaling factor $\gamma$ for each method. You should convince me that you are not using the test accuracy as a metric to pick $\alpha$ and $\gamma$ and that all methods have their learning rate properly tuned.

2. Add a more realistic baseline of fixed $c_t$ with good hyperparameter tuning for every experiments to showcase that *adaptive* re-weighting it crucial, only comparing to $c_t=0.1$ is arbitrary.

2. Report the clock time for the different methods in every experiments to give a clear view for practitioners, at least in Appendix with a mention in the main text.

3. Report the results on ImageNet in the main paper.

4. Improve the covariance analysis of the gradient to motivate your method by studying (a) gradients over the course of training (b) applying gradient updates rather than Gaussian perturbations.

5. Replace the "up to 52.1% less epoch" claim by a typical speedup across multiple settings rather than the most optimist estimate.

5. (optional) I would argue that AdSVRG left in Appendix E.1 is a better name than the contrived name AGENT (Adaptive GradiENt correcTion).



Minor (typos and improvements):
- Equation in assumption 1: bold x and y
- Beginning of section 6, three -> four pipelines
- In Table 1, name the epoch column and remove the repeating th
- sutups -> setups (page 8)
- In named paragraph on page 9: directly name the criterions instead of referring to (a) and (b)
- Figure 6: use another color than pink for Adam (distinct from any color in Figure 5)
- Clarify which dataset Section 6.5 and Table 4 are based on.
- Should the step size be $\eta_t$ instead of $h_t$ in the algorithm in Appendix A1?
- Capitalize titles in sections A4 and A4.1
- Rename "ADSVRG" in Appendix B.4 and E.1.




**Strengths And Weaknesses:**


The proposed method exhibits an improvement in final accuracy (Section 6.2) and this is to me the most impressive contribution of this method. If this is not due to hyper-parameters tunning on the test metric, this is a strong point for AGENT. This point is more convincing than being faster in terms of iterations, as what matters for practitioners is speedup in terms of wall-clock time which I suspect is not improved.

The analysis of the covariance between gradients displayed in Figure 1 and explained in Section B.6 could be more realistic. It would be more interesting to study these covariances over the course of the training instead of taking a fully trained model. Moreover, only adding random noise to the trained weights is not reasonable. I would suggest doing the analysis on the "real" gradients and weights over training, maybe zooming on epochs 0, 50, and 100. Also does this analysis transfer across architectures and training procedures?

The theory does not give a hint at how to tune the two new hyper-parameters except that they should lie in $[0,1]$.

I find the methodology for tuning the new method and the baseline very weak. Section E.2 only states that the scaling parameter $\gamma$ was set to 0.1 without explanation. At the same time, Section B.3 illustrates the good choice of $\gamma=0.1$ by looking at the test set without mentioning a validation set. Can you also clarify how $\alpha$ was tuned?

It seems that the proposed method does not improve training on ImageNet in Appendix B. Could you report the final performance? This would be more readable than only the curves which seem identical.

Up to 52% fewer epochs but compute 2x more gradients. Do you win something in terms of training time?
Where get the bold improvement in abstract
I find the claim of "need up to 52.1% less epoch" very confusing. It seems cherry-picked in Figure 4 (b) and is more an artifact of the better early performance of A-BSR vs BSR than to reach good performance. Hyper-parameters of the baseline were tuned for final accuracy, not early stop.

---

> ### Author Response · Authors · 2023-03-04
> **Response to reviewer th9n (Q1-Q2)**
>
> Thank you for the review and for bringing up these questions. We have revised the paper accordingly and the main changes are highlighted in blue. Please see our responses below to clarify the main concerns.
>
> **Q1: The proposed method exhibits an improvement in final accuracy and this is to me the most impressive contribution of this method. If this is not due to hyper-parameters tunning on the test metric, this is a strong point for AGENT. This point is more convincing than being faster in terms of iterations.**
>
> A1: (i) For accuracy, we find that our AGENT can provide comparable or improved final accuracy, which is one of the benefits of our AGENT. The final accuracy for RigL-based results is shown in Table 3 and Table 4, and our AGENT's accuracy is not significantly larger than that of RigL when considering the standard deviation. The final accuracy for BSR-Net-based results is shown in Table 5 and our AGENT can bring significantly higher final accuracy.
>
> (ii) For the acceleration, comparison in wall-clock time is important, and comparison in epoch is also widely-used [R1, R2, R3, R4], which potentially paves the way for the acceleration in wall-clock time. Based on the acceleration in epoch number, we can further reduce the wall-clock time via the following ways(discussed in Section 7):
> * We can use sparse gradients in sparse training, which effectively reduces the cost of backward in sparse training and can be easily applied to our method [R5].
> * We can use parallel computing. Since the additional forward and backward over the old model parameters are fully parallelizable, we can view it as doubling the mini-batch size [R1].
> * We can follow the idea of SAGA and store gradients for each sample. Then, we do not need extra forward and backward steps, saving the wall-clock time [R6].
>
> [R1] Zeyuan Allen-Zhu, et al. Variance reduction for faster non-convex optimization. ICML 2016.
>
> [R2] Niladri Chatterji, et al. On the theory of variance reduction for stochastic gradient Monte Carlo. ICML 2018.
>
> [R3] Difan Zou, et al. Subsampled stochastic variance-reduced gradient Langevin dynamics. UAI 2018.
>
> [R4] Ashok Cutkosky, et al. Momentum-based variance reduction in non-convex SGD. NeurIPS 2019.
>
> [R5] Melih Elibol, et al. Variance reduction with sparse gradients. arXiv preprint arXiv:2001.09623, 2020.
>
> [R6] Aaron Defazio, et al. Saga: A fast incremental gradient method with support for non-strongly convex composite objectives. NeurIPS 2014.
>
> **Q2: The analysis of the covariance between gradients displayed in Figure 1 and explained in Section B.6 could be more realistic. It would be more interesting to study these covariances over the course of the training instead of taking a fully trained model. Moreover, only adding random noise to the trained weights is not reasonable. I would suggest improving the covariance analysis of the gradient.**
>
> A2: We follow the reviewer's suggestion and add more experiments to analyze the covariance between gradients, which is summarized in Section B.6.
>
> **Correlation over the course of training:**
> (i) We analyze the gradient correlation during the standard training of sparse VGG-C on CIFAR-10 using SET-ITOP. The results are summarized in Figure 27.
>
> At early training stage,
> * Both dense and sparse training have relatively lower gradient correlation.
>
> After early training stage,
> * For 50\% sparsity, the correlation between dense and sparse training are close.
> * For 80\% sparsity, sparse training tends to have a lower correlation compared to dense training, especially in late training stages.
> * For 90\% and 95\%, sparse training also gives lower relevance than dense training, and the differences become larger with increasing sparsity.
>
> (ii) We also analyze the gradient correlation during the standard training of sparse ResNet-50 on CIFAR-100 using RigL. The results are summarized in Figure 28.
> At early training stage,
> * Both dense and sparse training have relatively lower gradient correlation.
>
> After early training stage,
> * For 50\% sparsity, the correlation between dense and sparse training is similar.
> * For 80\% sparsity, sparse training tends to have lower relevance than dense training, especially in late training stages.
> * For 90\% and 95\%, sparse training also gives lower correlation compared to dense training, and the differences become larger with increasing sparsity.
>
> **Correlation of the fully-trained model:**
> * We begin with fully-trained checkpoints from ResNet-50 on CIFAR-100 with RigL and SET at 0\%, 50\%, 80\%, 90\%, and 95\% sparsity.
> * We calculate and store the gradient of each weight on all training data.
> * Then, we add Gaussian perturbations (std = 0.015) to all the weights and calculate the gradients again.
> * Lastly, we calculate the correlation between the gradient of the new perturbed weights and the old original weights.
> * As shown in Table 8, the correlation decreases with increasing sparsity, which indicates a weaker correlation in sparse training.

---

> > ### Author Response · Authors · 2023-03-04
> > **Response to reviewer th9n (Q3-Q6)**
> >
> > **Q3: The theory does not give a hint at how to tune the two new hyper-parameters except that they should lie in [0,1].**
> >
> > A3: The good hyper-parameter needs to satisfy the assumptions.
> >
> > (i) Thus, we can check whether the value of $\gamma c$ meets the assumptions given the dataset information. As shown in Theorem 1 of the Section A.5.1, we need proper choice of $\eta$ and $\gamma c$ so that we can have proper constant $\nu, \mu, \alpha,  \kappa > 0$, which enables $\phi \geq \frac{n \nu}{L N^{\alpha}}$ and $\xi \leq \kappa L$.
> >
> > (ii) For $\eta$, it is defined as $\eta = \frac{\mu n }{L N^{\alpha}} \quad (0 < \mu < 1) \quad and \quad (0 < \alpha \leq 1)$, $\zeta = \frac{L}{N^{\alpha/2}}$.
> >
> > (iii) For $c_s$, its relationship between other parameters is included in Lemma 4 (Section A.4.4), where $b_{m}^s = 0$, $\eta_{t}^s = \eta$, $\zeta_{t}^s  = \zeta$ and $b_{t}^s = b_{t+1}^s(1+\eta \zeta + \frac{4 c_{s}^2 \eta L^2}{n}) + 2\frac{c_{s}^2 \eta^2 L^2}{n}$, $\Phi_t^s = \eta - \frac{b_{t+1}^s \eta}{\zeta_{t}} - \eta^2 L - 2b_{t+1}^s\eta^2$, and $\phi \coloneqq \min_{t,s} \Phi_{t}^s$.
> >
> > (iv) In previous variance correction methods, $c_s$ is usually set as 1. In our AGENT, $c_s=\gamma c$ is smaller than 1, leading to smaller $b_{m}^s$, larger $\Phi_t^s$, and larger $\phi$. And larger $\phi$ enables $\phi \geq \frac{n \nu}{L N^{\alpha}}$.
> >
> > **Q4: I find the methodology for tuning the new method and the baseline very weak. Section E.2 only states that the scaling parameter was set to 0.1 without explanation. At the same time, Section B.3 illustrates the good choice of by looking at the test set without mentioning a validation set. Can you also clarify how was tuned?**
> >
> > A4: Intuitively, the scaling parameter $\gamma$ needs to be not close to 1 or 0.
> >
> > (i) The optimal value depends on many factors, i.e., the datasets and model architectures. We find 0.1 works well across different setups. It is possible to get faster convergence if we tune $\gamma$ according to each dataset and model architecture.
> >
> > (ii) We tune the scaling parameter $\gamma$ at a validation set that is randomly chosen from the training set (10%). For simplicity, we use the same $\gamma$ for each scenario and find $\gamma=0.1$ can bring acceleration with comparable accuracy across different cases.
> >
> > **Q5: It seems that the proposed method does not improve training on ImageNet in Appendix B. Could you report the final performance?**
> >
> > A5: (i) We show the training curves of ImageNet-2012 in Figure 4 of Section 6.1.
> > * For 80\% sparsity, the red curve is above the blue curve, demonstrating the acceleration effect of our AGENT, especially in the early stages.
> > * For 90\% sparsity, the red curve is more stable than the blue curve, which shows the stable effect of our AGENT on large data sets.
> >
> > (ii) We also include the final accuracy for ImageNet-2012-based results in Table 4 of Section 6.2. Our method usually maintains accuracy compared to the baselines.
> >
> > **Q6: Up to 52% fewer epochs but compute 2x more gradients. Do you win something in terms of training time? It seems cherry-picked in Figure 4 (b) and is more an artifact of the better early performance of A-BSR vs BSR than to reach good performance.**
> >
> > A6: For the epoch number & clock-time comparison,
> > * Our method achieves acceleration in terms of epoch number, which is also widely used [R1, R2, R3, R4] and potentially pave the way for the acceleration in wall-clock time.
> > * We can further reduce the wall-clock time via the ways discussed in Section 7 and A1 (ii).
> >
> > To avoid worrying about an artifact of better early performance,
> > * We show that our method can achieve acceleration over different training budgets (i.e., number of training epochs), rather than being a pseudo-proposition of better early performance compared to the baseline method.
> > * To demonstrate this, in Section B.11, we add experiments under the different total numbers of training epochs and change the learning rate scheduler accordingly to allow convergence. Take the SET-ITOP as an example.
> > * In the main paper, we follow the baseline paper where the epoch number is 250 and the learning rate scheduler is set as the stepwise learning rate with decay points 125 (i.e., 0.5$\times$250) and 187 (i.e., 0.75$\times$250).
> > * To reduce the epoch number and allow convergence, we set the epoch number as 50 and 100 where the decay points are set as $\{25, 37\}$ and $\{50, 75\}$, respectively.
> > * As shown in Figure 31 in Section B.11, blue curves (our A-SET-ITOP) are usually on top of pink curves (SET-ITOP), implying acceleration from our AGENT.
> >
> > [R1] Zeyuan Allen-Zhu, et al. Variance reduction for faster non-convex optimization. ICML 2016.
> >
> > [R2] Niladri Chatterji, et al. On the theory of variance reduction for stochastic gradient Monte Carlo. ICML 2018.
> >
> > [R3] Difan Zou, et al. Subsampled stochastic variance-reduced gradient Langevin dynamics. UAI 2018.
> >
> > [R4] Ashok Cutkosky, et al. Momentum-based variance reduction in non-convex SGD. NeurIPS 2019.

---

> > > ### Author Response · Authors · 2023-03-04
> > > **Response to reviewer th9n (Q7-Q11)**
> > >
> > > **Q7: Provide extensive explanations on your methodology to tune the learning rate, the smoothing factor $\alpha$, and the scaling factor $\gamma$ for each method. You should convince me that you are not using the test accuracy as a metric to pick $\alpha$ and $\gamma$ and that all methods have their learning rate properly tuned.**
> > >
> > > A7: Our hyperparameter tuning is based on a validation set.
> > >
> > > (i) For CIFAR-10 and CIFAR-100, before training, we randomly separate the training formed by randomly selecting 10% from the training set, separately from the training set.
> > >
> > > (i) For learning rate in baseline methods, we use their default settings. For the learning rate in our method, we mainly tune the initial learning rate, which usually works well between 0.2 and 0.01. Specifically, for BSR-Net-based models, we use 0.01; for RigL-based models, we use 0.2; for ITOP-based models, we use 0.02.
> > >
> > > (ii) For smoothing factor $\alpha$, we follow the default value in [R7] which is set as 0.3. We add some experiments to test the influence of $\alpha$ in Section B.12.
> > > * Apart from 0.3, we further test $\alpha=0.05,0.5,0.9$ in SET-ITOP. The results are depicted in Figure 32.
> > > * We find that the results are not sensitive to the choice of $\alpha$. Thus, we can follow the default 0.3.
> > >
> > > (iii) For scaling factor $\gamma$, we add more results in Section B.3.
> > > * BSR-Net-based results are depicted in Figure 23. We can see that $\gamma=0.5$ leads to model divergence and $\gamma=0.01$ also leads to a slower convergence compared to $\gamma=0.1$.
> > > * SET-ITOP-based results are depicted in Figure 24. We can see that the results of setting $\gamma=0.01,0.5$ are similar to that of $\gamma=0.1$, and the results of setting $\gamma=0.9$ are worse than that of $\gamma=0.1$. This may be due to the fact that 0.9 is too large for the relatively low gradient correlation.
> > >
> > > [R7] Wei Deng, et al. Accelerating convergence of replica exchange stochastic gradient MCMC via variance reduction. ICLR 2021.
> > >
> > > **Q8: Add a more realistic baseline of fixed $c_t$ with good hyperparameter tuning for every experiment to showcase that adaptive re-weighting is crucial, only comparing to $c_t=0.1$ is arbitrary.**
> > >
> > > A8: We add a more realistic baseline of "Fixed $c_t$" with good hyperparameter tuning to show that adaptive re-weighting is crucial in Section B.13.
> > >
> > > (i) Specifically, we further check different $c_t$ in "Fixed $c_t$" and compare their validation accuracy with our A-SET-ITOP.
> > >
> > > (ii) As shown in Figure 33, when $c_t$ is fixed as 0.001, 0.001, and 0.1, the pink curves ( "Fixed $c_t$") are lower than the blue curve (A-SET-ITOP) in the early stages, indicating slower early convergence in "Fixed $c_t$".
> > >
> > > (iii) When fixing $c_t$ as 0.5, 0.8, and 1.0, the whole pink curves ( "Fixed $c_t$") are below the blue curves (A-SET-ITOP), implying slower convergence in "Fixed $c_t$".
> > >
> > > **Q9: Report the clock time for the different methods in every experiment to give a clear view for practitioners, at least in Appendix with a mention in the main text.**
> > >
> > > A9: We summarize the time comparison in Section B.17.
> > > * Our current training time does not have advantages over baseline methods.
> > > * Our training time can be easily reduced by several methods mentioned in A1.
> > >
> > > **Q10: Report the results on ImageNet in the main paper.**
> > >
> > > A10: We have added the results on ImageNet in the main paper.
> > >
> > > (i) The convergence speed and stability comparisons are included in Section 6.1 shown in Figure 4.
> > >
> > > (ii) The final accuracy comparison is included in Section 6.2 as shown in Table 4.
> > >
> > > **Q11: Minor (typos and improvements).**
> > >
> > > A11: We have followed the reviewers' suggestions and revised them in the new version.

---

### Decision · Action_Editors · 2023-03-30

**Recommendation:** Reject

**Comment:**

All three reviewers were leaning to reject. Reviewers have raised quite a few concerns, e.g., the setting of the experiments was quite restrictive, the acceleration method may not be practical enough, and the theoretical justifications are not convincing. The authors may need to carefully address the reviewers' concerns and provide more robust evidence of the method's effectiveness if they consider a future resubmission.




**Audience:**

It is possible that some audiences may be interested in knowing the findings of this paper. However, the concerns raised by the reviewers regarding the theoretical justifications and the limitations of the experimental results may limit the potential interest of the audience. The lack of convincing evidence to support the claims made in the paper may also make it less appealing to researchers and practitioners in the field.

**Claims And Evidence:**

It appears that the claims made in the submission may not be supported by accurate, convincing, and clear evidence. In particular,  reviewers expressed concerns about unjustified and incorrect claims about the theoretical properties of the proposed method, which were not well addressed by the authors. Also,  the experimental section could be weak and the performance claimed in the abstract might be too strong and misleading for practitioners. Furthermore, the new experiments did not confirm the primary motivation for applying the new methods to sparse training.